# MIABench: Full-Pipeline Evaluation of Membership Inference Attacks

## Abstract

Membership inference attacks (MIAs) are widely used to assess a model's vulnerability to privacy leakage by determining whether specific data instances were part of its training set. Despite their significance as a privacy metric, existing evaluations of MIAs are often limited to isolated and inconsistent scenarios, hindering comprehensive comparisons and practical insights. To address this limitation, we analyze the full pipeline of training models and conducting MIAs, and present a comprehensive benchmark for evaluating various MIA methods on deep learning models. We establish a reproducible benchmark suite with code and models, leaderboards, detailed insights into the mechanisms of different MIA approaches, and practical guidance for selecting and applying MIAs effectively. This work enhances the understanding and application of MIAs, providing a solid foundation for advancing privacy-preserving machine learning research.

## 1 Introduction

The past decade has witnessed rapid advancements in machine learning. Overparameterized deep neural networks (Vaswani et al., 2017; Devlin et al., 2019; Achiam et al., 2023) have pushed the boundaries of numerous application domains. However, unlike classical machine learning methods, whose solutions can often be expressed analytically, training deep neural networks involves solving ultra-high-dimensional nonconvex optimization problems, where the optima can only be approximated through numerical methods. As a result, despite their impressive performance, these models are often regarded as black boxes with limited interpretability. In other words, deep neural networks may rely on feature representations vastly different from those used by humans, leading to various reliability concerns (Goodfellow et al., 2015; Zhang & Zhu, 2018; Zhu et al., 2019).

In this work, we focus on the issue of membership inference attacks (MIA) (Shokri et al., 2017), which address the potential risks of data privacy leakage in machine learning. Specifically, an MIA algorithm, denoted as $f$, solves a hypothesis testing problem: it determines whether a given data instance $x$ was included in the training set of a model $M$. A higher attack success rate indicates a greater risk of data privacy leakage in the model. Thus, the performance of MIA serves as a critical metric for assessing the privacy-preserving capabilities of machine learning algorithms and models.

Given the diversity of MIA methodologies and privacy requirements across different application scenarios, it is both important and challenging to *fairly and comprehensively* benchmark the performance of various MIA methods. These challenges stem from different factors in different stages of the pipeline consisting of training a machine learning model and conducting membership inference attacks. However, existing works usually conduct their evaluations in some specific scenarios, making their results incomparable. Therefore, benchmarking different MIAs under different settings can enable us to build a broader view of how different categories of MIA work and how to pick MIA methods wisely under different circumstances.

In this work, we focus on MIA methods targeting classical deep learning rather than generative models, as the former are more extensively studied and better understood. Within this scope, we conduct a comprehensive investigation of their performance across diverse scenarios. In addition, we provide in-depth analyses of several consistent and intriguing patterns observed in the results. Finally, we propose a general guideline for practitioners to effectively select appropriate MIA methods for real-world applications. Our contributions are summarized as follows:

1. **Benchmark Suite:** We provide a publicly available suite, including code, datasets and model zoos, to support reproducible evaluation of MIAs in the full pipeline of model training and privacy evaluation. Notably, depending on the knowledge of data membership, we demonstrate the attacking and auditing mode of MIA methods to ensure fair and comprehensive evaluation. Altogether, we establish leaderboards to demonstrate the performance of MIAs.

2. **Insights into MIA Mechanisms:** We demonstrate the crucial role of the generalization gap in current MIA performance and the high agreement among the top-performed MIA methods. We also highlight the importance of selecting appropriate hyperparameters for MIA methods.

3. **MIA Selection Guidance:** Based on the observations in our benchmark, we offer guidance to practitioners for selecting appropriate MIA methods tailored to different tasks.

## 2 RELATED WORKS AND SCOPE OF THE BENCHMARK

Table 1: An overview of different MIA methods (details in Appendix D) and their properties.

| Technique | LiRA | RMIA | Metric MIA | Quantile | Merlin | MLLeak1 | MLLeak3 | Shadow | BlindMI |
|---|---|---|---|---|---|---|---|---|---|
| Scalar Threshold | ✓ | | ✓ | | ✓ | | ✓ | | |
| Sample-wise Threshold | | ✓ | | ✓ | | | | | |
| Binary Classifier | | | | | | ✓ | | ✓ | |
| Pairwise-distance | | ✓ | | | | | | | ✓ |
| Shadow Model | ✓ | ✓ | ✓ | | ✓ | ✓ | | ✓ | |

Membership inference attacks (MIAs) have emerged as critical concerns in machine learning privacy, serving as powerful tools for privacy auditing. Extensive research efforts have been dedicated to comprehending and enhancing MIAs across diverse scenarios. *In this study, our benchmark concentrates on MIAs targeting deep learning classifiers, presuming black-box access for the adversary*. Specifically, the attacker is limited to accessing the output logits of the model, as opposed to the model parameters and their gradients. While white-box MIAs (Chen et al., 2022; Watson et al., 2022; Leemann et al., 2024) present a more severe threat to privacy, black-box MIAs align more closely with real-world scenarios, especially given the increasing prevalence of machine learning as a service (MLaaS) in various applications. Mathematically, the MIA method we investigate in this work can be formulated as a hypothesis testing function $m$ that maps the data instance $x$ and the corresponding output of the target model $f(x)$ to a binary output $m(x, f(x)) \in \{0, 1\}$: with 1 indicating that $x$ is in the training set for model $f$ (i.e., member) and 0 otherwise (i.e., non-member).

There are several popular strategies adopted by many MIA methods. First, threshold control selects *scalar thresholds* on a chosen metric, such as likelihood ratio or modified entropy of the model outputs, to separate members from non-members (Song & Mittal, 2021; Carlini et al., 2022). A natural extension of scalar threshold is *sample-wise threshold*, which computes a threshold for each sample (Zarifzadeh et al., 2024; Bertran et al., 2024). In contrast to a scalar threshold, several MIAs (Shokri et al., 2017; Salem et al., 2018) employ *binary classifiers* to directly distinguish members and non-members. In addition, Hui et al. (2021); Zarifzadeh et al. (2024) leverage *pairwise distances* between data points to better distinguish between member and non-member data. Furthermore, the *shadow model* (Shokri et al., 2017; Ye et al., 2022) is a popular technique that trains separate shadow models using the same distribution as the original training dataset. With the information on training members for shadow models, we can better design the criteria to detect members. Because shadow models are trained to mimic the behavior of the target model, the same criteria are then used for the target models. We summarize the key techniques of different MIAs *within the scope of our investigation* in Table 1 below. More details about these methods are deferred to Appendix D.

## 3 BENCHMARKING MEMBERSHIP INFERENCE ATTACKS

### 3.1 PIPELINE AND CHALLENGES

The full pipeline of training a machine learning model and conducting membership inference attacks (MIA) is demonstrated in Figure 1, including **(1)** sample data to form a training set; **(2)** select the model architecture; **(3)** use the training set to train the target model; **(4)** tune the hyper-parameters

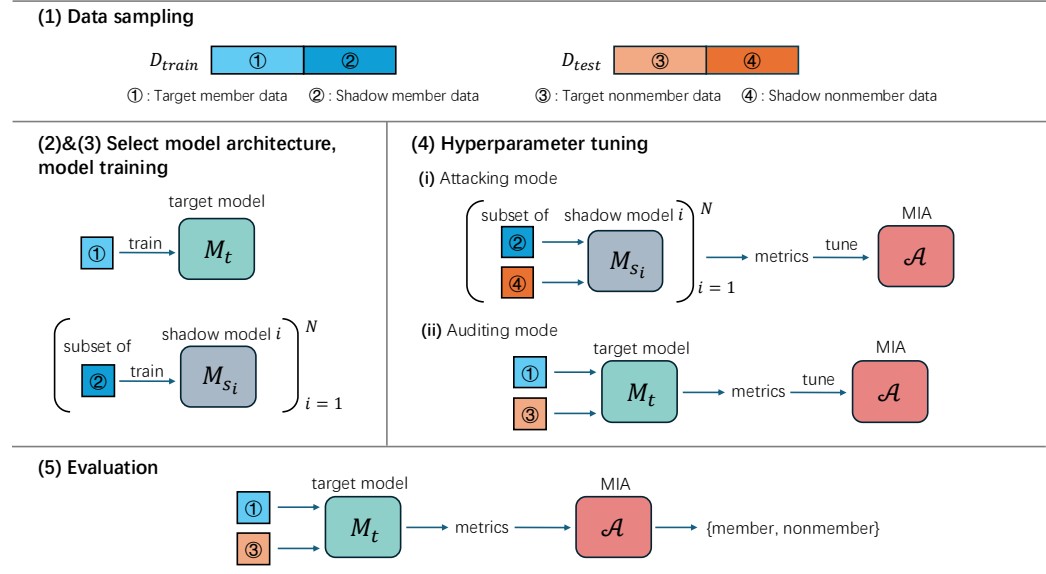

Figure 1: Pipeline of MIA. The pipeline consists of five steps: **(1)** data sampling; **(2)** select the model architecture; **(3)** model training; **(4)** tune the hyper-parameters of the MIA method; **(5)** evaluate the target model on a data set consisting of both member and non-member data. Two distinct scenarios correspond to deploying MIAs for **attacking** or **auditing**, respectively. In the context of attacking mode, MIA serves as a tool for **attackers** to determine whether a specific data point was used in the training of a target model. In the context of auditing mode, **model owners** employ MIA to audit the target model's privacy-preserving capabilities.

of the MIA method; **(5)** evaluate the target model on a data set consisting of both members and non-members. Each of these five phases may have different settings in various application scenarios, making it challenging to *fairly and comprehensively* benchmark MIA methods.

**Data.** Recent studies (Niu et al., 2024; 2025) highlight the significant impact of data properties on MIA performance, including training data and data being inferred. In addition, out-of-distribution data, such as ambiguous or mislabelled data, further complicate the evaluation of MIA methods.

**Models.** The performance of MIA methods is highly dependent on the characteristics of the target model, especially the model capacity reflected by the architectures and number of parameters. Intuitively, models with limited capacity struggle to encode data-specific information, while highly expressive models are more likely to leak such information, thereby facilitating MIAs.

**Training Algorithms.** Key factors of training algorithms, including the initialization, the training phases and the optimization methods, can affect how susceptible the model is to leaking data privacy, because all of them can affect how the model encode data-specific information.

**Hyperparameter Selection.** Hyperparameters play a crucial role in determining the effectiveness of MIA methods and should be chosen *carefully but fairly*. In this context, the availability of data membership for evaluation is crucial. In Section 3.2, we discuss two modes for selecting the optimal hyperparameters depending on the availability of data membership for evaluation.

**Evaluation Metrics.** The performance of MIA methods can be evaluated by various metrics, such as precision, recall, and area under the curve (AUC), making it inherently a multi-objective optimization problem. Different application scenarios often prioritize different metrics, which may lead to different preferences for MIA methods. Some cases would prefer a low false negative rate, such as using machine unlearning (Bourtoule et al., 2021) to remove the influence of sensitive data. Conversely, some cases would prefer a low false positive rate, such as extracting training data from generative models for reuse in fine-tuning. In many other cases, it is necessary to balance both false positive and false negative rates, making metrics such as the F1 score a natural choice.

## 3.2 ATTACKING MODE AND AUDITING MODE OF MIAS

One key stage in the pipeline as shown in Figure 1 is the selection of MIA hyperparameters, which depends critically on whether membership information of the inference data is available. These two scenarios correspond to the use of MIAs as either privacy attacks or auditing tools. In the attack scenario, attackers lack access to member and non-member datasets, making hyper-parameter selection a major challenge. In contrast, when used as an auditing tool, certain MIAs can leverage membership information to optimize hyperparameters, achieving stronger performance on fixed datasets and models. To ensure fair and reproducible comparisons, it is essential to evaluate different MIAs under consistent conditions. In this regard, we summarize two practical MIA scenarios below and provide pseudocode for both in Appendix E.

**Attacking Mode.** In this scenario, adversaries use MIA to determine whether a specific data point was part of the target model's training data. As attackers lack access to the target model's actual training set, to select hyperparameters like scalar thresholds, they train shadow models using shadow data that follows a distribution similar to that of the target model's training data. The hyperparameters are then tuned using the performance *of these shadow models*. As shown in Figure 1, both the original training and test sets are partitioned into dedicated subsets for the target model and the shadow models. The MIA's final performance is evaluated on the target model's test set.

**Auditing Mode.** In this scenario, model owners employ MIA as a privacy audit tool to evaluate the information leakage of their model. With explicit knowledge of which data points are members (training data) and non-members, certain MIA methods, such as LiRA, RMIA, and Quantile, can directly use this information to select optimal hyperparameters (e.g., thresholds) and improve performance. As illustrated in Figure 1, we use the same member and non-member datasets for both hyperparameter tuning and final evaluation.

One important application to use MIAs for privacy auditing is machine unlearning (MU). MU fine-tunes pretrained models to remove the influence of data to forget (i.e. forget data) meanwhile preserving model predictions on the remaining training set (i.e. retain data). For fair comparison, we follow the same data split as in Figure 1 for MU except that the forget data, which is part of the original training set but should be unlearned during finetuning, is considered as non-members in the evaluation phase. This is consistent with existing MU literature (Cao & Yang, 2015; Fan et al., 2023; Tarun et al., 2023; Huang et al., 2024).

Finally, we need to point out that the discrepancy between the attack mode and auditing mode only exists for MIAs with hyperparameters to distinguish members and non-members. MIAs without such hyperparameters, such as the ones using shadow models (Shokri et al., 2017) or using binary classifiers Salem et al. (2018), use exactly the same workflow for both attacking and auditing purposes. Therefore, we do not distinguish attacking or auditing mode for these MIA methods.

## 4 EXPERIMENTS

### 4.1 EXPERIMENTAL SETUPS

We comprehensively evaluate different aspects that may affect the MIA performance as below.

**MIA Methods.** We consider a variety of MIA methods in this work, including **LiRA**, **RMIA**, **Metric MIA**, **Quantile**, **Merlin**, **ML-Leaks**, **Shadow**, and **BlindMI**. Specifically, for Metric MIA, we employ three distinct metrics **Entropy, Modified Entropy and Confidence**; for ML-Leaks, we implement its two settings **MLLeak1, MLLeak3**; and for BlindMI, we incorporate the four techniques **One-class, Diff-W, Diff-Single and Diff-bi**. Further details regarding the literature, the implementation and configuration of these methods are provided in Appendix D.

**Data.** We consider vision data, e.g., CIFAR10 (Krizhevsky et al., 2009), CIFAR100 (Krizhevsky et al., 2009), ImageNet100 (Russakovsky et al., 2015). We also consider the superclass and the mislabel scenarios. The details are deferred to Appendix F.2.

**Architectures.** We consider neural networks of different sizes, e.g., 4-layer CNN (Krizhevsky et al., 2012), VGG-11 (Simonyan & Zisserman, 2015), ResNet-18 (He et al., 2016), ResNet-34, ResNet-50, WideResNet-28-2 (Zagoruyko & Komodakis, 2016), and Swin Transformer-Tiny (Liu et al., 2021). The size of each model is presented in Table 5 of Appendix F.3.

**Training Algorithms.** We include various training algorithms, including training from scratch and fine-tuning. We also investigate several privacy-enhancing algorithms, including differential private SGD (DP-SGD) (Abadi et al., 2016) and machine unlearning algorithms (Bourtoule et al., 2021). For each training algorithms, we monitor the MIA performance in the whole training duration. Specifically, for machine unlearning methods (SalUn (Fan et al., 2024) and SFR-on (Huang et al., 2024)), we monitor the MIA performance across pretraining, unlearning and compare them with retraining by retained data. We report the MIA performance on retain data, forget data and test data.

**Hyperparameter Selection.** We consider both attacking mode and auditing mode introduced in Section 3.2 to select MIA hyperparameters in different contexts.

**Evaluation Metrics.** We use TP, FP, TN, FN to represent the number of true positive, false positive, true negative and false negative, respectively. We consider the following metrics for comprehensive evaluation: (1) **Accuracy**: the overall accuracy $\frac{\text{TP}+\text{TN}}{\text{TP}+\text{TN}+\text{FP}+\text{FN}}$; (2) **Prec@P**: precision on the positive cases $\frac{\text{TP}}{\text{TP}+\text{FP}}$; (3) **Rec@P**: recall on the positive cases $\frac{\text{TP}}{\text{TP}+\text{FN}}$; (4) **F1@P**: F1-score on the positive cases $\frac{2\times\text{Prec@P}\cdot\text{Rec@P}}{\text{Prec@P}+\text{Rec@P}}$; (5) **Prec@N**: precision on the negative cases $\frac{\text{TN}}{\text{TN}+\text{FN}}$; (6) **Rec@N**: recall on the negative cases $\frac{\text{TN}}{\text{TN}+\text{FP}}$; (7) **F1@N**: F1-score on the negative cases $\frac{2\times\text{Prec@N}\cdot\text{Rec@N}}{\text{Prec@N}+\text{Rec@N}}$.

For MIAs using threshold to distinguish member and non-members, we also introducing the following popular metrics in evaluating MIA: (1) **AUC**: the area under the ROC curve; (2) **TPR@0.1%FPR**: the true positive rate when the false positive rate is 0.1%. However, these metrics cannot be computed during attacking mode. The calculation of these metrics requires prior knowledge of data membership, which contradicts the fundamental assumption of the attack scenario. Therefore, we only report results on these metrics in the auditing mode.

## 4.2 EXPERIMENTAL RESULTS AND OBSERVATION

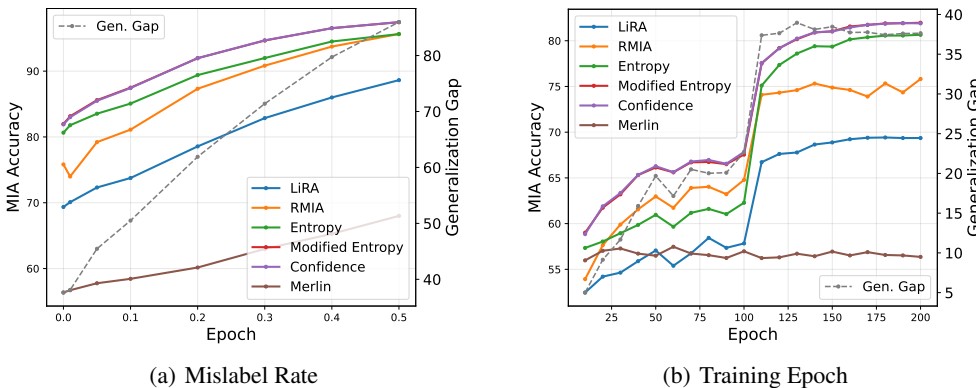

(a) Mislabel Rate          (b) Training Epoch

Figure 2: Curves of MIA accuracy. The model is ResNet-18 trained on CIFAR-100. **(a)** MIA accuracy vs. mislabel rate. The mislabel rates are 0.01, 0.05, 0.1, 0.2, 0.3, 0.4 and 0.5. **(b)** MIA accuracy vs. training epoch. The evaluation is conducted every 10 epochs. Gray dashed lines indicate the generalization gap. LiRA, RMIA, three variants of Metric MIA, and Merlin in auditing mode are tested. Due to prohibitive computational complexity, Quantile is not included in the evaluation. Note that Merlin exhibits trivial performance in both settings.

In this section, we comprehensively study the performance of different MIA methods on each step in the pipeline as discussed in Section 3.1.

### 4.2.1 DATA

We evaluate MIA methods on different datasets, including CIFAR10 (Table 15), CIFAR100 (Table 9) and ImageNet100 (Table 16). These results highlight the broad applicability of the MIA methods under investigation.

Table 2: Average Gap between auditing mode and attacking mode. For Different Dataset, we evaluate MIA performance on CIFAR10(Table 15), CIFAR100(Table 9) and ImageNet(Table 16). For Different Arch, we evaluate MIA performance on 4-layer CNN (Table 10) / VGG-11 (Table 11) / ResNet18 (Table 9) / ResNet-34 (Table 12) / ResNet-50 (Table 13). For Different Algorithm, we evaluate MIA performance of SGD (Table 9) and DP-SGD (Table 20).

| Epochs | Setting | Methods | | | | | | |
|--------|---------|------|------|---------|---------------------|------------|----------|--------|
| | | LiRA | RMIA | Entropy | Modified Entropy | Confidence | Quantile | Merlin |
| 200 | Different Dataset | 2.85 | 2.24 | 3.96 | 3.67 | 3.54 | 0.11 | 1.76 |
| | Different Architecture | 2.41 | 1.98 | 3.01 | 2.30 | 2.33 | 0.04 | 0.70 |
| | Different Algorithm | 2.17 | 1.17 | 3.28 | 2.63 | 2.54 | 0.30 | -1.17 |
| 110 | Different Dataset | 1.73 | 0.63 | 2.19 | 2.50 | 2.35 | 0.14 | 1.26 |
| | Different Architecture | 3.22 | 1.00 | 3.38 | 3.53 | 3.41 | 0.26 | 1.44 |
| | Different Algorithm | 1.18 | 0.51 | 1.98 | 2.09 | 2.06 | 0.21 | 0.62 |
| 10 | Different Dataset | 0.03 | 0.19 | 3.52 | 3.99 | 3.93 | 0.31 | 3.01 |
| | Different Architecture | 0.53 | 0.46 | 7.01 | 6.77 | 6.64 | 0.82 | 5.47 |
| | Different Algorithm | 0.22 | 0.13 | 3.53 | 3.79 | 3.69 | 0.29 | 2.49 |

Additionally, we benchmark MIA performance on CIFAR100 using superclass labels [1], with results presented in Table 18. The findings reveal a performance degradation for most MIAs when models are trained with superclasses. This is because, under the black-box access assumption in our study, training with superclasses reduces the dimensionality of the model's output, thereby limiting the information available to MIA methods and making attacks more challenging.

Furthermore, we investigate how mislabeled data affects the performance of MIAs. Furthermore, we examine the impact of mislabeled data on MIA performance. As shown in Figure 2 (a) and Table 17, mislabeled data generally makes MIAs more effective. A higher proportion of mislabeled data leads to improved MIA performance until it eventually saturates. This is because mislabeled data increases the conditional variance between input and label, forcing the model to learn more data-specific correlations to fit the noisy labels, thereby enhancing the vulnerability to MIAs.

### 4.2.2 ARCHITECTURES

We evaluate MIAs on various models for CIFAR100. The model architectures, from small to big, include a 4-layer CNN (Table 10), VGG-11 (Table 11), ResNet-18 (Table 9), ResNet-34 (Table 12), ResNet-50 (Table 13), and Swin Transformer-Tiny (Table 14).The result indicate that MIA performance varies among different architectures. The underlying reasons for this phenomenon will be further explored in Section 5.

### 4.2.3 TRAINING ALGORITHMS

We monitor the performance of various MIAs throughout the training process. For each setting considered, we report MIA performance at the early, intermediate, and final phases of training. Figure 2 (b) provides a detailed view of the fine-grained trends in MIA performance during training. These results consistently demonstrate that the performance of most MIA methods improves as the training progresses, suggesting that the model increasingly encodes information from the training data, thereby making it more susceptible to MIAs.

We also examine the impact of using a pretrained model on MIA performance. As shown in Table 19 and Table 13, finetuning a pretrained model reduces the model's vulnerability to MIAs. Compared with training from scratch, finetuning typically employs a smaller learning rate and produces smaller gradients, which implicitly slows down the process of encoding training data features into the model.

Furthermore, we report the performance of MIAs when the model is trained by privacy-enhancing algorithms, such as DP-SGD in Table 20. Our result indicates that DP-SGD can indeed defend models against various MIA methods while guarantee comparable utility.

For machine unlearning, we report results only in auditing mode. As shown in Table 22, most MIA methods exhibit consistent performances across 4 models, including the pretrained model, unlearned models by two MU methods and the retrained model, which indicates the effectiveness of unlearning

---

[1]CIFAR100 consists of 20 superclasses, each containing 5 classes.

Table 3: Mean and Variance of Accuracy for each MIA Methods via Different Architecture and Datasets.

| Epochs | | | Metric MIA | | | | | | | | | BlindMI | | |
|--------|------|------|---------|---------------------|------------|----------|--------|---------|---------|--------|-----------|--------|-------------|---------|
| | LiRA | RMIA | Entropy | Modified Entropy | Confidence | Quantile | Merlin | MLLeak1 | MLLeak3 | Shadow | One-class | Diff-w | Diff-single | Diff-bi |
| 200 | 65.01 (10.08) | 69.41 (7.72) | 73.86 (9.47) | 75.64 (8.68) | 75.56 (8.67) | 73.29 (10.20) | 56.09 (1.57) | 64.81 (11.20) | 64.69 (9.43) | 58.88 (5.19) | 65.49 (8.97) | 70.21 (10.70) | 68.52 (9.63) | 61.25 (7.85) |
| 110 | 60.95 (6.39) | 66.32 (9.06) | 68.67 (7.95) | 70.53 (8.41) | 70.53 (8.40) | 66.94 (10.05) | 55.64 (1.60) | 54.12 (5.18) | 56.27 (5.56) | 56.58 (5.36) | 63.22 (10.19) | 65.69 (10.91) | 64.05 (9.40) | 59.24 (7.59) |
| 10 | 51.68 (1.15) | 52.47 (1.22) | 57.45 (0.72) | 58.30 (0.69) | 58.28 (0.70) | 52.43 (1.01) | 55.94 (1.07) | 50.55 (0.00) | 50.33 (0.26) | 50.13 (0.09) | 50.51 (0.73) | 50.54 (1.02) | 50.79 (0.74) | 50.55 (0.78) |

Table 4: Mean and Variance of AUC and TPR@0.1 % FPR for each MIA Methods via Different Architecture and Datasets.

| Epochs | Metrics | Methods | | | | | | |
|--------|---------|--------------|--------------|--------------|------------------|--------------|--------------|--------------|
| | | LiRA | RMIA | Entropy | Modified Entropy | Confidence | Quantile | Merlin |
| 200 | AUC | 0.659(0.101) | 0.764(0.097) | 0.731(0.116) | 0.755(0.098) | 0.751(0.099) | 0.749(0.109) | 0.522(0.029) |
| | TPR@0.1%FPR | 0.85(0.764) | 6.9(4.466) | 4.7(3.571) | 4.9(3.851) | 4.8(3.789) | 0.33(0.227) | 0.97(0.860) |
| 110 | AUC | 0.635(0.083) | 0.725(0.120) | 0.673(0.115) | 0.695(0.110) | 0.693(0.112) | 0.686(0.114) | 0.517(0.028) |
| | TPR@0.1%FPR | 0.49(0.5) | 5.2(3.2) | 4.3(2.8) | 4.4(3.1) | 4.3(3.0) | 0.35(0.26) | 0.97(0.98) |
| 10 | AUC | 0.517(0.018) | 0.532(0.019) | 0.512(0.009) | 0.530(0.013) | 0.529(0.012) | 0.528(0.013) | 0.507(0.004) |
| | TPR@0.1%FPR | 0.1(0.08) | 1.1(0.89) | 2.1(1.4) | 2.2(1.7) | 2.2(1.7) | 0.4(0.5) | 0.9(0.9) |

process. In Table 21, we focus more on MIA performance on the forget set which we consider as non-members. The results indicate that metric-based MIAs (including LiRA, RMIA, MetricMIA, Quantile and Merlin) generates similar results for unlearned and retrain models, whereas classifier-based MIAs (including ML-Leaks and BlindMI) exhibit large discrepancies. From the perspective of auditing privacy risk, this indicates that metric-based MIAs, which rely on thresholding statistics, may overestimate unlearning effectiveness. In contrast, classifier-based MIAs expose residual differences and enable a more stringent and informative assessment by learning a decision function between members and non-members.

### 4.2.4 HYPERPARAMETER SELECTION

We assess the performance of MIAs in both attacking mode and auditing mode. For MIA methods requiring setting threshold, selecting appropriate hyperparameters in auditing mode yields better results than in attacking mode. In Table 2, we report the accuracy gap between the two modes for these methods. Table 2 shows that, across different epochs and settings, the Quantile MIA method exhibits the smallest gap. In contrast, Metric MIAs (Entropy, Modified Entropy, Confidence) tend to have larger gaps, indicating a higher sensitivity of their performance on hyperparameters.

### 4.2.5 EVALUATION METRICS

We evaluate the performance of MIA methods across various datasets (CIFAR10(Table 15), CIFAR100(Table 9) and different architecture(4-layer CNN (Table 10) / VGG-11 (Table 11) / ResNet-18 (Table 9) / ResNet-34 (Table 12) / ResNet-50 (Table 13)). Tables 3 and 4 present the mean and variance of accuracy (maximum of auditing and attacking mode), AUC, and TPR@0.1%FPR under the aforementioned scenarios. The results indicate that Metric MIA methods provide the best accuracy among all methods across every epoch. Regarding AUC and TPR@0.1%FPR, the RMIA method outperforms others. Moreover, RMIA maintains a high AUC with smaller variance compared to other methods under the AUC curve, demonstrating its stability across different scenarios.

Furthermore, our extensive results reveal that various MIA methods are more likely to achieve near-perfect performance in Rec@P and Prec@N compared to other metrics. This suggests that these methods exhibit a bias toward predicting inputs as members, emphasizing the importance of using multiple metrics to ensure a comprehensive and balanced evaluation.

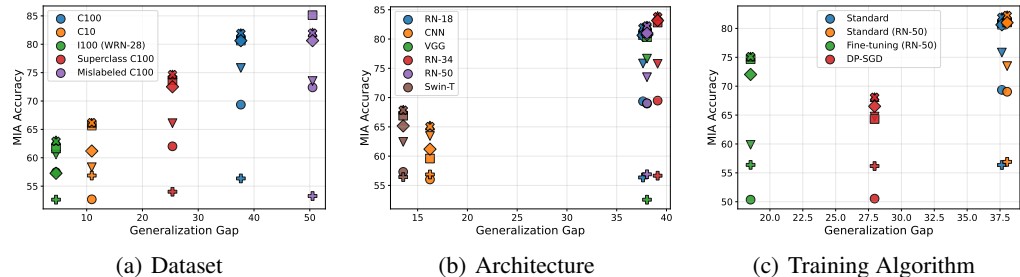

|  (a) Dataset | (b) Architecture | (c) Training Algorithm |

Figure 3: Relationship between MIA accuracy and the degree of overfitting. The analysis is conducted from different aspects, e.g., **(a)** different datasets, "C" denotes CIFAR, "I" denotes ImageNet, **(b)** different architectures, and **(c)** different training algorithms. The x-axis is the generalization gap. A larger generalization gap indicates severer overfitting. The y-axis is the accuracy of MIA methods. Different colors represent different settings. Different markers represent different MIA methods. LiRA (●), RMIA (▼), Entropy (■), Modified Entropy (▲), Confidence (✕), Quantile (◆), and Merlin (✚) in auditing mode are tested here.

## 5 FURTHER DISCUSSIONS

### 5.1 OVERFITTING LEADS TO MIA VULNERABILITY

We illustrate the relationship between MIA accuracy and the degree of overfitting in Figure 3. The analysis is conducted from different aspects, e.g., different datasets, architectures, and training algorithms, with the same settings as those in Section 4.2. Note that the degree of overfitting is indicated by the generalization gap, i.e., the gap between the training accuracy and the test accuracy. Detailed performance statistics of different settings are summarized in Table 6 of Appendix G.1.

As depicted in Figure 3, the accuracy of most MIA methods exhibits a general upward trend as the generalization gap of target models increases. In Figure 2, we also plot the generalization gap and observe that it is strongly correlated with the MIA performance. Therefore, many of the observations in Section 4.2, including analyses across different datasets, models, and algorithms, can be understood through the lens of overfitting. Specifically, overfitting plays a significant role in increasing the model's vulnerability to MIAs.

The underlying mechanism for this association can be attributed to the working principle of existing MIA methods: these attacks determine the membership of a given data sample by leveraging discrepancies in the output distributions of member and non-member data. Critically, such distributional discrepancies are closely correlated with the extent of model overfitting: overfitting causes models to "memorize" idiosyncrasies of their training data rather than learning generalizable patterns, thereby amplifying the distinction between the outputs of member and non-member samples. Consequently, models afflicted by more severe overfitting will exhibit larger distributional gaps between member and non-member data, rendering them more susceptible to successful MIAs.

### 5.2 AGREEMENT OF TOP-PERFORMED MIA METHODS

We investigate the agreement among the results by strong MIA methods. Based on the analyses in Section 4.2, we pick four strong and methodologically distinct MIA methods: Quantile, Modified Entropy (Metric MIA), RMIA and Diff-w (BlindMI). Figure 4 illustrates the agreement results across different architectures on the CIFAR100 dataset. As depicted in the figure, a strong agreement is observed among these MIA methods. The results indicate that top-performed MIAs generate very similar instancewise predictions. In this context, if we ensemble these MIA methods together by adding them in the descending order of performance, then we can hardly see any improvement. In practice, we can choose either of these MIAs to get satisfactory results.

### 5.3 TAKE-AWAY: A GUIDELINE TO CHOOSE MIA METHODS TO EVALUATE PRIVACY

Membership Inference Attacks (MIAs) have emerged as a popular and critical tool for evaluating the privacy risks of neural network models. Effectively leveraging MIAs for privacy attacking and auditing necessitates a nuanced understanding of their varied performance and applicability under

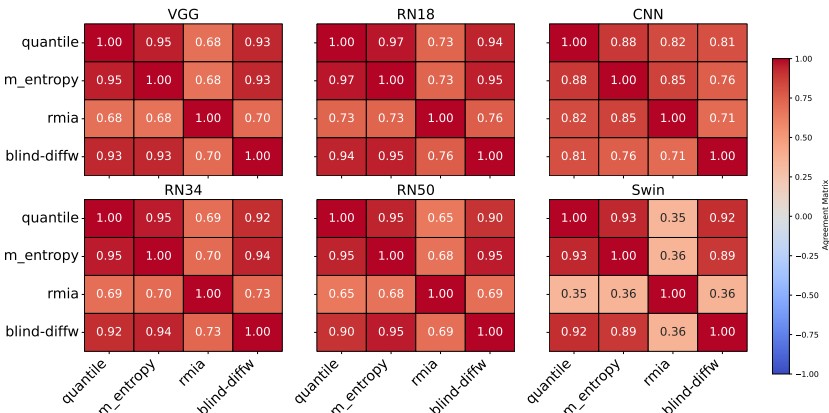

Figure 4: Agreement between different MIAs under CIFAR100 dataset with different architecture

different settings. Drawing from our comprehensive benchmark, this section offers a practical guideline for selecting the most appropriate MIA method for various scenarios.

1. Overfitting is crucial for the performance of all MIA methods. Our benchmark indicates that the generalization gap is a consistent indicator of MIA vulnerability. We can introduce some overfitting mitigation mechanisms during training to protect data privacy.

2. Even with a consistent generalization gap, the performance of MIA methods can vary significantly. There is not yet a single MIA universally optimal across all settings. Specifically, our benchmark demonstrates that Metric MIA (especially Modified Entropy) or RMIA exhibits the best performance in a majority of scenarios. To audit the effectiveness of machine unlearning, we recommend using classier-based MIAs in contrast to metric-based MIAs.

3. Hyperparameter selection also plays a crucial role in the performance of MIA methods. When we use MIA for privacy attack and have no membership information, we should utilise shadow models to select hyperparameters. Based on the attacking mode and auditing mode introduced in Section 3.2, we conclude that MetricMIA (especially Modified Entropy) achieves the best performance in the auditing mode. However, we see notable performance degradation for MetricMIA in the attack mode, where RMIA or Quantile MIA obtain the superior performance.

4. There is no perfect MIA, as we always see a trade-off between false positive and false negative. Ensembling multiple top-ranked MIA methods usually does not lead to improvement, as these MIA methods have consistent instancewise predictions and similar performance.

## 6 CONCLUSION

In this paper, we present a comprehensive study of Membership Inference Attacks (MIAs) conducted within the full pipeline of machine learning model training and privacy evaluation. Our experimental design meticulously encompasses various critical aspects, including data, model architectures, training algorithms, hyperparameter selection and evaluation metrics. Based on our experimental results, we provide practicable guidance for practitioners to select the most suitable MIA method for their specific needs across general scenarios. Furthermore, we introduce an available and expandable suite designed to empower practitioners to conveniently evaluate various MIA methods across diverse scenarios.

Our benchmark results also empirically demonstrate the underlying mechanisms of MIAs. Specifically, we find that for current MIA methods, the generalization gap is consistently a good indicator for MIA performance. Furthermore, the results from the strongest existing MIA methods exhibit a high degree of agreement, indicating performance saturation.

Given that our benchmark work explores MIAs across multiple dimensions, the experimental complexity is considerably high, which consequently limits our exploration in other axes. Therefore, our benchmark presented in this paper only focuses on vision data and traditional classification models. Extending our benchmark to multimodal data and large generative models will be our future work.

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

## A  ETHICS STATEMENT

This work presents a benchmark for membership inference attacks (MIAs), which are designed to assess a model's susceptibility to privacy leakage. All experiments are conducted on publicly available datasets and standard deep learning models, without involving sensitive or personally identifiable data. The purpose of this benchmark is to improve the evaluation and understanding of MIAs, thereby advancing privacy-preserving machine learning rather than enabling misuse. By providing reproducible tools and practical guidance, our study aims to promote responsible development of trustworthy AI systems.

## B  REPRODUCIBILITY STATEMENT

To ensure reproducibility, we will include anonymized supplementary materials containing complete algorithm implementations, experiment scripts for executing all the experiments.

## C  THE USE OF LLMS

Large language models (LLMs) were employed only as writing assistants, restricted to surface-level editing such as grammar correction, clarity improvement, and formatting. LLMs were not used to generate research ideas, claims, analyses, or conclusions. All text refined with LLM assistance was carefully reviewed and, when necessary, rewritten by the authors. The authors bear full responsibility for the final content of this paper.

## D  DETAILS ABOUT MIA METHODS

In our benchmark, we consider MIA methods targeting classical deep learning as introduced below:

- **Shadow:** Shadow (Shokri et al., 2017) is the first membership inference attack method. It first trains shadow models using datasets drawn from the same distribution as the target model's training data, enabling them to replicate its behavior. Since we know the training set for these shadow models, we then then use the shadow models' outputs on their member and non-member data as samples to train a binary classifier to map the model's output to the membership prediction.

- **ML-Leaks:** ML-Leaks (Salem et al., 2018) simplifies Shadow (Shokri et al., 2017) in three different ways. The first one uses one shadow model and train one attack model; the second one further improved by training shadow model with a different dataset; the third method discard the shadow model technique and try to find a threshold depending on exact requirements. Given the impact of the shadow model's training dataset on MIA performance, the second method introduces further complexities. Therefore, in this paper, we mainly consider the first simplification method (**MLLeak1**) and the third method (**MLLeak3**).

- **LiRA:** LiRA (Carlini et al., 2022) trains shadow models in the same manner as Shokri et al. (2017). It then calibrates confidence scores of the classifier for member and non-member, and applies a likelihood-ratio statistical test to infer membership in the target model's training data.

- **Blind MI:** Blind MI (Hui et al., 2021) is based on two key insights. First, non-member samples can be regarded as a feature, and machine learning methods such as SVM can be utilized to learn this feature for classification, which gives rise to the **BlindMI-oneclass** method. The second insight is that removing or adding a non-member to a dataset shifts the dataset away from or towards non-members in the hyper-dimensional space. Consequently, the authors propose the Blind-Diff method. The paper lists three variants **BlindMI Diff-W, BlindMI Diff-Single, BlindMI Diff-bi**, which consider different ways to build the original member and non-member datasets.

- **Metric MIA:** Metric-based MI (Song & Mittal, 2021) considers three different scores: **confidence, entropy, modified entropy**. These scores are believed to distinguish between member and non-member samples. For each label, the method calculates its corresponding score and determines a threshold based on the shadow model for the attack.

- **Merlin:** Merlin (Jayaraman et al., 2020) based on the intuition that, as a result of overfitting, member records are more likely to be near local minima than non-member records, which suggests

that for members, loss is more likely to increase at perturbed points near the original, whereas it is equally likely to increase or decrease for non-members. Hence, it add small perturbation to the input and use the output differences as score to distinguish the member and non-member data.

- **Quantile:** Quantile MIA Bertran et al. (2024) exploits the empirical observation that training losses are typically lower than test losses. It trains a quantile-regression model to estimate the quantile score of each sample and employs this score to discriminate members from non-members.

- **RMIA:** RMIAZarifzadeh et al. (2024) is an extension of LiRa, it proposes the Pairwise Likelihood ratio as scores to distinguish the member and non-member data. Specifically, pairwise likelihood ratio tests testing the membership of a data point relative to another data point. Hence, it is more robust with low computational overhead.

# E PSEUDOCODE

We provide the pseudocode of the pipeline of MIA in Algorithm 1, with unique segments for attacking and auditing modes highlighted in red and green, respectively.

---

**Algorithm 1** Pipeline of MIA with Attacking / Auditing Mode.

---

**Input:** MIA method $\mathcal{A}$; Training dataset $D_{train}$; Test dataset $D_{test}$; Number of shadow models $N$; Split rate for shadow dataset $r_1 = 0.5$; Sampling rate for sub-shadow dataset $r_2 = 0.8$.
    *# Data sampling*
1: Target member data $D_{tm}$, shadow member data $D_{sm} \leftarrow$ Split $D_{train}$ with split rate $r_1$
2: Target non-member data $D_{tn}$, shadow non-member data $D_{sn} \leftarrow$ Split $D_{test}$ with split rate $r_1$
    *# Select model architecture and model training*
3: Train target model $M_t$ on $D_{tm}$
4: **for** $i$ in 1, ..., $N$ **do**
5:    $D_{sm_i} \leftarrow$ Sample $D_{sm}$ with sampling rate $r_2$
6:    $D_{sn_i} \leftarrow$ Sample $D_{sn}$ with sampling rate $r_2$
7:    Train shadow model $M_{s_i}$ on $D_{sm_i}$
8: **end for**
    *# Hyperparameter tuning*
9: Collect metrics from $M_t$ on $D_{tm}$ and $D_{tn}$ / $\{M_{s_i}\}_{i=1}^N$ on $\{D_{sm_i}\}_{i=1}^N$ and $\{D_{sn_i}\}_{i=1}^N$
10: Tuning the hyperparameters in $\mathcal{A}$ with the collected metrics
    *# Evaluation*
11: Get the prediction results of $D_{sm}$ and $D_{sn}$ using the tuned $\mathcal{A}$
**Output:** Prediction results (member/non-member)

---

# F ADDITIONAL EXPERIMENTAL SETTINGS

## F.1 EXPERIMENTS SETUPS FOR DIFFERENT MIA METHODS

Below we detail the hyper-parameters used for each MIA method; many values are directly borrowed from the original papers.

For shadow-based attacks(Shadow, LiRA and RMIA), the utility is tpyically correlated with the number of shadow models. However, training shadow models usually incurs substantial resource consumption. Therefore, to ensure a fair comparison with other methods, we set the number of shadow models to five in our experiments. Each shadow model is trained with the identical settings described in Section F.4. All shadow-based attacks use the same shadow model. In Table 24, we also provide a table showing how the utility of shadow-based MIA methods varies with the number of models under the CIFAR-100 ResNet-18 scenario. Additionally, for Metric-based MIA and Merlin, which need a single shadow model to set the threshold under the attack mode, we reuse the same shadow models; the threshold is simply derived from the first model, ensuring complete consistency across methods.

When an explicit attack model is needed (Shadow MIA, ML-Leak and Blind-MI), we train a small binary classifier on the shadow logits. We use SGD with momentum 0.9, weight decay 5e-4, initial lr 0.1 reduced ×0.1 at steps 100, 150 and 200—is reused here to ensure consistency.

For quantile MIA method, our paper follows the same setting as Bertran et al. (2024). We employ a ConvNeXt-Tiny (Liu et al., 2022) backbone pretrained on ImageNet, freeze its feature extractor, and fit a quantile-regression head with Adam for five epochs. Separate quantile models are trained for epochs 10th, 110th and 200th epochs.

## F.2 DETAILS OF DATASETS

We provide the details of the datasets evaluated in this work as follows:

- **CIFAR-100:** CIFAR-100 (Krizhevsky et al., 2009) consists of 60000 $32 \times 32$ color images in 100 classes, with 600 images per class. There are 50000 training images and 10000 test images. The 100 classes in the CIFAR-100 are grouped into 20 superclasses. Each image comes with a "fine" label (the class to which it belongs) and a "coarse" label (the superclass to which it belongs).

- **CIFAR-10:** Similar to CIFAR-100, CIFAR-10 (Krizhevsky et al., 2009) consists of 60000 $32 \times 32$ color images in 10 classes. There are 50000 training images and 10000 test images.

- **ImageNet-100:** ImageNet-100 is a subset of ImageNet-1k (Russakovsky et al., 2015). It contains random 100 classes. The training set contains 1300 $224 \times 224$ color images for each class, and the validation set contains 50 images for each class.

We obtain target and shadow datasets from the original dataset by using Algorithm 1, where the split rate for shadow dataset $r_1 = 0.5$ and sampling rate for sub-shadow dataset $r_2 = 0.8$ by default.

## F.3 DETAILS OF ARCHITECTURES

We summarize the sizes of the evaluated models, e.g., 4-layer CNN (Krizhevsky et al., 2012), VGG-11 (Simonyan & Zisserman, 2015), ResNet-18 (He et al., 2016), ResNet-34, ResNet-50, WideResNet-28-2 (Zagoruyko & Komodakis, 2016), and Swin Transformer-Tiny (Liu et al., 2021), in Table 5.

Table 5: Sizes of different model architectures.

| Model | 4-layer CNN | VGG-11 | ResNet-18 | ResNet-34 | ResNet-50 | WideResNet-28-2 | Swin Transformer-Tiny |
|---|---|---|---|---|---|---|---|
| Size (MB) | 2.13 | 35.43 | 42.87 | 81.50 | 90.73 | 5.72 | 105.56 |

## F.4 DETAILS OF TRAINING ALGORITHMS

We provide the details of the training algorithms evaluated in this work as follows:

- **Standard:** For standard setting, we train the model from scratch for 200 epochs, and the batch size is 128. If the models are CNNs, the adopted optimizer is SGD with an initial learning rate of 0.1. If the model is Transformers, the adopted optimizer is Adam with an initial learning rate of $1 \times 10^{-4}$. The weight decay factor is $5 \times 10^{-4}$. We adopt a step-wise learning rate decay scheduler, where the learning rate decays by a factor of 0.1 at the 100th and 150th epoch.

- **Fine-tuning:** For the fine-tuning setting, the initial learning rate is set to 0.01 and $1 \times 10^{-5}$ for SGD and Adam, respectively. Other configurations are the same as those in standard training.

- **Privacy-enhancing:** For privacy-enhancing setting, we adopt DP-SGD optimizer (Abadi et al., 2016). Specifically, the $\sigma$ of noise added to the gradient is 0.001 and the maximum gradient norm is 10 to maintain utility. Additionally, the batch normalization layers are replaced by group normalization layers. Other configurations are the same as those in standard training.

- **Machine-unlearning:** For machine unlearning setting, we adopt two competitive machine unlearning methods, SalUn (Fan et al., 2023) and SFR-on (Huang et al., 2024). We sweep the hyper-parameters for each method and select the one yielding minimal accuracy gap in terms of forget, retain and test data to the retrain model. For SalUn, the learning rate is 0.01, epoch is 10

and saliency mask ratio is $50\%$. For SFR-on, the learning rate for forget data is 0.5, learning rate for retain data is 0.01 and $\gamma$ is 10.

# G ADDITIONAL EXPERIMENTAL RESULTS

## G.1 PERFORMANCE OF TARGET MODEL IN OUR EXPERIMENTS

To indicate the overfitting degree of different settings, we summarize the performance of different model architectures and the performance of models trained on different datasets and training algorithms in Table 6. Additionally, we provide performance summaries with fine-grained mislabel rates and training epochs in Table 7 and 8, respectively.

Table 6: Performance summary in different architectures, datasets and training algorithms. For different architectures, we train the model on CIFAR100 by a standard training scheme. For different datasets, the model is ResNet18 if not specified. Note that the model trained on ImageNet-100 is WideResNet28 (WRN-28). The mislabel ratio is 0.1 for mislabeled CIFAR-100. For fine-tuning setting, we fine-tune a ResNet50 (RN-50) pretrained on ImageNet1k. For privacy-enhancing setting, we adopt DP-SGD optimizer to train the model.

| | Training Accuracy | Training Loss | Test Accuracy | Test Loss | Accuracy Gap | Loss Gap |
|---|---|---|---|---|---|---|
| *Different Datasets*, *ResNet18*, *Standard* | | | | | | |
| CIFAR100 | 99.98 | 0.0054 | 62.34 | 1.6833 | 37.64 | 1.6779 |
| CIFAR10 | 99.97 | 0.0020 | 89.06 | 0.4733 | 10.91 | 0.4713 |
| ImageNet100 (WRN-28) | 80.90 | 0.7006 | 76.40 | 0.8521 | 4.50 | 0.1515 |
| Superclass CIFAR100 | 99.96 | 0.0029 | 74.58 | 1.1872 | 25.38 | 1.1843 |
| Mislabeled CIFAR100 | 99.96 | 0.0067 | 49.46 | 2.7693 | 50.50 | 2.7626 |
| *CIFAR100*, *Different Architectures*, *Standard* | | | | | | |
| ResNet-18 | 99.98 | 0.0054 | 62.34 | 1.6833 | 37.64 | 1.6779 |
| 4-layer CNN | 64.06 | 1.2525 | 47.82 | 2.1517 | 16.24 | 0.8992 |
| VGG11 | 99.98 | 0.0079 | 61.94 | 1.5941 | 38.04 | 1.5862 |
| ResNet34 | 99.99 | 0.0031 | 60.88 | 1.7420 | 39.11 | 1.7389 |
| ResNet50 | 99.98 | 0.0029 | 61.94 | 1.8573 | 38.04 | 1.8544 |
| Swin-T (pretrained) | 98.85 | 0.0548 | 85.30 | 0.6903 | 13.55 | 0.6355 |
| *CIFAR100*, *ResNet18*, *Different Training Algorithms* | | | | | | |
| Standard | 99.98 | 0.0054 | 62.34 | 1.6833 | 37.64 | 1.6779 |
| Standard (RN-50) | 99.98 | 0.0029 | 61.94 | 1.8573 | 38.04 | 1.8544 |
| Fine-tuning (RN-50) | 99.99 | 0.0006 | 81.46 | 0.2462 | 18.53 | 0.2456 |
| Privacy-enhancing | 95.70 | 0.2050 | 67.74 | 2.0177 | 27.96 | 1.8727 |

Table 7: Performance summary in different mislabel rates. The model is ResNet-18 trained on CIFAR-100 by a standard training scheme.

| Mislabel Rate | Training Accuracy | Training Loss | Test Accuracy | Test Loss | Accuracy Gap | Loss Gap |
|---|---|---|---|---|---|---|
| 0 | 99.98 | 0.0054 | 62.34 | 1.6833 | 37.64 | 1.6779 |
| 0.01 | 99.98 | 0.0051 | 61.88 | 1.7731 | 38.10 | 1.7680 |
| 0.05 | 99.98 | 0.0056 | 54.52 | 2.3324 | 45.46 | 2.3268 |
| 0.1 | 99.96 | 0.0067 | 49.46 | 2.7693 | 50.50 | 2.7626 |
| 0.2 | 99.98 | 0.0076 | 38.08 | 3.5541 | 61.90 | 3.5465 |
| 0.3 | 99.95 | 0.0077 | 28.56 | 4.3316 | 71.39 | 4.3239 |
| 0.4 | 99.96 | 0.0072 | 20.24 | 4.8633 | 79.72 | 4.8561 |
| 0.5 | 99.95 | 0.0077 | 13.96 | 5.3637 | 85.99 | 5.3560 |

Table 8: Performance summary at different trainind epochs. The model is ResNet-18 trained on CIFAR-100 by a standard training scheme.

| Epoch | Training Accuracy | Training Loss | Test Accuracy | Test Loss | Accuracy Gap | Loss Gap |
|-------|-------------------|---------------|---------------|-----------|--------------|----------|
| 200 | 99.98 | 0.0054 | 62.34 | 1.6833 | 37.64 | 1.6779 |
| 190 | 99.98 | 0.0051 | 62.38 | 1.6935 | 37.60 | 1.6884 |
| 180 | 99.97 | 0.0053 | 62.52 | 1.7020 | 37.45 | 1.6967 |
| 170 | 99.96 | 0.0060 | 67.20 | 1.7033 | 37.76 | 1.6973 |
| 160 | 99.96 | 0.0068 | 62.22 | 1.7070 | 37.74 | 1.7002 |
| 150 | 99.96 | 0.0098 | 61.50 | 1.7345 | 38.46 | 1.7246 |
| 140 | 99.92 | 0.0112 | 61.80 | 1.7590 | 38.12 | 1.7479 |
| 130 | 99.84 | 0.0155 | 60.88 | 1.7741 | 38.96 | 1.7586 |
| 120 | 99.62 | 0.0255 | 61.98 | 1.7527 | 37.64 | 1.7272 |
| 110 | 99.02 | 0.0538 | 61.64 | 1.6987 | 37.38 | 1.6449 |
| 100 | 71.39 | 0.9672 | 48.74 | 2.2259 | 22.65 | 1.2587 |
| 90 | 67.45 | 1.1063 | 47.36 | 2.2759 | 20.09 | 1.1696 |
| 80 | 68.92 | 1.0604 | 48.72 | 2.2110 | 20.20 | 1.1505 |
| 70 | 70.27 | 0.9937 | 49.76 | 2.0981 | 20.51 | 1.1044 |
| 60 | 61.20 | 1.3343 | 44.04 | 2.3934 | 17.16 | 1.0591 |
| 50 | 67.82 | 1.0763 | 48.14 | 2.1618 | 19.68 | 1.0855 |
| 40 | 64.04 | 1.2320 | 48.14 | 2.0953 | 15.90 | 0.8633 |
| 30 | 58.26 | 1.4585 | 46.28 | 2.1908 | 11.68 | 0.7323 |
| 20 | 53.98 | 1.6317 | 44.84 | 2.1480 | 9.14 | 0.5163 |
| 10 | 42.63 | 2.1265 | 37.62 | 2.3711 | 5.01 | 0.2506 |

## G.2 PERFORMANCE OF MIA METHODS IN DIFFERENT ARCHITECTURES

In this section, we first evaluate the performance of each MIA method on CIFAR-100 using ResNet-18 as the baseline model for our paper. Subsequently, we present the performance of these MIA methods on CIFAR-100 using a 4-layer CNN, VGG-11, ResNet-34, ResNet-50 (trained from scratch) models.

For ResNet-18, Table 9 shows that Modified Entropy achieved the highest accuracy in audit mode, while quantile MIA performed best in attack mode at epoch 200 and 110; at epochs 110 and 10. However, Metric MIA showed the largest performance gap between audit and attack modes, while RMIA exhibited the smallest difference.

Table 9: The performance of different MIA methods for CIFAR-100 under ResNet-18. We pick three checkpoints in the early, middle and final training phases for MIA evaluation. For MIA methods without a threshold hyper-parameter, they follow the same workflow for both attacking and auditing.

| Epoch | Metric | Mode | Membership Inference Attack Algorithms | | | | | | | | | | | | | |
| | | | Metric MIA | | | | | | | ML-Leaks | | | BlindMI | | | |
| | | | LiRA | RMIA | Entropy | Modified Entropy | Confidence | Quantile | Merlin | MLLeak1 | MLLeak3 | Shadow | One-class | Diff-w | Diff-single | Diff-bi |
|---|---|---|---|---|---|---|---|---|---|---|---|---|---|---|---|---|
| 200th | Acc | Attack | 65.42 | 74.25 | 78.49 | 80.25 | 80.30 | 80.63 | 56.19 | 73.63 | 75.05 | 63.00 | 68.40 | 77.95 | 75.43 | 67.17 |
| | | Audit | 69.36 | 75.82 | 80.65 | 81.97 | 81.90 | 80.63 | 56.36 | | | | | | | |
| | Prec@P | Attack | 68.81 | 70.12 | 71.29 | 72.87 | 72.77 | 73.88 | 53.37 | 65.52 | 74.62 | 57.47 | 61.29 | 69.49 | 67.17 | 62.99 |
| | | Audit | 69.99 | 68.16 | 75.49 | 76.20 | 76.24 | 73.60 | 56.91 | | | | | | | |
| | Rec@P | Attack | 56.40 | 84.50 | 81.60 | 96.38 | 96.84 | 94.46 | 97.93 | 99.78 | 75.92 | 100.00 | 99.92 | 99.66 | 99.48 | 83.24 |
| | | Audit | 67.78 | 96.92 | 90.76 | 92.90 | 92.68 | 95.52 | 52.40 | | | | | | | |
| | F1@P | Attack | 61.99 | 76.64 | 76.10 | 82.99 | 83.10 | 82.91 | 87.52 | 78.30 | 75.27 | 72.99 | 75.97 | 81.88 | 80.19 | 71.72 |
| | | Audit | 68.87 | 80.03 | 82.43 | 83.75 | 83.66 | 83.14 | 54.56 | | | | | | | |
| | Prec@N | Attack | 63.06 | 80.50 | 93.05 | 94.66 | 95.28 | 92.32 | 14.44 | 99.60 | 75.49 | 100.00 | 99.78 | 99.40 | 99.00 | 75.30 |
| | | Audit | 68.77 | 94.67 | 88.42 | 90.91 | 90.67 | 93.62 | 55.89 | | | | | | | |
| | Rec@N | Attack | 74.44 | 64.00 | 61.58 | 64.12 | 63.76 | 66.60 | 69.09 | 44.84 | 74.18 | 26.00 | 36.88 | 56.24 | 51.38 | 51.10 |
| | | Audit | 70.94 | 54.71 | 70.54 | 71.04 | 71.12 | 65.74 | 60.32 | | | | | | | |
| | F1@N | Attack | 68.28 | 71.31 | 74.11 | 76.45 | 76.40 | 77.38 | 24.90 | 61.84 | 74.83 | 41.27 | 53.86 | 71.84 | 67.65 | 69.88 |
| | | Audit | 69.84 | 69.35 | 78.47 | 79.76 | 79.71 | 77.24 | 58.02 | | | | | | | |
| | AUC | Audit | 0.745 | 0.839 | 0.818 | 0.828 | 0.826 | 0.829 | 0.510 | – | – | – | – | – | – | – |
| | TPR@0.1%FPR | Audit | 2.0 | 8.5 | 8.5 | 8.9 | 8.7 | 0.36 | 1.4 | – | – | – | – | – | – | – |
| 110th | Acc | Attack | 62.09 | 73.31 | 71.95 | 73.51 | 73.83 | 74.61 | 55.32 | 52.37 | 51.31 | 61.40 | 72.64 | 74.99 | 71.64 | 66.09 |
| | | Audit | 66.72 | 74.09 | 75.10 | 77.52 | 77.57 | 75.12 | 56.23 | | | | | | | |
| | Prec@P | Attack | 65.37 | 69.49 | 69.00 | 70.88 | 70.70 | 67.32 | 52.89 | 51.21 | 69.29 | 56.45 | 67.77 | 68.20 | 65.21 | 64.12 |
| | | Audit | 67.12 | 66.68 | 70.98 | 72.19 | 72.11 | 68.84 | 57.30 | | | | | | | |
| | Rec@P | Attack | 51.42 | 83.12 | 79.72 | 79.80 | 81.38 | 95.64 | 97.26 | 100.00 | 6.64 | 100.00 | 86.34 | 93.64 | 92.76 | 73.06 |
| | | Audit | 65.40 | 96.32 | 84.92 | 89.52 | 89.92 | 91.78 | 48.92 | | | | | | | |
| | F1@P | Attack | 57.56 | 75.69 | 73.97 | 75.08 | 75.67 | 79.02 | 83.00 | 67.73 | 12.00 | 72.99 | 75.94 | 78.92 | 76.59 | 68.30 |
| | | Audit | 66.27 | 78.80 | 77.33 | 79.93 | 80.04 | 78.67 | 52.78 | | | | | | | |
| | Prec@N | Attack | 59.96 | 79.00 | 75.99 | 76.89 | 78.07 | 92.47 | 13.38 | 47.72 | 50.69 | 100.00 | 81.18 | 89.86 | 84.47 | 68.70 |
| | | Audit | 66.29 | 93.37 | 81.23 | 86.21 | 86.61 | 87.67 | 55.44 | | | | | | | |
| | Rec@N | Attack | 72.76 | 63.50 | 64.18 | 67.22 | 66.28 | 53.58 | 68.52 | 67.74 | 95.98 | 26.00 | 58.92 | 56.34 | 50.52 | 59.12 |
| | | Audit | 68.04 | 51.86 | 65.28 | 65.52 | 65.22 | 58.46 | 63.54 | | | | | | | |
| | F1@N | Attack | 65.75 | 70.41 | 69.59 | 71.73 | 71.69 | 67.85 | 23.05 | 55.95 | 66.34 | 41.27 | 68.30 | 69.26 | 64.05 | 63.55 |
| | | Audit | 67.15 | 66.68 | 72.39 | 74.45 | 74.41 | 70.15 | 59.21 | | | | | | | |
| | AUC | Audit | 0.711 | 0.829 | 0.772 | 0.792 | 0.791 | 0.788 | 0.504 | – | – | – | – | – | – | – |
| | TPR@0.1%FPR | Audit | 1.4 | 5.8 | 6.0 | 6.4 | 6.3 | 0.42 | 1.5 | – | – | – | – | – | – | – |
| 10th | Acc | Attack | 52.37 | 53.88 | 50.01 | 50.01 | 50.08 | 52.81 | 51.03 | 50.00 | 50.15 | 50.09 | 50.45 | 50.58 | 50.85 | 51.05 |
| | | Audit | 52.46 | 53.93 | 57.34 | 59.01 | 58.85 | 53.65 | 55.98 | | | | | | | |
| | Prec@P | Attack | 54.36 | 53.31 | 50.29 | 50.26 | 52.02 | 51.80 | 50.63 | 50.00 | 51.24 | 50.05 | 52.31 | 51.10 | 50.62 | 51.24 |
| | | Audit | 52.88 | 53.34 | 53.18 | 59.80 | 59.85 | 53.99 | 56.71 | | | | | | | |
| | Rec@P | Attack | 29.56 | 62.56 | 1.72 | 1.94 | 2.06 | 81.00 | 83.92 | 100.00 | 6.20 | 97.36 | 10.20 | 26.92 | 69.88 | 43.40 |
| | | Audit | 45.14 | 62.80 | 58.01 | 55.00 | 53.78 | 59.36 | 50.54 | | | | | | | |
| | F1@P | Attack | 38.30 | 57.56 | 3.33 | 3.74 | 3.96 | 63.19 | 53.04 | 66.67 | 11.06 | 66.11 | 17.07 | 35.26 | 58.71 | 47.00 |
| | | Audit | 52.15 | 57.68 | 55.49 | 57.30 | 56.65 | 51.57 | 53.45 | | | | | | | |
| | Prec@N | Attack | 51.63 | 54.70 | 50.01 | 50.01 | 50.04 | 56.44 | 18.16 | 0.00 | 50.08 | 51.65 | 50.25 | 50.39 | 51.37 | 50.91 |
| | | Audit | 59.78 | 54.78 | 56.78 | 58.34 | 58.04 | 53.36 | 55.39 | | | | | | | |
| | Rec@N | Attack | 75.18 | 45.20 | 98.30 | 98.08 | 98.10 | 24.62 | 63.15 | 66.67 | 94.10 | 2.82 | 90.70 | 74.24 | 31.82 | 58.70 |
| | | Audit | 59.78 | 45.06 | 61.50 | 63.02 | 63.92 | 57.94 | 61.42 | | | | | | | |
| | F1@N | Attack | 61.22 | 49.50 | 66.29 | 66.24 | 66.27 | 34.28 | 27.05 | 0.00 | 65.37 | 5.35 | 64.67 | 60.04 | 39.30 | 54.53 |
| | | Audit | 55.70 | 49.45 | 59.04 | 60.59 | 60.84 | 55.56 | 58.25 | | | | | | | |
| | AUC | Audit | 0.542 | 0.549 | 0.514 | 0.551 | 0.550 | 0.549 | 0.511 | – | – | – | – | – | – | – |
| | TPR@0.1%FPR | Audit | 0.02 | 0.10 | 2.0 | 2.4 | 2.4 | 0.4 | 1.4 | – | – | – | – | – | – | – |

Table 10: The performance of different MIA methods for CIFAR-100 under 4-layer CNN.

| Epoch | Metric | Mode | LiRA | RMIA | Entropy | Modified Entropy | Confidence | Quantile | Merlin | MLLeak1 | MLLeak3 | Shadow | One-class | Diff-w | Diff-single | Diff-bi |
|---|---|---|---|---|---|---|---|---|---|---|---|---|---|---|---|---|
| | | | | | | Metric MIA | | | | ML-Leaks | | | BlindMI | | | |
| 200th | Acc | Attack | 55.93 | 63.31 | 53.18 | 60.94 | 60.90 | 61.03 | 52.61 | 50.00 | 50.42 | 51.16 | 50.12 | 55.61 | 53.63 | 53.57 |
| | | Audit | 56.02 | 63.44 | 59.59 | 65.22 | 65.05 | 61.19 | 56.87 | | | | | | | |
| | Prec@P | Attack | 60.26 | 60.75 | 53.10 | 57.82 | 57.75 | 57.25 | 51.71 | 50.00 | 55.80 | 50.60 | 62.00 | 53.50 | 52.99 | 54.17 |
| | | Audit | 60.10 | 61.55 | 60.25 | 63.45 | 63.13 | 57.24 | 56.21 | | | | | | | |
| | Rec@P | Attack | 34.84 | 75.22 | 54.46 | 80.92 | 81.24 | 87.14 | 78.84 | 100.00 | 4.04 | 98.18 | 0.62 | 85.68 | 64.38 | 46.36 |
| | | Audit | 35.82 | 71.60 | 56.38 | 71.80 | 72.34 | 88.52 | 62.18 | | | | | | | |
| | F1@P | Attack | 44.15 | 67.21 | 53.77 | 67.44 | 67.51 | 69.10 | 62.46 | 66.67 | 7.53 | 66.78 | 1.23 | 65.87 | 58.13 | 49.96 |
| | | Audit | 44.89 | 66.20 | 58.25 | 67.37 | 67.42 | 69.52 | 59.04 | | | | | | | |
| | Prec@N | Attack | 54.17 | 67.47 | 53.26 | 68.22 | 68.37 | 73.08 | 55.49 | 0.00 | 50.22 | 69.46 | 50.06 | 64.07 | 54.62 | 53.12 |
| | | Audit | 54.29 | 66.06 | 59.01 | 67.53 | 67.62 | 74.68 | 57.69 | | | | | | | |
| | Rec@N | Attack | 77.02 | 51.40 | 51.90 | 40.96 | 40.56 | 34.92 | 26.38 | 0.00 | 96.80 | 4.14 | 99.62 | 25.54 | 42.88 | 60.78 |
| | | Audit | 76.22 | 55.28 | 62.80 | 58.64 | 57.76 | 33.86 | 51.56 | | | | | | | |
| | F1@N | Attack | 63.61 | 58.35 | 52.57 | 51.17 | 50.92 | 47.26 | 35.76 | 0.00 | 66.13 | 7.81 | 66.64 | 36.52 | 48.04 | 56.69 |
| | | Audit | 63.41 | 60.19 | 60.85 | 62.77 | 62.30 | 46.59 | 54.45 | | | | | | | |
| | AUC | Audit | 0.566 | 0.688 | 0.558 | 0.652 | 0.650 | 0.631 | 0.509 | – | – | – | – | – | – | – |
| | TPR@0.1%FPR | Audit | 0.16 | 0.84 | 0.02 | 0.03 | 0.03 | 0.00 | 0.03 | – | – | – | – | – | – | – |
| 110th | Acc | Attack | 53.11 | 57.73 | 52.10 | 54.92 | 55.20 | 56.05 | 51.89 | 50.00 | 52.16 | 50.31 | 50.06 | 52.08 | 51.99 | 51.22 |
| | | Audit | 56.68 | 57.80 | 58.32 | 61.20 | 61.21 | 56.77 | 56.58 | | | | | | | |
| | Prec@P | Attack | 51.73 | 57.47 | 51.78 | 53.75 | 53.90 | 54.16 | 51.20 | 50.00 | 54.98 | 50.16 | 53.41 | 51.24 | 51.51 | 51.61 |
| | | Audit | 56.80 | 57.63 | 59.28 | 61.12 | 61.29 | 54.20 | 56.31 | | | | | | | |
| | Rec@P | Attack | 93.10 | 59.44 | 61.18 | 70.44 | 71.94 | 78.72 | 80.66 | 100.00 | 23.84 | 97.52 | 0.94 | 85.96 | 68.00 | 39.14 |
| | | Audit | 55.82 | 58.92 | 53.14 | 61.58 | 60.86 | 87.28 | 58.68 | | | | | | | |
| | F1@P | Attack | 66.50 | 58.44 | 56.09 | 60.98 | 61.62 | 64.17 | 62.64 | 66.67 | 33.26 | 66.25 | 1.85 | 64.21 | 58.62 | 44.52 |
| | | Audit | 56.30 | 58.27 | 56.04 | 61.35 | 61.07 | 66.88 | 57.47 | | | | | | | |
| | Prec@N | Attack | 65.53 | 58.00 | 52.57 | 57.13 | 57.82 | 61.07 | 54.45 | 0.00 | 51.38 | 55.56 | 50.03 | 56.45 | 52.93 | 50.98 |
| | | Audit | 56.57 | 57.98 | 57.54 | 61.29 | 61.13 | 67.37 | 56.87 | | | | | | | |
| | Rec@N | Attack | 13.12 | 56.02 | 43.02 | 39.40 | 38.46 | 33.38 | 23.12 | 0.00 | 80.48 | 3.10 | 99.18 | 18.20 | 35.98 | 63.30 |
| | | Audit | 57.54 | 56.68 | 63.50 | 60.82 | 61.56 | 26.26 | 54.48 | | | | | | | |
| | F1@N | Attack | 21.86 | 56.99 | 47.32 | 46.64 | 46.19 | 43.17 | 32.46 | 0.00 | 62.72 | 5.87 | 66.51 | 27.53 | 42.84 | 56.48 |
| | | Audit | 57.05 | 57.32 | 60.37 | 61.05 | 61.35 | 37.79 | 55.65 | | | | | | | |
| | AUC | Audit | 0.583 | 0.604 | 0.540 | 0.591 | 0.591 | 0.571 | 0.497 | – | – | – | – | – | – | – |
| | TPR@0.1%FPR | Audit | 0.00 | 2.2 | 0.03 | 0.03 | 0.03 | 0.0 | 0.0 | – | – | – | – | – | – | – |
| 10th | Acc | Attack | 50.55 | 51.28 | 49.35 | 52.21 | 52.46 | 51.40 | 50.41 | 50.00 | 50.55 | 50.12 | 50.11 | 49.81 | 50.46 | 49.98 |
| | | Audit | 51.86 | 52.01 | 56.79 | 57.74 | 57.75 | 51.80 | 56.93 | | | | | | | |
| | Prec@P | Attack | 50.30 | 52.13 | 49.38 | 51.83 | 52.00 | 50.91 | 50.35 | 50.00 | 50.96 | 50.06 | 53.22 | 49.73 | 50.31 | 49.98 |
| | | Audit | 51.35 | 51.55 | 57.64 | 58.76 | 58.20 | 51.15 | 57.62 | | | | | | | |
| | Rec@P | Attack | 92.52 | 31.34 | 52.04 | 62.66 | 62.92 | 78.54 | 59.78 | 100.00 | 29.06 | 97.32 | 1.82 | 34.70 | 75.06 | 44.08 |
| | | Audit | 70.96 | 67.02 | 51.24 | 51.94 | 54.98 | 80.14 | 52.42 | | | | | | | |
| | F1@P | Attack | 65.17 | 39.15 | 50.68 | 56.73 | 56.96 | 61.77 | 54.66 | 66.67 | 37.01 | 66.12 | 3.52 | 40.88 | 60.24 | 46.84 |
| | | Audit | 59.58 | 58.27 | 54.25 | 55.14 | 56.55 | 62.44 | 54.90 | | | | | | | |
| | Prec@N | Attack | 53.42 | 50.92 | 49.31 | 52.79 | 53.11 | 53.06 | 50.50 | 0.00 | 50.38 | 52.16 | 50.06 | 49.85 | 50.91 | 49.98 |
| | | Audit | 53.01 | 52.87 | 56.11 | 56.94 | 57.34 | 54.16 | 56.36 | | | | | | | |
| | Rec@N | Attack | 8.58 | 71.22 | 46.66 | 41.76 | 42.00 | 24.26 | 41.04 | 0.00 | 72.04 | 2.90 | 98.40 | 64.92 | 25.86 | 55.88 |
| | | Audit | 32.76 | 37.00 | 62.34 | 63.54 | 60.52 | 23.46 | 61.44 | | | | | | | |
| | F1@N | Attack | 14.79 | 59.38 | 47.95 | 46.63 | 46.91 | 33.30 | 45.28 | 0.00 | 59.30 | 5.49 | 66.36 | 56.40 | 34.30 | 52.77 |
| | | Audit | 40.49 | 43.53 | 59.06 | 60.06 | 58.89 | 32.74 | 58.79 | | | | | | | |
| | AUC | Audit | 0.524 | 0.526 | 0.510 | 0.526 | 0.527 | 0.523 | 0.505 | – | – | – | – | – | – | – |
| | TPR@0.1%FPR | Audit | 0.0 | 1.5 | 0.02 | 0.03 | 0.03 | 0.00 | 0.00 | – | – | – | – | – | – | – |

For 4-layer CNN, Table 10 indicates that Modified Entropy achieved nearly the highest accuracy in audit mode at epoch 200, 110 and 10, while RMIA MIA performed best in attack mode at epoch 200, 110 and 10; Moreover, RMIA also consistently shows the best performance in AUC and TPR@0.1%FPR. Additional, Metric MIA showed the largest performance gap between audit and attack modes, while RMIA exhibited the smallest difference.

For CIFAR100 under VGG, Table 11 indicate: Modified Entropy attained the highest accuracy under audit mode, whereas quantile MIA achieved the best accuracy in attack mode at 200,110, and 10 epochs. Regarding AUC and TPR at 0.1% FPR, RMIA outperformed other methods at epochs 200 and 110, while entropy MIA performed best at epoch 10. Merlin showed the largest performance discrepancy between audit and attack modes.

Table 11: The performance of different MIA methods for CIFAR-100 under VGG-11.

| Epoch | Metric | Mode | | | Metric MIA | | | | | ML-Leaks | | | BlindMI | | | |
|---|---|---|---|---|---|---|---|---|---|---|---|---|---|---|---|---|
| | | | LiRA | RMIA | Entropy | Modified Entropy | Confidence | Quantile | Merlin | MLLeak1 | MLLeak3 | Shadow | One-class | Diff-w | Diff-single | Diff-bi |
| 200th | Acc | Attack | 66.07 | 75.20 | 78.41 | 80.61 | 80.14 | 80.68 | 57.51 | 73.89 | 62.32 | 63.00 | 71.41 | 78.91 | 76.25 | 66.79 |
| | | Audit | 68.97 | 76.65 | 80.30 | 82.03 | 81.97 | 80.73 | 52.56 | | | | | | | |
| | Prec@P | Attack | 69.52 | 73.63 | 72.12 | 73.73 | 73.69 | 73.19 | 54.24 | 65.73 | 57.03 | 57.47 | 63.64 | 70.55 | 67.96 | 63.43 |
| | | Audit | 70.92 | 68.32 | 74.57 | 75.88 | 75.84 | 73.26 | 65.57 | | | | | | | |
| | Rec@P | Attack | 57.24 | 78.52 | 92.60 | 95.10 | 93.74 | 96.84 | 96.02 | 99.84 | 100.0 | 100.0 | 99.82 | 99.26 | 99.32 | 79.32 |
| | | Audit | 64.30 | 99.38 | 91.96 | 93.90 | 93.82 | 96.78 | 10.78 | | | | | | | |
| | F1@P | Attack | 62.78 | 76.00 | 81.09 | 83.06 | 82.52 | 83.37 | 69.32 | 79.27 | 72.63 | 72.99 | 77.73 | 82.48 | 80.70 | 70.49 |
| | | Audit | 67.45 | 80.97 | 82.36 | 83.94 | 83.88 | 83.40 | 18.52 | | | | | | | |
| | Prec@N | Attack | 63.66 | 76.99 | 89.66 | 93.10 | 91.41 | 95.33 | 82.68 | 99.67 | 100.0 | 100.0 | 99.58 | 98.75 | 98.74 | 72.40 |
| | | Audit | 67.35 | 98.86 | 89.51 | 92.00 | 91.90 | 95.26 | 51.39 | | | | | | | |
| | Rec@N | Attack | 74.90 | 71.88 | 64.20 | 66.12 | 66.54 | 64.52 | 19.00 | 47.94 | 24.64 | 26.00 | 42.98 | 58.56 | 53.18 | 54.26 |
| | | Audit | 73.64 | 53.92 | 68.64 | 70.16 | 70.12 | 64.68 | 94.34 | | | | | | | |
| | F1@N | Attack | 68.82 | 74.35 | 74.83 | 77.32 | 77.01 | 76.96 | 30.9 | 64.74 | 39.54 | 41.27 | 60.04 | 73.52 | 69.13 | 62.03 |
| | | Audit | 70.35 | 69.78 | 77.70 | 79.61 | 79.55 | 77.05 | 66.54 | | | | | | | |
| | AUC | Audit | 0.734 | 0.856 | 0.810 | 0.822 | 0.821 | 0.820 | 0.587 | – | – | – | – | – | – | – |
| | TPR@0.1%FPR | Audit | 0.881 | 10.96 | 7.244 | 7.779 | 7.593 | 0.150 | 0.000 | – | – | – | – | – | – | – |
| 110th | Acc | Attack | 65.18 | 74.43 | 74.03 | 76.62 | 76.64 | 76.85 | 56.85 | 62.31 | 67.2 | 61.4 | 72.64 | 76.18 | 73.47 | 66.04 |
| | | Audit | 67.57 | 75.45 | 76.04 | 78.73 | 78.66 | 76.82 | 53.01 | | | | | | | |
| | Prec@P | Attack | 68.18 | 72.35 | 68.52 | 70.48 | 70.45 | 70.02 | 53.9 | 57.05 | 60.51 | 56.45 | 65.55 | 69.3 | 66.62 | 63.94 |
| | | Audit | 70.34 | 67.45 | 71.78 | 73.47 | 73.34 | 69.77 | 61.82 | | | | | | | |
| | Rec@P | Attack | 56.94 | 79.08 | 88.92 | 91.62 | 91.78 | 93.9 | 94.56 | 99.58 | 99.04 | 99.84 | 95.42 | 94.02 | 94.08 | 73.58 |
| | | Audit | 60.76 | 98.36 | 85.82 | 89.94 | 90.06 | 94.66 | 15.47 | | | | | | | |
| | F1@P | Attack | 62.05 | 75.57 | 77.40 | 79.67 | 79.71 | 80.22 | 68.67 | 72.54 | 75.12 | 72.12 | 77.72 | 79.79 | 78.00 | 68.42 |
| | | Audit | 65.20 | 80.03 | 78.17 | 80.87 | 80.84 | 80.33 | 25.09 | | | | | | | |
| | Prec@N | Attack | 63.03 | 76.93 | 84.22 | 88.03 | 88.21 | 90.74 | 77.87 | 98.35 | 97.36 | 99.31 | 91.59 | 90.7 | 89.93 | 68.89 |
| | | Audit | 65.46 | 96.97 | 82.37 | 87.03 | 87.12 | 91.70 | 51.72 | | | | | | | |
| | Rec@N | Attack | 73.42 | 69.78 | 59.14 | 61.62 | 61.50 | 59.80 | 19.14 | 25.04 | 35.36 | 22.96 | 49.86 | 58.34 | 52.86 | 58.5 |
| | | Audit | 74.38 | 52.54 | 66.26 | 67.52 | 67.26 | 58.98 | 90.28 | | | | | | | |
| | F1@N | Attack | 67.83 | 73.18 | 69.49 | 72.4 | 72.47 | 72.09 | 30.73 | 39.92 | 51.88 | 37.3 | 64.57 | 71.01 | 66.58 | 63.27 |
| | | Audit | 69.64 | 68.15 | 73.44 | 76.04 | 75.91 | 71.79 | 60.77 | | | | | | | |
| | AUC | Audit | 0.721 | 0.845 | 0.777 | 0.789 | 0.796 | 0.789 | 0.585 | – | – | – | – | – | – | – |
| | TPR@0.1%FPR | Audit | 0.960 | 7.800 | 7.228 | 7.525 | 7.413 | 0.100 | 0.000 | – | – | – | – | – | – | – |
| 10th | Acc | Attack | 51.64 | 50.73 | 50.35 | 51.31 | 51.09 | 50.8 | 50.47 | 50.00 | 50.20 | 50.09 | 50.44 | 49.68 | 50.51 | 51.33 |
| | | Audit | 51.70 | 51.16 | 56.99 | 57.34 | 57.37 | 51.16 | 54.99 | | | | | | | |
| | Prec@P | Attack | 51.40 | 50.83 | 50.23 | 50.98 | 50.84 | 50.40 | 50.39 | 50.00 | 53.52 | 50.05 | 50.76 | 46.06 | 50.36 | 51.59 |
| | | Audit | 50.90 | 50.89 | 57.96 | 58.18 | 57.96 | 50.92 | 55.97 | | | | | | | |
| | Rec@P | Attack | 62.34 | 44.74 | 76.42 | 68.34 | 65.86 | 74.94 | 61.4 | 100.0 | 3.04 | 97.36 | 29.26 | 3.74 | 70.46 | 43.24 |
| | | Audit | 56.34 | 66.18 | 50.88 | 52.18 | 53.68 | 64.04 | 46.78 | | | | | | | |
| | F1@P | Attack | 56.34 | 47.59 | 60.62 | 58.4 | 57.38 | 60.37 | 55.35 | 66.67 | 5.75 | 66.11 | 37.12 | 6.92 | 58.74 | 47.05 |
| | | Audit | 57.47 | 57.54 | 54.19 | 55.02 | 55.74 | 56.73 | 50.96 | | | | | | | |
| | Prec@N | Attack | 52.16 | 50.65 | 50.73 | 51.99 | 51.55 | 51.55 | 50.60 | 0 | 50.10 | 51.65 | 50.31 | 49.83 | 50.85 | 51.14 |
| | | Audit | 51.66 | 51.66 | 56.23 | 56.65 | 56.86 | 51.56 | 54.29 | | | | | | | |
| | Rec@N | Attack | 41.06 | 56.72 | 24.28 | 34.28 | 36.32 | 26.66 | 39.54 | 0 | 97.36 | 97.36 | 71.62 | 95.62 | 30.56 | 59.42 |
| | | Audit | 36.36 | 36.14 | 63.10 | 62.50 | 61.06 | 38.28 | 63.20 | | | | | | | |
| | F1@N | Attack | 45.95 | 53.51 | 32.84 | 41.32 | 42.61 | 35.14 | 44.39 | 0 | 66.16 | 66.16 | 59.1 | 65.52 | 38.18 | 54.97 |
| | | Audit | 42.68 | 42.53 | 59.47 | 59.43 | 59.44 | 43.94 | 58.40 | | | | | | | |
| | AUC | Audit | 0.515 | 0.513 | 0.510 | 0.518 | 0.518 | 0.512 | 0.509 | – | – | – | – | – | – | – |
| | TPR@0.1%FPR | Audit | 0.222 | 1.280 | 2.507 | 2.189 | 2.234 | 0.320 | 0.233 | – | – | – | – | – | – | – |

Table 12: The performance of different MIA methods for CIFAR-100 under ResNet-34.

| Epoch | Metric | Mode | LiRA | RMIA | Entropy | Modified Entropy | Confidence | Quantile | Merlin | MLLeak1 | MLLeak3 | Shadow | One-class | Diff-w | Diff-single | Diff-bi |
|---|---|---|---|---|---|---|---|---|---|---|---|---|---|---|---|---|
| | | | | | | | | | | Membership Inference Attack Algorithms | | | | | | |
| | | | | | Metric MIA | | | | | ML-Leaks | | | BlindMI | | | |
| 200th | Acc | Attack | 65.05 | 74.27 | 81.20 | 82.76 | 82.79 | 83.09 | 56.57 | 75.69 | 63.86 | 63.39 | 72.46 | 79.84 | 77.22 | 69.54 |
| | | Audit | 69.48 | 75.77 | 82.82 | 83.93 | 83.78 | 83.16 | 56.66 | | | | | | | |
| | Prec@P | Attack | 69.10 | 72.12 | 74.19 | 75.62 | 75.47 | 75.82 | 53.62 | 67.30 | 58.05 | 57.73 | 64.49 | 71.31 | 68.74 | 64.63 |
| | | Audit | 71.80 | 67.38 | 77.30 | 78.43 | 78.18 | 76.40 | 56.80 | | | | | | | |
| | Rec@P | Attack | 54.44 | 79.12 | 95.70 | 96.70 | 97.16 | 97.16 | 97.28 | 99.94 | 100.00 | 100.00 | 99.96 | 99.84 | 99.84 | 86.30 |
| | | Audit | 64.16 | 99.90 | 92.92 | 93.60 | 93.72 | 95.96 | 55.60 | | | | | | | |
| | F1@P | Attack | 60.90 | 75.46 | 83.58 | 84.87 | 84.95 | 85.18 | 69.14 | 80.43 | 73.45 | 73.20 | 78.40 | 83.20 | 81.42 | 73.91 |
| | | Audit | 67.77 | 80.48 | 84.40 | 85.35 | 85.25 | 85.07 | 56.20 | | | | | | | |
| | Prec@N | Attack | 62.42 | 76.88 | 93.94 | 95.42 | 96.01 | 96.05 | 85.36 | 99.88 | 100.00 | 100.00 | 99.91 | 99.73 | 99.71 | 79.39 |
| | | Audit | 67.61 | 99.81 | 91.13 | 92.07 | 92.16 | 94.57 | 56.52 | | | | | | | |
| | Rec@N | Attack | 75.66 | 69.42 | 51.44 | 27.72 | 44.96 | 69.02 | 15.86 | 59.84 | 54.60 | 26.78 | 52.78 | 66.70 | 68.82 | 68.42 |
| | | Audit | 74.80 | 51.64 | 72.72 | 74.26 | 73.84 | 70.36 | 57.72 | | | | | | | |
| | F1@N | Attack | 68.40 | 72.96 | 67.91 | 43.41 | 62.01 | 80.32 | 26.75 | 74.80 | 70.56 | 42.25 | 63.41 | 78.01 | 79.97 | 79.90 |
| | | Audit | 71.02 | 68.06 | 80.89 | 82.21 | 81.99 | 80.69 | 57.11 | | | | | | | |
| | AUC | Audit | 0.741 | 0.844 | 0.836 | 0.846 | 0.843 | 0.856 | 0.508 | – | – | – | – | – | – | – |
| | TPR@0.1%FPR | Audit | 1.82 | 11.1 | 6.70 | 7.10 | 7.00 | 0.66 | 1.87 | – | – | – | – | – | – | – |
| 110th | Acc | Attack | 62.22 | 74.43 | 75.31 | 77.57 | 77.83 | 77.68 | 54.96 | 50.00 | 60.40 | 62.45 | 73.41 | 77.15 | 73.67 | 68.46 |
| | | Audit | 66.71 | 75.42 | 77.69 | 79.71 | 79.75 | 77.49 | 56.50 | | | | | | | |
| | Prec@P | Attack | 64.02 | 70.36 | 69.64 | 71.19 | 71.39 | 71.10 | 52.69 | 50.00 | 55.82 | 57.12 | 66.05 | 70.00 | 66.61 | 65.65 |
| | | Audit | 67.51 | 67.65 | 73.57 | 74.90 | 74.71 | 70.87 | 57.72 | | | | | | | |
| | Rec@P | Attack | 55.80 | 84.42 | 89.76 | 92.62 | 92.88 | 93.28 | 97.12 | 100.00 | 99.78 | 99.94 | 96.34 | 95.02 | 94.92 | 77.44 |
| | | Audit | 64.42 | 97.44 | 86.44 | 89.38 | 89.94 | 93.34 | 48.58 | | | | | | | |
| | F1@P | Attack | 59.63 | 76.75 | 78.43 | 80.50 | 80.73 | 80.69 | 68.32 | 66.67 | 71.59 | 72.69 | 78.37 | 80.61 | 78.28 | 71.06 |
| | | Audit | 65.93 | 79.86 | 79.49 | 81.50 | 81.62 | 80.57 | 52.76 | | | | | | | |
| | Prec@N | Attack | 60.83 | 80.53 | 85.60 | 89.44 | 89.81 | 90.23 | 81.63 | 0.00 | 98.96 | 99.76 | 93.24 | 92.25 | 91.17 | 72.50 |
| | | Audit | 65.98 | 95.43 | 83.56 | 86.83 | 87.36 | 90.25 | 55.61 | | | | | | | |
| | Rec@N | Attack | 68.64 | 64.44 | 60.86 | 62.52 | 62.78 | 62.00 | 12.80 | 0.00 | 21.02 | 24.96 | 50.48 | 59.28 | 52.42 | 59.48 |
| | | Audit | 69.00 | 53.40 | 70.04 | 69.56 | 61.64 | 64.42 | | | | | | | | |
| | F1@N | Attack | 64.50 | 71.59 | 71.14 | 73.60 | 73.90 | 73.55 | 22.13 | 0.00 | 34.68 | 39.93 | 65.50 | 72.18 | 66.57 | 65.35 |
| | | Audit | 67.46 | 68.48 | 75.55 | 77.54 | 77.45 | 73.25 | 59.69 | | | | | | | |
| | AUC | Audit | 0.712 | 0.842 | 0.795 | 0.812 | 0.811 | 0.810 | 0.507 | – | – | – | – | – | – | – |
| | TPR@0.1%FPR | Audit | 0.62 | 7.90 | 6.80 | 7.10 | 7.10 | 0.74 | 2.38 | – | – | – | – | – | – | – |
| 10th | Acc | Attack | 50.04 | 51.81 | 49.82 | 51.57 | 51.43 | 52.00 | 50.71 | 50.00 | 49.90 | 50.22 | 50.00 | 50.17 | 50.52 | 50.26 |
| | | Audit | 50.29 | 52.17 | 56.96 | 58.47 | 58.25 | 53.53 | 56.48 | | | | | | | |
| | Prec@P | Attack | 51.92 | 53.15 | 49.83 | 51.51 | 51.39 | 51.51 | 50.49 | 50.00 | 49.36 | 50.11 | 0.00 | 50.09 | 50.35 | 50.29 |
| | | Audit | 56.17 | 51.57 | 57.86 | 58.63 | 58.60 | 53.71 | 57.24 | | | | | | | |
| | Rec@P | Attack | 1.08 | 30.52 | 51.74 | 53.58 | 52.96 | 68.32 | 72.82 | 100.00 | 7.66 | 97.16 | 0.00 | 92.82 | 75.30 | 44.42 |
| | | Audit | 2.64 | 71.40 | 51.22 | 57.56 | 56.22 | 51.12 | 51.26 | | | | | | | |
| | F1@P | Attack | 2.12 | 38.78 | 50.77 | 52.52 | 52.16 | 58.73 | 59.63 | 66.67 | 13.26 | 66.12 | 0.00 | 65.07 | 60.35 | 47.18 |
| | | Audit | 5.04 | 59.88 | 54.34 | 58.09 | 57.38 | 52.38 | 54.08 | | | | | | | |
| | Prec@N | Attack | 50.02 | 51.27 | 49.81 | 51.64 | 51.48 | 52.97 | 51.27 | 0.00 | 49.95 | 53.39 | 50.00 | 51.16 | 51.03 | 50.23 |
| | | Audit | 50.15 | 53.53 | 56.24 | 58.32 | 57.93 | 53.37 | 55.87 | | | | | | | |
| | Rec@N | Attack | 99.00 | 73.10 | 47.90 | 49.56 | 49.90 | 35.68 | 28.60 | 0.00 | 92.14 | 3.28 | 100.00 | 7.52 | 25.74 | 56.10 |
| | | Audit | 97.94 | 32.94 | 62.70 | 59.38 | 60.28 | 55.94 | 61.70 | | | | | | | |
| | F1@N | Attack | 66.46 | 60.27 | 48.84 | 50.58 | 50.68 | 42.64 | 36.72 | 0.00 | 64.78 | 6.18 | 66.67 | 13.11 | 34.22 | 53.00 |
| | | Audit | 71.02 | 68.06 | 80.89 | 82.21 | 81.99 | 80.69 | 57.11 | | | | | | | |
| | AUC | Audit | 0.492 | 0.528 | 0.505 | 0.531 | 0.531 | 0.531 | 0.514 | – | – | – | – | – | – | – |
| | TPR@0.1%FPR | Audit | 0.16 | 1.40 | 2.10 | 2.10 | 2.10 | 0.12 | 1.62 | – | – | – | – | – | – | – |

For CIFAR-100 under ResNet-34, Table 12 indicates: Modified Entropy attained the highest accuracy under audit mode, whereas quantile MIA achieved the best accuracy in attack mode at 200,110, and 10 epochs.

For CIFAR-100 under ResNet-50(trained from scratch), Table 13 indicates: Modified Entropy attained the highest accuracy under audit mode, whereas quantile MIA achieved the best accuracy in attack mode at 200, 110, 10 epochs. Regarding AUC and TPR at 0.1% FPR, Modified Entropy also outperformed other methods at epochs 200 and 110 and 10 epochs.

For CIFAR-100 under Swin Transformer Tiny with pretrain, Table 14 indicates: Modified Entropy attained the highest accuracy under audit mode

Table 13: The performance of different MIA methods for CIFAR-100 under ResNet-50 **trained from scratch**.

| Epoch | Metric | Mode | Metric MIA | | | | | | | ML-Leaks | | | BlindMI | | | |
|---|---|---|---|---|---|---|---|---|---|---|---|---|---|---|---|---|
| | | | LiRA | RMIA | Entropy | Modified Entropy | Confidence | Quantile | Merlin | MLLeak1 | MLLeak3 | Shadow | One-class | Diff-w | Diff-single | Diff-bi |
| 200th | Acc | Attack | 64.52 | 72.25 | 78.32 | 80.25 | 80.26 | 80.89 | 55.77 | 71.24 | 76.85 | 62.32 | 75.45 | 77.91 | 74.71 | 66.46 |
| | | Audit | 69.04 | 73.48 | 81.04 | 82.29 | 82.22 | 81.00 | 56.90 | | | | | | | |
| | Prec@P | Attack | 67.18 | 71.23 | 70.59 | 72.33 | 72.36 | 73.29 | 53.10 | 63.50 | 75.72 | 57.03 | 67.18 | 69.41 | 66.47 | 60.60 |
| | | Audit | 70.82 | 65.36 | 75.95 | 76.90 | 76.98 | 73.73 | 57.69 | | | | | | | |
| | Rec@P | Attack | 56.78 | 74.64 | 97.10 | 97.98 | 97.94 | 97.22 | 98.74 | 99.88 | 79.04 | 100.00 | 99.54 | 99.80 | 99.74 | 83.44 |
| | | Audit | 64.74 | 99.92 | 90.84 | 92.30 | 91.94 | 96.32 | 51.76 | | | | | | | |
| | F1@P | Attack | 61.54 | 72.90 | 81.75 | 83.22 | 83.23 | 83.57 | 69.06 | 77.64 | 77.35 | 72.63 | 80.22 | 81.88 | 79.77 | 70.21 |
| | | Audit | 67.64 | 79.03 | 82.73 | 83.90 | 83.80 | 83.52 | 54.56 | | | | | | | |
| | Prec@N | Attack | 62.57 | 73.37 | 95.36 | 96.87 | 96.81 | 95.87 | 91.04 | 99.72 | 78.08 | 100.00 | 99.11 | 99.64 | 99.48 | 73.43 |
| | | Audit | 67.53 | 99.83 | 88.61 | 90.37 | 90.00 | 94.69 | 56.26 | | | | | | | |
| | Rec@N | Attack | 72.26 | 69.86 | 59.54 | 62.52 | 62.58 | 64.56 | 12.80 | 42.60 | 74.66 | 24.64 | 51.26 | 54.02 | 49.68 | 45.76 |
| | | Audit | 73.32 | 47.04 | 71.24 | 72.28 | 72.50 | 65.68 | 62.04 | | | | | | | |
| | F1@N | Attack | 67.07 | 71.57 | 73.31 | 75.99 | 76.02 | 77.16 | 22.44 | 59.70 | 76.33 | 39.64 | 67.66 | 71.72 | 66.27 | 56.38 |
| | | Audit | 70.30 | 63.95 | 78.98 | 80.32 | 80.31 | 77.56 | 59.01 | | | | | | | |
| | AUC | Audit | 0.734 | 0.824 | 0.825 | 0.836 | 0.834 | 0.840 | 0.518 | – | – | – | – | – | – | – |
| | TPR@0.1%FPR | Audit | 0.66 | 6.40 | 7.455 | 7.968 | 7.817 | 0.52 | 1.481 | – | – | – | – | – | – | – |
| 110th | Acc | Attack | 61.45 | 72.34 | 70.97 | 73.85 | 73.78 | 73.06 | 54.28 | 59.34 | 55.66 | 60.23 | 70.83 | 73.67 | 70.28 | 64.42 |
| | | Audit | 65.83 | 73.62 | 74.04 | 76.52 | 76.55 | 73.96 | 56.76 | | | | | | | |
| | Prec@P | Attack | 63.80 | 71.00 | 65.68 | 67.54 | 67.49 | 65.92 | 52.30 | 55.17 | 67.32 | 55.72 | 67.19 | 67.37 | 64.47 | 62.70 |
| | | Audit | 69.66 | 66.25 | 70.77 | 75.21 | 72.48 | 68.39 | 57.64 | | | | | | | |
| | Rec@P | Attack | 52.94 | 75.52 | 87.82 | 91.82 | 91.76 | 95.50 | 97.52 | 99.62 | 22.00 | 99.72 | 81.40 | 91.80 | 90.34 | 71.18 |
| | | Audit | 56.08 | 96.30 | 81.90 | 85.42 | 85.60 | 89.10 | 51.02 | | | | | | | |
| | F1@P | Attack | 57.86 | 73.19 | 75.16 | 77.83 | 77.78 | 78.00 | 68.08 | 71.02 | 33.16 | 71.49 | 73.62 | 77.71 | 75.25 | 66.67 |
| | | Audit | 62.14 | 78.50 | 75.93 | 78.44 | 78.50 | 77.38 | 54.13 | | | | | | | |
| | Prec@N | Attack | 59.78 | 73.86 | 81.63 | 87.23 | 87.13 | 91.84 | 81.66 | 98.05 | 53.38 | 98.67 | 76.41 | 87.14 | 83.87 | 66.67 |
| | | Audit | 63.25 | 93.23 | 78.52 | 82.26 | 82.42 | 84.37 | 56.06 | | | | | | | |
| | Rec@N | Attack | 69.96 | 69.19 | 54.12 | 55.88 | 55.80 | 50.62 | 11.04 | 19.06 | 89.32 | 20.74 | 60.26 | 55.54 | 50.22 | 57.66 |
| | | Audit | 75.58 | 50.94 | 66.18 | 67.62 | 67.50 | 58.82 | 62.50 | | | | | | | |
| | F1@N | Attack | 64.47 | 71.43 | 65.09 | 68.12 | 68.03 | 65.27 | 19.45 | 31.92 | 66.83 | 34.28 | 67.38 | 67.84 | 62.82 | 61.84 |
| | | Audit | 68.87 | 65.88 | 71.83 | 74.23 | 74.22 | 69.31 | 59.11 | | | | | | | |
| | AUC | Audit | 0.698 | 0.823 | 0.756 | 0.783 | 0.781 | 0.772 | 0.517 | – | – | – | – | – | – | – |
| | TPR@0.1%FPR | Audit | 0.60 | 4.70 | 5.788 | 6.397 | 6.148 | 0.36 | 1.598 | – | – | – | – | – | – | – |
| 10th | Acc | Attack | 50.87 | 51.68 | 50.39 | 51.24 | 50.78 | 51.36 | 50.24 | 50.00 | 50.66 | 50.06 | 50.31 | 50.07 | 49.82 | 49.81 |
| | | Audit | 51.22 | 51.73 | 51.26 | 57.77 | 57.78 | 51.81 | 56.04 | | | | | | | |
| | Prec@P | Attack | 52.25 | 51.62 | 50.32 | 50.91 | 50.56 | 50.87 | 50.17 | 50.00 | 51.17 | 50.03 | 51.26 | 50.49 | 49.88 | 49.78 |
| | | Audit | 51.13 | 51.71 | 58.20 | 58.33 | 58.14 | 51.72 | 54.30 | | | | | | | |
| | Rec@P | Attack | 20.18 | 53.64 | 61.78 | 69.28 | 70.12 | 79.62 | 71.36 | 100.00 | 28.92 | 98.44 | 12.60 | 7.22 | 73.58 | 42.82 |
| | | Audit | 55.02 | 52.32 | 49.54 | 54.42 | 55.54 | 54.44 | 53.96 | | | | | | | |
| | F1@P | Attack | 29.12 | 52.61 | 55.46 | 58.69 | 58.76 | 62.08 | 58.92 | 66.67 | 39.65 | 66.34 | 20.23 | 12.63 | 59.45 | 46.04 |
| | | Audit | 53.01 | 52.01 | 53.52 | 56.31 | 56.81 | 53.04 | 55.11 | | | | | | | |
| | Prec@N | Attack | 50.54 | 51.75 | 50.51 | 51.94 | 51.27 | 53.13 | 50.42 | 0.00 | 50.46 | 51.85 | 50.18 | 50.04 | 49.66 | 49.83 |
| | | Audit | 51.32 | 51.75 | 56.08 | 57.28 | 57.45 | 51.91 | 55.80 | | | | | | | |
| | Rec@N | Attack | 81.56 | 49.72 | 39.00 | 33.20 | 31.44 | 23.10 | 29.12 | 0.00 | 72.40 | 1.68 | 88.02 | 92.92 | 26.06 | 56.80 |
| | | Audit | 47.42 | 51.14 | 64.42 | 61.12 | 60.02 | 49.18 | 58.12 | | | | | | | |
| | F1@N | Attack | 62.41 | 50.71 | 44.01 | 40.51 | 38.98 | 32.20 | 36.92 | 0.00 | 59.47 | 3.25 | 63.92 | 65.05 | 34.18 | 53.09 |
| | | Audit | 49.29 | 51.44 | 59.96 | 59.14 | 58.71 | 50.51 | 56.94 | | | | | | | |
| | AUC | Audit | 0.511 | 0.521 | 0.510 | 0.522 | 0.522 | 0.519 | 0.501 | – | – | – | – | – | – | – |
| | TPR@0.1%FPR | Audit | 0.06 | 0.20 | 2.311 | 1.990 | 1.987 | 0.200 | 2.025 | – | – | – | – | – | – | – |

Table 14: The performance of different MIA methods for CIFAR-100 under Swin Transformer-Tiny **with pretrain**. The model is pretrained on ImageNet-1k.

| Epoch | Metric | Mode | Membership Inference Attack Algorithms | | | | | | | | | | | | | |
| | | | Metric MIA | | | | | | | ML-Leaks | | | BlindMI | | | |
| | | | LiRA | RMIA | Entropy | Modified Entropy | Confidence | Quantile | Merlin | MLLeak1 | MLLeak3 | Shadow | One-class | Diff-w | Diff-single | Diff-bi |
| 200th | Acc | Attack | 56.59 | 57.16 | 63.94 | 65.01 | 64.78 | 65.24 | 52.51 | 56.31 | 57.59 | 55.19 | 61.27 | 61.89 | 63.04 | 53.40 |
| | | Audit | 57.31 | 62.43 | 66.91 | 67.93 | 67.84 | 65.16 | 56.43 | | | | | | | |
| | Prec@P | Attack | 62.21 | 70.69 | 59.17 | 59.88 | 59.71 | 59.78 | 51.33 | 53.37 | 54.11 | 52.74 | 58.83 | 56.76 | 57.66 | 52.57 |
| | | Audit | 59.22 | 57.15 | 62.98 | 63.52 | 63.49 | 59.60 | 56.75 | | | | | | | |
| | Rec@P | Attack | 33.58 | 24.46 | 89.96 | 90.94 | 90.86 | 93.16 | 97.18 | 99.98 | 99.94 | 99.98 | 75.12 | 99.78 | 98.12 | 69.56 |
| | | Audit | 46.94 | 99.32 | 82.04 | 84.26 | 83.96 | 94.14 | 54.06 | | | | | | | |
| | F1@P | Attack | 43.62 | 36.34 | 71.39 | 72.21 | 72.07 | 72.83 | 67.17 | 69.59 | 70.21 | 69.05 | 65.98 | 72.36 | 72.64 | 59.88 |
| | | Audit | 52.37 | 72.55 | 71.26 | 72.43 | 72.30 | 72.99 | 55.37 | | | | | | | |
| | Prec@N | Attack | 54.51 | 54.33 | 79.07 | 81.18 | 80.89 | 84.51 | 73.55 | 99.84 | 99.61 | 99.81 | 65.59 | 99.09 | 93.70 | 55.02 |
| | | Audit | 56.05 | 97.41 | 74.25 | 76.63 | 76.33 | 86.06 | 56.14 | | | | | | | |
| | Rec@N | Attack | 79.60 | 89.86 | 37.92 | 39.08 | 38.70 | 37.32 | 7.84 | 12.64 | 15.24 | 10.40 | 47.42 | 24.00 | 27.96 | 37.24 |
| | | Audit | 67.68 | 25.54 | 51.78 | 51.60 | 51.72 | 36.18 | 58.80 | | | | | | | |
| | F1@N | Attack | 64.71 | 67.72 | 51.26 | 52.76 | 52.35 | 51.78 | 14.17 | 22.44 | 26.44 | 18.84 | 55.04 | 38.64 | 43.07 | 44.42 |
| | | Audit | 61.32 | 40.47 | 61.01 | 61.67 | 61.66 | 50.94 | 57.44 | | | | | | | |
| | AUC | Audit | 0.585 | 0.682 | 0.644 | 0.667 | 0.652 | 0.637 | 0.514 | – | – | – | – | – | – | – |
| | TPR@0.1%FPR | Audit | 0.26 | 9.68 | 2.81 | 2.36 | 2.16 | 0.18 | 1.83 | – | – | – | – | – | – | – |
| 110th | Acc | Attack | 56.20 | 58.93 | 63.44 | 64.38 | 64.05 | 64.37 | 52.45 | 58.94 | 57.44 | 55.09 | 59.24 | 61.58 | 62.65 | 52.82 |
| | | Audit | 57.21 | 61.07 | 66.59 | 67.50 | 67.41 | 64.70 | 56.34 | | | | | | | |
| | Prec@P | Attack | 64.37 | 68.55 | 58.81 | 59.60 | 59.44 | 59.17 | 51.30 | 54.93 | 54.03 | 52.68 | 57.95 | 56.58 | 57.49 | 52.11 |
| | | Audit | 59.26 | 56.23 | 62.58 | 63.32 | 63.23 | 59.33 | 56.54 | | | | | | | |
| | Rec@P | Attack | 27.78 | 33.00 | 89.68 | 89.28 | 88.48 | 92.76 | 96.82 | 99.58 | 99.86 | 99.98 | 67.32 | 99.58 | 97.10 | 69.68 |
| | | Audit | 46.14 | 99.86 | 82.52 | 83.20 | 83.20 | 93.50 | 54.78 | | | | | | | |
| | F1@P | Attack | 38.81 | 44.55 | 71.04 | 71.48 | 71.11 | 72.25 | 67.06 | 70.80 | 70.12 | 69.00 | 62.29 | 72.16 | 72.22 | 59.63 |
| | | Audit | 51.88 | 71.95 | 71.18 | 71.91 | 71.85 | 72.59 | 55.65 | | | | | | | |
| | Prec@N | Attack | 53.95 | 55.88 | 78.28 | 78.65 | 77.47 | 83.25 | 71.76 | 97.76 | 99.08 | 99.80 | 61.02 | 98.25 | 90.68 | 54.25 |
| | | Audit | 55.90 | 99.38 | 74.35 | 75.51 | 75.45 | 84.67 | 56.15 | | | | | | | |
| | Rec@N | Attack | 84.62 | 84.86 | 37.20 | 39.48 | 39.62 | 35.98 | 8.08 | 18.30 | 15.02 | 10.20 | 51.16 | 23.58 | 28.20 | 35.96 |
| | | Audit | 68.28 | 22.28 | 50.66 | 51.80 | 51.62 | 35.90 | 57.90 | | | | | | | |
| | F1@N | Attack | 65.89 | 67.39 | 50.43 | 52.57 | 52.43 | 50.24 | 14.52 | 30.83 | 26.09 | 18.51 | 55.66 | 38.03 | 43.02 | 43.25 |
| | | Audit | 61.47 | 36.40 | 60.26 | 61.45 | 61.30 | 50.42 | 57.01 | | | | | | | |
| | AUC | Audit | 0.584 | 0.670 | 0.641 | 0.664 | 0.650 | 0.640 | 0.515 | – | – | – | – | – | – | – |
| | TPR@0.1%FPR | Audit | 0.22 | 9.86 | 2.89 | 2.45 | 2.03 | 0.58 | 1.96 | – | – | – | – | – | – | – |
| 10th | Acc | Attack | 52.68 | 54.14 | 52.04 | 52.92 | 53.16 | 52.28 | 50.79 | 50.00 | 50.52 | 50.29 | 52.12 | 52.66 | 52.18 | 51.64 |
| | | Audit | 53.63 | 54.78 | 58.55 | 59.31 | 59.23 | 53.26 | 56.40 | | | | | | | |
| | Prec@P | Attack | 54.75 | 56.79 | 51.65 | 52.38 | 52.46 | 68.87 | 50.43 | 50.00 | 50.27 | 50.15 | 51.96 | 52.02 | 51.72 | 51.56 |
| | | Audit | 54.87 | 53.73 | 59.21 | 59.68 | 59.66 | 52.11 | 56.89 | | | | | | | |
| | Rec@P | Attack | 30.88 | 34.62 | 63.82 | 64.28 | 67.32 | 8.32 | 92.00 | 100.00 | 96.34 | 98.52 | 56.24 | 68.38 | 65.72 | 54.14 |
| | | Audit | 40.90 | 68.94 | 55.48 | 57.40 | 57.02 | 80.36 | 52.86 | | | | | | | |
| | F1@P | Attack | 39.49 | 43.02 | 57.09 | 57.72 | 58.97 | 14.85 | 65.15 | 66.67 | 66.07 | 66.46 | 54.01 | 59.09 | 57.88 | 52.82 |
| | | Audit | 46.87 | 60.39 | 57.24 | 58.52 | 58.31 | 63.23 | 54.80 | | | | | | | |
| | Prec@N | Attack | 51.87 | 52.98 | 52.67 | 53.78 | 54.41 | 51.21 | 54.49 | 0.00 | 56.22 | 58.19 | 52.31 | 53.88 | 52.99 | 51.73 |
| | | Audit | 52.89 | 56.67 | 58.06 | 58.97 | 58.84 | 57.12 | 55.98 | | | | | | | |
| | Rec@N | Attack | 74.48 | 73.66 | 40.26 | 41.56 | 39.00 | 96.24 | 9.58 | 0.00 | 4.70 | 2.06 | 48.00 | 36.94 | 38.64 | 49.14 |
| | | Audit | 66.36 | 40.62 | 61.62 | 61.22 | 61.44 | 26.16 | 59.94 | | | | | | | |
| | F1@N | Attack | 61.15 | 61.63 | 45.64 | 46.89 | 45.43 | 66.85 | 16.30 | 0.00 | 8.67 | 3.98 | 50.06 | 43.83 | 44.69 | 50.40 |
| | | Audit | 58.87 | 47.32 | 59.78 | 60.07 | 60.11 | 35.88 | 57.89 | | | | | | | |
| | AUC | Audit | 0.534 | 0.573 | 0.534 | 0.549 | 0.548 | 0.542 | 0.505 | – | – | – | – | – | – | – |
| | TPR@0.1%FPR | Audit | 0.18 | 2.72 | 3.23 | 3.23 | 3.25 | 1.40 | 1.48 | – | – | – | – | – | – | – |

### G.3 PERFORMANCE OF MIA METHODS IN DIFFERENT DATASETS

Table 15: The performance of different MIA methods for CIFAR-10 under ResNet-18.

| Epoch | Metric | Mode | Membership Inference Attack Algorithms | | | | | | | | | | | | | |
|---|---|---|---|---|---|---|---|---|---|---|---|---|---|---|---|---|
| | | | Metric MIA | | | | | | | ML-Leaks | | | BlindMI | | | |
| | | | LiRA | RMIA | Entropy | Modified Entropy | Confidence | Quantile | Merlin | MLLeak1 | MLLeak3 | Shadow | One-class | Diff-w | Diff-single | Diff-bi |
| 200th | Acc | Attack | 50.93 | 55.41 | 59.87 | 60.55 | 60.68 | 60.98 | 53.51 | 52.90 | 59.74 | 54.13 | 59.30 | 59.38 | 59.34 | 51.88 |
| | | Audit | 52.68 | 58.33 | 65.68 | 66.17 | 66.17 | 61.19 | 56.87 | | | | | | | |
| | Prec@P | Attack | 52.72 | 65.68 | 56.33 | 56.62 | 56.76 | 57.39 | 52.18 | 51.49 | 57.38 | 52.15 | 55.33 | 55.21 | 55.33 | 51.31 |
| | | Audit | 53.91 | 54.58 | 61.04 | 61.22 | 61.23 | 57.24 | 56.21 | | | | | | | |
| | Rec@P | Attack | 18.04 | 22.66 | 87.80 | 90.20 | 89.68 | 85.26 | 83.92 | 100.00 | 75.74 | 100.00 | 96.60 | 99.36 | 96.94 | 73.40 |
| | | Audit | 36.92 | 99.28 | 86.70 | 88.20 | 88.16 | 88.52 | 62.18 | | | | | | | |
| | F1@P | Attack | 26.88 | 33.70 | 68.63 | 69.57 | 69.52 | 68.60 | 64.36 | 67.98 | 65.29 | 68.55 | 70.36 | 70.98 | 70.45 | 60.40 |
| | | Audit | 43.18 | 70.44 | 71.64 | 72.28 | 72.27 | 69.52 | 59.04 | | | | | | | |
| | Prec@N | Attack | 50.56 | 53.27 | 72.36 | 75.92 | 75.43 | 71.35 | 58.97 | 100.00 | 64.32 | 100.00 | 86.61 | 96.81 | 87.66 | 53.30 |
| | | Audit | 52.04 | 96.02 | 77.05 | 78.91 | 78.86 | 74.68 | 57.69 | | | | | | | |
| | Rec@N | Attack | 83.82 | 88.16 | 31.94 | 30.90 | 31.68 | 36.70 | 23.08 | 5.80 | 43.74 | 8.26 | 22.00 | 19.40 | 21.74 | 30.36 |
| | | Audit | 68.44 | 17.38 | 44.66 | 44.14 | 44.18 | 33.86 | 51.56 | | | | | | | |
| | F1@N | Attack | 63.07 | 66.41 | 44.32 | 43.92 | 44.62 | 48.47 | 33.18 | 10.96 | 52.07 | 15.25 | 35.09 | 32.32 | 34.84 | 38.69 |
| | | Audit | 59.12 | 29.43 | 56.55 | 56.61 | 56.63 | 46.59 | 54.45 | | | | | | | |
| | AUC | Audit | 0.508 | 0.621 | 0.631 | 0.635 | 0.637 | 0.631 | 0.509 | – | – | – | – | – | – | – |
| | TPR@0.1%FPR | Audit | 0.16 | 0.70 | 0.36 | 0.41 | 0.36 | 0.40 | 0.22 | – | – | – | – | – | – | – |
| 110th | Acc | Attack | 50.54 | 56.22 | 56.06 | 56.43 | 56.67 | 56.71 | 52.46 | 50.00 | 55.80 | 51.63 | 55.65 | 56.97 | 55.97 | 52.64 |
| | | Audit | 52.81 | 57.95 | 61.66 | 62.40 | 62.31 | 56.77 | 56.58 | | | | | | | |
| | Prec@P | Attack | 51.81 | 61.17 | 53.88 | 54.02 | 54.15 | 54.21 | 51.32 | 50.00 | 53.54 | 50.83 | 53.48 | 54.25 | 53.70 | 51.94 |
| | | Audit | 54.24 | 54.69 | 58.19 | 58.49 | 58.55 | 54.20 | 56.31 | | | | | | | |
| | Rec@P | Attack | 15.48 | 34.06 | 84.10 | 86.38 | 87.12 | 86.34 | 95.58 | 100.00 | 87.74 | 99.84 | 86.88 | 88.96 | 86.70 | 70.70 |
| | | Audit | 35.92 | 92.66 | 82.84 | 85.40 | 84.34 | 87.28 | 58.68 | | | | | | | |
| | F1@P | Attack | 23.84 | 43.76 | 65.68 | 66.47 | 66.78 | 66.60 | 66.78 | 66.67 | 66.50 | 67.36 | 66.20 | 67.40 | 66.32 | 59.88 |
| | | Audit | 43.22 | 68.78 | 68.36 | 69.43 | 69.11 | 66.88 | 57.47 | | | | | | | |
| | Prec@N | Attack | 50.32 | 54.31 | 63.80 | 66.03 | 67.06 | 66.47 | 67.88 | 0.00 | 66.06 | 95.53 | 65.05 | 69.35 | 65.49 | 54.13 |
| | | Audit | 52.10 | 76.00 | 70.23 | 72.96 | 72.01 | 67.37 | 56.87 | | | | | | | |
| | Rec@N | Attack | 85.60 | 78.38 | 28.02 | 26.48 | 26.22 | 27.08 | 9.34 | 0.00 | 23.86 | 3.42 | 24.42 | 24.98 | 25.24 | 34.58 |
| | | Audit | 69.70 | 23.24 | 40.48 | 39.40 | 40.28 | 26.26 | 54.48 | | | | | | | |
| | F1@N | Attack | 63.38 | 64.16 | 38.94 | 37.80 | 37.70 | 38.48 | 16.42 | 0.00 | 35.06 | 6.60 | 35.51 | 36.73 | 36.44 | 42.20 |
| | | Audit | 59.63 | 35.60 | 51.36 | 51.17 | 51.66 | 37.79 | 55.65 | | | | | | | |
| | AUC | Audit | 0.520 | 0.620 | 0.565 | 0.578 | 0.575 | 0.571 | 0.503 | – | – | – | – | – | – | – |
| | TPR@0.1%FPR | Audit | 0.06 | 1.1 | 0.34 | 0.32 | 0.34 | 0.26 | 0.09 | – | – | – | – | – | – | – |
| 10th | Acc | Attack | 50.09 | 51.44 | 50.70 | 51.06 | 50.97 | 51.42 | 49.86 | 50.00 | 50.31 | 50.02 | 50.15 | 50.86 | 51.21 | 49.76 |
| | | Audit | 50.11 | 52.13 | 58.01 | 57.90 | | 51.80 | 56.93 | | | | | | | |
| | Prec@P | Attack | 50.55 | 51.39 | 50.57 | 50.62 | 50.55 | 50.83 | 49.91 | 50.00 | 50.29 | 50.01 | 50.40 | 51.67 | 51.04 | 49.72 |
| | | Audit | 51.18 | 51.65 | 57.20 | 57.34 | 57.19 | 51.15 | 57.62 | | | | | | | |
| | Rec@P | Attack | 8.32 | 53.30 | 62.42 | 86.10 | 88.42 | 86.64 | 75.42 | 100.00 | 54.28 | 99.92 | 18.80 | 26.62 | 59.66 | 42.16 |
| | | Audit | 4.76 | 66.72 | 59.30 | 62.58 | 62.82 | 80.14 | 52.42 | | | | | | | |
| | F1@P | Attack | 14.29 | 52.33 | 55.87 | 63.76 | 64.33 | 64.07 | 60.07 | 66.67 | 52.21 | 66.66 | 27.39 | 35.14 | 55.01 | 45.63 |
| | | Audit | 8.71 | 58.22 | 58.23 | 59.85 | 59.87 | 62.44 | 54.90 | | | | | | | |
| | Prec@N | Attack | 50.05 | 51.50 | 50.91 | 53.54 | 53.86 | 54.80 | 49.71 | 0.00 | 50.32 | 60.00 | 50.09 | 50.58 | 51.46 | 49.59 |
| | | Audit | 50.56 | 53.01 | 57.75 | 58.82 | 58.76 | 54.16 | 56.36 | | | | | | | |
| | Rec@N | Attack | 91.86 | 49.58 | 38.98 | 16.02 | 13.52 | 16.20 | 24.30 | 0.00 | 46.32 | 0.12 | 81.50 | 75.10 | 42.76 | 57.36 |
| | | Audit | 95.46 | 37.54 | 55.62 | 53.44 | 52.98 | 23.46 | 61.44 | | | | | | | |
| | F1@N | Attack | 64.80 | 50.52 | 44.15 | 24.66 | 21.61 | 25.01 | 32.64 | 0.00 | 48.26 | 0.24 | 62.05 | 60.45 | 46.71 | 53.31 |
| | | Audit | 65.68 | 43.95 | 55.66 | 56.00 | 55.72 | 32.74 | 58.79 | | | | | | | |
| | AUC | Audit | 0.492 | 0.521 | 0.505 | 0.518 | 0.518 | 0.523 | 0.505 | – | – | – | – | – | – | – |
| | TPR@0.1%FPR | Audit | 0.06 | 0.18 | 0.35 | 0.34 | 0.34 | 0.40 | 0.07 | – | – | – | – | – | – | – |

For CIFAR-10 under ResNet-18. Table 15 indicates that: Modified Entropy attained the highest accuracy under audit mode, whereas quantile MIA achieved the best accuracy in attack mode at 200,110, and 10 epochs. Regarding AUC and TPR at 0.1% FPR, RMIA outperformed other methods at epochs 200 and 110, while Modified Entropy MIA performed best at epoch 10. Additional, Metric MIA showed the largest performance gap between audit and attack modes, while Quantile MIA exhibited the smallest difference.

### G.4 PERFORMANCE OF MIA METHODS UNDER MISLABEL SCENARIO

Table 16: The performance of different MIA methods for ImageNet100 under WideResNet-28-2.

| Epoch | Metric | Mode | Metric MIA | | | | | | | ML-Leaks | | | BlindMI | | | |
|---|---|---|---|---|---|---|---|---|---|---|---|---|---|---|---|---|
| | | | LiRA | RMIA | Entropy | Modified Entropy | Confidence | Quantile | Merlin | MLLeak1 | MLLeak3 | Shadow | One-class | Diff-w | Diff-single | Diff-bi |
| 200th | Acc | Attack | 51.48 | 60.32 | 54.12 | 56.18 | 55.98 | 51.06 | 52.10 | 50.00 | 51.06 | 50.32 | 53.32 | 56.76 | 56.14 | 54.10 |
| | | Audit | 57.46 | 60.58 | 61.64 | 63.08 | 62.90 | 57.27 | 52.62 | | | | | | | |
| | Prec@P | Attack | 50.75 | 58.43 | 53.80 | 54.94 | 54.86 | 50.54 | 51.14 | 50.00 | 50.54 | 50.16 | 51.88 | 54.93 | 55.00 | 53.99 |
| | | Audit | 54.71 | 58.39 | 62.04 | 62.36 | 62.77 | 54.63 | 56.53 | | | | | | | |
| | Rec@P | Attack | 99.68 | 71.52 | 58.28 | 68.68 | 67.56 | 98.32 | 96.01 | 100.00 | 98.32 | 100.00 | 91.76 | 75.36 | 67.48 | 55.52 |
| | | Audit | 86.60 | 73.64 | 59.96 | 66.00 | 63.40 | 85.77 | 22.66 | | | | | | | |
| | F1@P | Attack | 67.26 | 64.32 | 55.95 | 61.05 | 60.55 | 66.77 | 66.73 | 66.67 | 66.77 | 66.81 | 66.28 | 63.54 | 60.61 | 54.74 |
| | | Audit | 67.06 | 65.13 | 60.98 | 64.13 | 63.08 | 66.75 | 32.35 | | | | | | | |
| | Prec@N | Attack | 91.11 | 63.30 | 54.49 | 58.24 | 57.78 | 69.34 | 67.00 | 0.00 | 69.34 | 100.00 | 64.36 | 60.76 | 57.94 | 54.22 |
| | | Audit | 67.88 | 64.32 | 61.26 | 63.89 | 63.03 | 66.91 | 51.64 | | | | | | | |
| | Rec@N | Attack | 3.28 | 49.12 | 49.96 | 43.68 | 44.40 | 3.80 | 8.12 | 0.00 | 3.80 | 0.64 | 14.88 | 38.16 | 44.80 | 52.68 |
| | | Audit | 28.32 | 47.52 | 63.32 | 60.16 | 62.40 | 28.78 | 82.57 | | | | | | | |
| | F1@N | Attack | 6.33 | 55.32 | 52.13 | 49.92 | 50.21 | 7.21 | 14.48 | 0.00 | 7.21 | 1.27 | 24.17 | 46.88 | 50.53 | 53.44 |
| | | Audit | 39.97 | 54.66 | 62.27 | 61.97 | 62.71 | 40.25 | 63.54 | | | | | | | |
| | AUC | Audit | 0.584 | 0.652 | 0.567 | 0.592 | 0.592 | 0.589 | 0.523 | – | – | – | – | – | – | – |
| | TPR@0.1%FPR | Audit | 0.0 | 5.4 | 5.3 | 6.5 | 6.2 | 0.3 | 0.0 | – | – | – | – | – | – | – |
| 110th | Acc | Attack | 50.74 | 54.62 | 52.84 | 53.90 | 53.92 | 53.58 | 50.00 | 50.00 | 50.20 | 50.16 | 51.32 | 52.90 | 52.70 | 52.22 |
| | | Audit | 54.04 | 55.14 | 59.70 | 60.70 | 60.84 | 53.92 | 53.08 | | | | | | | |
| | Prec@P | Attack | 50.38 | 54.06 | 53.60 | 53.62 | 53.60 | 53.65 | 50.00 | 50.00 | 50.10 | 50.08 | 50.73 | 52.21 | 52.25 | 52.51 |
| | | Audit | 52.86 | 54.12 | 60.34 | 61.59 | 61.27 | 53.87 | 57.01 | | | | | | | |
| | Rec@P | Attack | 99.20 | 61.52 | 42.28 | 57.72 | 58.44 | 53.15 | 100.00 | 100.00 | 98.88 | 99.84 | 91.76 | 68.52 | 62.72 | 46.36 |
| | | Audit | 74.72 | 67.48 | 58.00 | 56.88 | 58.92 | 54.52 | 25.02 | | | | | | | |
| | F1@P | Attack | 66.82 | 57.55 | 47.27 | 55.60 | 55.91 | 53.40 | 66.67 | 66.67 | 66.51 | 66.70 | 65.34 | 59.26 | 57.01 | 49.25 |
| | | Audit | 61.92 | 60.07 | 59.15 | 59.14 | 60.07 | 54.19 | 34.78 | | | | | | | |
| | Prec@N | Attack | 74.03 | 55.36 | 52.34 | 54.22 | 54.31 | 53.51 | 0.00 | 0.00 | 57.58 | 75.00 | 56.90 | 54.22 | 53.38 | 51.99 |
| | | Audit | 56.89 | 56.82 | 59.57 | 59.94 | 60.44 | 53.96 | 51.97 | | | | | | | |
| | Rec@N | Attack | 2.28 | 47.72 | 63.40 | 50.08 | 49.40 | 54.00 | 0.00 | 0.00 | 1.52 | 0.48 | 10.88 | 37.28 | 42.68 | 58.08 |
| | | Audit | 33.36 | 42.80 | 61.88 | 64.52 | 62.76 | 53.32 | 81.14 | | | | | | | |
| | F1@N | Attack | 4.42 | 51.26 | 57.34 | 52.07 | 51.74 | 53.75 | 0.00 | 0.00 | 2.96 | 0.95 | 18.27 | 44.18 | 47.43 | 54.86 |
| | | Audit | 42.06 | 48.83 | 60.70 | 62.15 | 61.58 | 53.64 | 63.36 | | | | | | | |
| | AUC | Audit | 0.548 | 0.567 | 0.535 | 0.553 | 0.552 | 0.552 | 0.514 | – | – | – | – | – | – | – |
| | TPR@0.1%FPR | Audit | 0.0 | 2.2 | 5.2 | 5.3 | 5.4 | 0.3 | 0.0 | – | – | – | – | – | – | – |
| 10th | Acc | Attack | 50.40 | 51.74 | 50.28 | 50.96 | 50.68 | 51.58 | 50.00 | 50.00 | 50.44 | 49.92 | 50.18 | 49.24 | 50.76 | 50.92 |
| | | Audit | 52.14 | 51.88 | 58.60 | 58.80 | 59.14 | 51.94 | 53.78 | | | | | | | |
| | Prec@P | Attack | 50.21 | 51.73 | 51.12 | 52.68 | 51.88 | 51.30 | 50.00 | 50.00 | 50.28 | 49.96 | 50.89 | 48.89 | 50.52 | 51.04 |
| | | Audit | 51.86 | 52.96 | 59.70 | 60.46 | 60.28 | 52.29 | 55.84 | | | | | | | |
| | Rec@P | Attack | 96.24 | 52.12 | 12.76 | 18.88 | 18.72 | 63.78 | 100.00 | 100.00 | 79.84 | 99.48 | 10.32 | 33.40 | 73.76 | 45.00 |
| | | Audit | 59.68 | 33.68 | 52.92 | 50.88 | 53.60 | 44.24 | 36.09 | | | | | | | |
| | F1@P | Attack | 65.99 | 51.92 | 20.42 | 27.80 | 27.51 | 56.86 | 66.67 | 66.67 | 61.70 | 66.52 | 17.16 | 39.69 | 59.97 | 47.83 |
| | | Audit | 55.50 | 41.17 | 56.11 | 55.26 | 56.74 | 47.93 | 43.85 | | | | | | | |
| | Prec@N | Attack | 54.81 | 51.75 | 50.16 | 50.58 | 50.41 | 52.04 | 0.00 | 0.00 | 51.07 | 40.91 | 50.10 | 49.42 | 51.41 | 50.82 |
| | | Audit | 52.52 | 51.38 | 57.72 | 57.60 | 58.23 | 51.68 | 52.79 | | | | | | | |
| | Rec@N | Attack | 4.56 | 51.36 | 87.80 | 83.04 | 82.64 | 39.36 | 0.00 | 0.00 | 21.04 | 0.36 | 90.04 | 65.08 | 27.76 | 56.84 |
| | | Audit | 44.60 | 70.08 | 64.28 | 66.72 | 64.68 | 59.63 | 71.46 | | | | | | | |
| | F1@N | Attack | 8.42 | 51.56 | 63.85 | 62.87 | 62.63 | 44.82 | 0.00 | 0.00 | 29.80 | 0.71 | 64.38 | 56.18 | 36.05 | 53.66 |
| | | Audit | 48.24 | 59.29 | 60.83 | 61.82 | 61.28 | 55.37 | 60.72 | | | | | | | |
| | AUC | Audit | 0.522 | 0.522 | 0.511 | 0.519 | 0.520 | 0.519 | 0.506 | – | – | – | – | – | – | – |
| | TPR@0.1%FPR | Audit | 0.0 | 1.1 | 4.3 | 5.6 | 5.4 | 4.0 | 0.0 | – | – | – | – | – | – | – |

Table 17: The performance of different MIA methods for mislabeled (10%) CIFAR-100 under ResNet-18.

| Epoch | Metric | Mode | LiRA | RMIA | Entropy | Modified Entropy | Confidence | Quantile | Merlin | ML-Leak1 | ML-Leak3 | Shadow | One-class | Diff-w | Diff-single | Diff-bi |
|---|---|---|---|---|---|---|---|---|---|---|---|---|---|---|---|---|
| 200th | Acc | Attack | 69.03 | 79.95 | 83.55 | 86.25 | 86.36 | 86.18 | 62.19 | 80.55 | 58.63 | 66.53 | 82.80 | 84.77 | 79.44 | 72.37 |
|  |  | Audit | 73.75 | 80.98 | 85.07 | 87.51 | 87.43 | 86.28 | 52.15 |  |  |  |  |  |  |  |
|  | Prec@P | Attack | 73.64 | 74.82 | 77.22 | 80.03 | 80.06 | 79.93 | 57.06 | 72.12 | 78.22 | 59.91 | 74.99 | 76.98 | 70.97 | 67.32 |
|  |  | Audit | 75.01 | 72.77 | 80.19 | 83.05 | 82.79 | 80.60 | 67.89 |  |  |  |  |  |  |  |
|  | Rec@P | Attack | 59.28 | 90.28 | 95.18 | 96.60 | 96.84 | 96.62 | 98.56 | 99.62 | 23.92 | 99.96 | 98.42 | 99.20 | 99.64 | 86.94 |
|  |  | Audit | 71.24 | 99.02 | 93.16 | 94.26 | 94.50 | 95.56 | 8.16 |  |  |  |  |  |  |  |
|  | F1@P | Attack | 65.68 | 81.83 | 85.26 | 87.54 | 87.65 | 87.49 | 72.27 | 83.67 | 36.64 | 74.92 | 85.12 | 86.69 | 82.90 | 75.88 |
|  |  | Audit | 73.07 | 83.89 | 86.19 | 88.30 | 88.26 | 87.45 | 14.57 |  |  |  |  |  |  |  |
|  | Prec@N | Attack | 65.92 | 87.75 | 93.72 | 95.71 | 96.00 | 95.73 | 94.72 | 99.39 | 55.09 | 99.88 | 97.70 | 98.88 | 99.40 | 81.57 |
|  |  | Audit | 72.61 | 98.47 | 91.84 | 93.36 | 93.59 | 94.55 | 51.14 |  |  |  |  |  |  |  |
|  | Rec@N | Attack | 78.78 | 69.62 | 71.92 | 75.90 | 75.88 | 75.74 | 25.82 | 61.48 | 93.34 | 33.10 | 67.18 | 70.34 | 59.24 | 57.80 |
|  |  | Audit | 76.26 | 62.94 | 76.98 | 80.76 | 80.36 | 77.00 | 96.14 |  |  |  |  |  |  |  |
|  | F1@N | Attack | 71.78 | 77.64 | 81.39 | 84.66 | 84.76 | 84.57 | 40.58 | 75.97 | 69.29 | 49.72 | 79.62 | 82.20 | 74.24 | 67.66 |
|  |  | Audit | 74.39 | 76.79 | 83.76 | 86.61 | 86.47 | 84.88 | 66.77 |  |  |  |  |  |  |  |
|  | AUC | Audit | 0.797 | 0.895 | 0.871 | 0.892 | 0.891 | 0.896 | 0.626 | – | – | – | – | – | – | – |
|  | TPR@0.1%FPR | Audit | 0.48 | 14.0 | 8.1 | 9.2 | 8.9 | 0.48 | 0.0 | – | – | – | – | – | – | – |
| 110th | Acc | Attack | 66.84 | 80.08 | 78.08 | 81.15 | 81.54 | 81.40 | 61.98 | 70.54 | 50.35 | 65.60 | 76.54 | 81.62 | 75.60 | 70.18 |
|  |  | Audit | 70.94 | 80.03 | 80.14 | 83.29 | 83.43 | 81.54 | 53.09 |  |  |  |  |  |  |  |
|  | Prec@P | Attack | 69.61 | 74.74 | 73.49 | 76.02 | 75.98 | 74.72 | 57.03 | 63.05 | 71.08 | 59.28 | 76.19 | 75.49 | 68.45 | 67.68 |
|  |  | Audit | 73.16 | 71.98 | 77.06 | 79.64 | 79.51 | 75.10 | 66.56 |  |  |  |  |  |  |  |
|  | Rec@P | Attack | 59.78 | 90.86 | 87.76 | 91.00 | 92.24 | 94.90 | 97.24 | 99.22 | 1.18 | 99.66 | 77.20 | 93.64 | 94.98 | 77.26 |
|  |  | Audit | 66.14 | 98.34 | 85.84 | 89.44 | 90.08 | 94.38 | 12.42 |  |  |  |  |  |  |  |
|  | F1@P | Attack | 64.32 | 82.02 | 79.99 | 82.84 | 83.32 | 83.61 | 71.89 | 77.11 | 2.32 | 74.34 | 76.69 | 83.59 | 79.56 | 72.15 |
|  |  | Audit | 69.47 | 83.12 | 81.21 | 84.26 | 84.46 | 83.64 | 20.93 |  |  |  |  |  |  |  |
|  | Prec@N | Attack | 64.76 | 88.35 | 84.81 | 88.79 | 90.13 | 93.01 | 90.64 | 98.17 | 50.18 | 98.93 | 76.90 | 91.63 | 91.80 | 73.51 |
|  |  | Audit | 69.11 | 97.38 | 84.02 | 87.96 | 88.56 | 92.44 | 51.70 |  |  |  |  |  |  |  |
|  | Rec@N | Attack | 73.90 | 69.30 | 71.30 | 70.84 | 67.90 |  | 26.72 | 41.86 | 99.52 | 31.54 | 75.88 | 69.60 | 56.22 | 63.10 |
|  |  | Audit | 75.74 | 61.72 | 74.44 | 77.14 | 76.78 | 68.70 | 93.76 |  |  |  |  |  |  |  |
|  | F1@N | Attack | 69.03 | 77.67 | 75.69 | 79.09 | 79.33 | 78.50 | 41.27 | 58.69 | 66.72 | 47.83 | 76.38 | 79.11 | 69.73 | 67.91 |
|  |  | Audit | 72.27 | 75.55 | 78.94 | 82.19 | 82.25 | 78.82 | 66.65 |  |  |  |  |  |  |  |
|  | AUC | Audit | 0.764 | 0.890 | 0.835 | 0.866 | 0.866 | 0.862 | 0.625 | – | – | – | – | – | – | – |
|  | TPR@0.1%FPR | Audit | 1.5 | 8.9 | 8.7 | 9.4 | 9.5 | 0.2 | 0.0 | – | – | – | – | – | – | – |
| 10th | Acc | Attack | 51.65 | 53.36 | 50.35 | 52.73 | 53.00 | 52.30 | 52.41 | 50.00 | 50.41 | 50.11 | 50.47 | 50.20 | 51.59 | 50.07 |
|  |  | Audit | 51.12 | 53.87 | 57.48 | 58.74 | 58.92 | 53.19 | 53.47 |  |  |  |  |  |  |  |
|  | Prec@P | Attack | 52.84 | 54.13 | 50.37 | 52.27 | 52.46 | 51.50 | 51.44 | 50.00 | 56.77 | 50.06 | 52.13 | 50.66 | 51.12 | 50.08 |
|  |  | Audit | 52.91 | 53.81 | 57.70 | 59.30 | 59.31 | 52.53 | 55.52 |  |  |  |  |  |  |  |
|  | Rec@P | Attack | 30.68 | 44.06 | 47.10 | 62.74 | 64.06 | 78.98 | 85.82 | 100.00 | 3.44 | 97.06 | 11.52 | 15.40 | 72.86 | 42.04 |
|  |  | Audit | 20.34 | 54.60 | 56.08 | 55.74 | 56.80 | 66.26 | 34.90 |  |  |  |  |  |  |  |
|  | F1@P | Attack | 38.82 | 48.58 | 48.68 | 57.03 | 57.68 | 62.35 | 64.33 | 66.67 | 6.49 | 66.05 | 18.87 | 23.62 | 60.08 | 45.71 |
|  |  | Audit | 29.38 | 54.20 | 56.88 | 57.46 | 58.03 | 58.60 | 42.86 |  |  |  |  |  |  |  |
|  | Prec@N | Attack | 51.16 | 52.83 | 50.33 | 53.41 | 53.85 | 54.93 | 57.26 | 0.00 | 50.21 | 51.80 | 50.26 | 50.12 | 52.77 | 50.06 |
|  |  | Audit | 50.69 | 53.93 | 57.28 | 58.25 | 58.56 | 54.32 | 52.53 |  |  |  |  |  |  |  |
|  | Rec@N | Attack | 72.62 | 62.66 | 53.60 | 42.72 | 41.94 | 25.62 | 19.00 | 0.00 | 97.38 | 3.16 | 89.42 | 85.00 | 30.32 | 58.10 |
|  |  | Audit | 81.90 | 53.14 | 58.88 | 61.74 | 61.04 | 40.12 | 72.04 |  |  |  |  |  |  |  |
|  | F1@N | Attack | 60.03 | 57.33 | 51.91 | 47.47 | 47.16 | 34.94 | 28.53 | 0.00 | 66.26 | 5.96 | 64.35 | 63.06 | 38.51 | 53.78 |
|  |  | Audit | 62.62 | 53.53 | 58.07 | 59.94 | 59.77 | 46.15 | 60.76 |  |  |  |  |  |  |  |
|  | AUC | Audit | 0.511 | 0.547 | 0.518 | 0.548 | 0.550 | 0.542 | 0.526 | – | – | – | – | – | – | – |
|  | TPR@0.1%FPR | Audit | 0.18 | 0.18 | 1.8 | 2.6 | 2.6 | 0.22 | 0.00 | – | – | – | – | – | – | – |

The Metric MIA columns comprise LiRA, RMIA, Entropy, Modified Entropy, Confidence, Quantile, and Merlin. The ML-Leaks columns comprise ML-Leak1, ML-Leak3, and Shadow. The BlindMI columns comprise One-class, Diff-w, Diff-single, and Diff-bi.

## G.5 Performance of MIA Methods under Meta Class Scenario

## G.6 Performance of MIA Methods under Different Algorithms

Table 18: The performance of different MIA methods for superclass CIFAR-100 under ResNet-18.

| Epoch | Metric | Mode | Metric MIA | | | | | | | ML-Leaks | | | BlindMI | | | |
|---|---|---|---|---|---|---|---|---|---|---|---|---|---|---|---|---|
| | | | LiRA | RMIA | Entropy | Modified Entropy | Confidence | Quantile | Merlin | MLLeak1 | MLLeak3 | Shadow | One-class | Diff-w | Diff-single | Diff-bi |
| 200th | Acc | Attack | 58.27 | 65.34 | 71.20 | 72.31 | 72.28 | 71.96 | 55.71 | 66.09 | 68.17 | 59.12 | 70.91 | 69.13 | 67.56 | 55.56 |
| | | Audit | 62.02 | 66.09 | 73.66 | 74.67 | 74.65 | 72.51 | 54.02 | | | | | | | |
| | Prec@P | Attack | 62.24 | 65.65 | 64.67 | 65.00 | 65.05 | 64.50 | 53.22 | 59.64 | 67.18 | 55.02 | 63.88 | 61.89 | 60.76 | 53.98 |
| | | Audit | 64.01 | 59.62 | 67.70 | 68.44 | 68.29 | 65.90 | 64.00 | | | | | | | |
| | Rec@P | Attack | 42.06 | 64.36 | 94.46 | 96.68 | 96.32 | 97.66 | 94.38 | 99.56 | 71.06 | 100.00 | 96.24 | 99.58 | 99.16 | 75.42 |
| | | Audit | 55.51 | 99.74 | 90.48 | 91.56 | 92.02 | 93.28 | 18.38 | | | | | | | |
| | F1@P | Attack | 50.20 | 65.00 | 76.63 | 77.74 | 77.65 | 77.69 | 68.06 | 74.59 | 69.06 | 70.98 | 76.79 | 76.34 | 75.35 | 62.92 |
| | | Audit | 5922 | 74.63 | 77.45 | 78.33 | 78.40 | 77.24 | 28.56 | | | | | | | |
| | Prec@N | Attack | 56.25 | 65.05 | 89.64 | 93.52 | 92.91 | 95.19 | 75.20 | 98.67 | 69.28 | 100.00 | 92.38 | 98.93 | 97.72 | 59.22 |
| | | Audit | 60.59 | 99.20 | 85.65 | 87.25 | 87.77 | 88.50 | 52.35 | | | | | | | |
| | Rec@N | Attack | 74.48 | 66.32 | 47.94 | 47.94 | 48.24 | 46.26 | 17.04 | 32.62 | 65.28 | 18.24 | 45.58 | 38.68 | 35.96 | 35.70 |
| | | Audit | 69.02 | 32.44 | 56.84 | 57.78 | 57.28 | 51.74 | 89.66 | | | | | | | |
| | F1@N | Attack | 64.09 | 65.68 | 62.47 | 63.39 | 63.51 | 62.26 | 27.78 | 49.03 | 65.22 | 30.85 | 61.04 | 55.61 | 52.57 | 44.55 |
| | | Audit | 64.53 | 48.89 | 68.33 | 69.52 | 69.32 | 65.30 | 66.10 | | | | | | | |
| | AUC | Audit | 0.637 | 0.733 | 0.741 | 0.755 | 0.747 | 0.750 | 0.565 | – | – | – | – | – | – | – |
| | TPR@0.1%FPR | Audit | 0.4 | 1.0 | 0.4 | 0.5 | 0.3 | 0.1 | 0.0 | – | – | – | – | – | – | – |
| 110th | Acc | Attack | 56.00 | 65.33 | 61.62 | 63.77 | 63.67 | 66.07 | 54.89 | 50.00 | 54.32 | 56.70 | 63.64 | 66.17 | 62.84 | 58.94 |
| | | Audit | 58.98 | 65.29 | 66.58 | 68.77 | 68.83 | 66.17 | 54.39 | | | | | | | |
| | Prec@P | Attack | 58.75 | 65.53 | 60.75 | 62.28 | 62.28 | 60.46 | 52.73 | 50.00 | 59.35 | 53.60 | 60.10 | 60.90 | 58.55 | 57.50 |
| | | Audit | 60.36 | 66.04 | 62.35 | 63.23 | 63.32 | 60.67 | 58.98 | | | | | | | |
| | Rec@P | Attack | 40.28 | 64.68 | 65.68 | 69.84 | 69.32 | 92.90 | 94.48 | 100.00 | 27.42 | 99.80 | 81.14 | 90.32 | 87.92 | 68.56 |
| | | Audit | 52.32 | 62.94 | 83.6 | 89.72 | 89.52 | 91.96 | 28.84 | | | | | | | |
| | F1@P | Attack | 47.79 | 65.10 | 63.12 | 65.84 | 65.61 | 73.25 | 67.68 | 66.67 | 37.51 | 69.74 | 69.06 | 72.75 | 70.29 | 62.54 |
| | | Audit | 56.05 | 64.45 | 71.46 | 74.18 | 74.17 | 73.11 | 38.74 | | | | | | | |
| | Prec@N | Attack | 54.56 | 65.13 | 62.65 | 65.67 | 65.41 | 84.68 | 73.49 | 0.00 | 52.81 | 98.55 | 70.98 | 81.28 | 75.76 | 61.07 |
| | | Audit | 57.92 | 64.60 | 75.20 | 82.31 | 82.12 | 83.40 | 52.91 | | | | | | | |
| | Rec@N | Attack | 71.72 | 65.98 | 57.56 | 57.70 | 58.02 | 39.24 | 15.30 | 0.00 | 81.22 | 13.60 | 46.14 | 42.02 | 37.76 | 49.32 |
| | | Audit | 65.64 | 67.64 | 49.48 | 47.82 | 48.14 | 40.38 | 79.94 | | | | | | | |
| | F1@N | Attack | 61.98 | 65.55 | 60.00 | 61.43 | 61.49 | 53.63 | 25.33 | 0.00 | 64.00 | 23.90 | 55.93 | 55.40 | 50.40 | 54.57 |
| | | Audit | 61.54 | 66.09 | 59.69 | 60.49 | 60.70 | 54.41 | 63.67 | | | | | | | |
| | AUC | Audit | 0.613 | 0.733 | 0.661 | 0.685 | 0.683 | 0.681 | 0.555 | – | – | – | – | – | – | – |
| | TPR@0.1%FPR | Audit | 0.4 | 1.7 | 0.4 | 0.4 | 0.3 | 0.1 | 0.0 | – | – | – | – | – | – | – |
| 10th | Acc | Attack | 51.52 | 52.65 | 50.16 | 50.06 | 50.25 | 51.97 | 50.76 | 50.00 | 50.05 | 50.00 | 50.24 | 50.19 | 50.18 | 50.07 |
| | | Audit | 52.01 | 52.75 | 50.75 | 56.90 | 56.74 | 52.31 | 53.98 | | | | | | | |
| | Prec@P | Attack | 52.92 | 52.56 | 53.31 | 51.06 | 54.79 | 51.22 | 50.54 | 50.00 | 50.21 | 50.00 | 50.67 | 50.68 | 50.15 | 50.09 |
| | | Audit | 53.10 | 52.63 | 56.02 | 57.26 | 57.67 | 52.08 | 56.72 | | | | | | | |
| | Rec@P | Attack | 27.52 | 54.50 | 2.58 | 2.88 | 2.86 | 82.78 | 70.64 | 100.00 | 11.68 | 99.96 | 18.28 | 14.16 | 60.98 | 39.98 |
| | | Audit | 34.46 | 55.00 | 52.80 | 54.40 | 50.68 | 57.94 | 33.58 | | | | | | | |
| | F1@P | Attack | 36.21 | 53.51 | 4.92 | 5.45 | 5.44 | 63.28 | 58.93 | 66.67 | 18.95 | 66.66 | 26.87 | 22.14 | 55.04 | 44.47 |
| | | Audit | 41.80 | 53.79 | 54.36 | 55.79 | 53.95 | 54.85 | 42.19 | | | | | | | |
| | Prec@N | Attack | 51.03 | 52.75 | 50.08 | 50.03 | 50.13 | 55.13 | 51.26 | 0.00 | 50.03 | 50.00 | 50.15 | 50.11 | 50.23 | 50.06 |
| | | Audit | 51.49 | 52.88 | 55.36 | 56.57 | 56.01 | 52.60 | 52.83 | | | | | | | |
| | Rec@N | Attack | 75.52 | 50.80 | 97.74 | 97.24 | 97.64 | 21.16 | 30.88 | 0.00 | 88.42 | 0.04 | 82.20 | 86.22 | 39.38 | 60.16 |
| | | Audit | 69.56 | 50.50 | 58.54 | 59.40 | 62.80 | 46.68 | 74.38 | | | | | | | |
| | F1@N | Attack | 60.90 | 51.76 | 66.23 | 66.07 | 66.25 | 30.58 | 38.54 | 0.00 | 63.90 | 0.08 | 62.29 | 63.38 | 44.15 | 54.65 |
| | | Audit | 59.17 | 51.66 | 66.91 | 57.95 | 59.21 | 49.46 | 61.78 | | | | | | | |
| | AUC | Audit | 0.520 | 0.534 | 0.510 | 0.533 | 0.533 | 0.532 | 0.513 | – | – | – | – | – | – | – |
| | TPR@0.1%FPR | Audit | 0.1 | 1.1 | 0.4 | 0.4 | 0.4 | 0.2 | 0.0 | – | – | – | – | – | – | – |

Table 19: The performance of different MIA methods for CIFAR-100 under ResNet-50 **with pretrain**. The model is pretrained on ImageNet-1k.

| Epoch | Metric | Mode | | | Metric MIA | | | | | ML-Leaks | | | BlindMI | | | |
|---|---|---|---|---|---|---|---|---|---|---|---|---|---|---|---|---|
| | | | LiRA | RMIA | Entropy | Modified Entropy | Confidence | Quantile | Merlin | MLLeak1 | MLLeak3 | Shadow | One-class | Diff-w | Diff-single | Diff-bi |
| 200th | Acc | Attack | 50.60 | 59.76 | 71.73 | 72.55 | 72.55 | 71.86 | 52.28 | 58.83 | 51.24 | 56.12 | 70.71 | 64.27 | 65.24 | 53.59 |
| | | Audit | 50.36 | 59.87 | 74.66 | 75.12 | 75.10 | 72.03 | 56.38 | | | | | | | |
| | Prec@P | Attack | 51.27 | 57.77 | 64.77 | 65.43 | 65.38 | 64.37 | 51.19 | 54.84 | 50.63 | 53.26 | 63.24 | 58.32 | 59.00 | 52.48 |
| | | Audit | 50.36 | 58.08 | 70.15 | 70.45 | 70.28 | 66.17 | 57.50 | | | | | | | |
| | Rec@P | Attack | 24.22 | 72.56 | 95.30 | 95.60 | 95.88 | 97.94 | 97.90 | 99.98 | 100.00 | 100.00 | 98.94 | 99.98 | 99.90 | 71.74 |
| | | Audit | 75.46 | 70.92 | 85.84 | 86.54 | 86.98 | 90.14 | 48.92 | | | | | | | |
| | F1@P | Attack | 32.90 | 64.33 | 77.12 | 77.69 | 77.74 | 77.68 | 67.23 | 70.83 | 67.22 | 69.50 | 77.16 | 73.67 | 74.19 | 60.62 |
| | | Audit | 60.41 | 63.86 | 77.21 | 77.67 | 77.74 | 76.32 | 52.86 | | | | | | | |
| | Prec@N | Attack | 50.39 | 63.12 | 91.11 | 91.84 | 92.28 | 95.69 | 76.03 | 70.83 | 100.00 | 100.00 | 97.57 | 99.93 | 99.67 | 55.36 |
| | | Audit | 51.08 | 62.67 | 81.76 | 82.56 | 82.92 | 84.54 | 55.55 | | | | | | | |
| | Rec@N | Attack | 76.98 | 46.96 | 48.16 | 49.50 | 49.22 | 45.78 | 6.66 | 17.68 | 2.48 | 12.24 | 42.48 | 28.56 | 30.58 | 35.04 |
| | | Audit | 25.62 | 48.82 | 63.48 | 63.70 | 63.22 | 53.92 | 63.84 | | | | | | | |
| | F1@N | Attack | 60.91 | 53.85 | 63.01 | 64.33 | 64.20 | 61.93 | 12.25 | 30.04 | 4.84 | 21.81 | 59.19 | 44.42 | 46.80 | 42.91 |
| | | Audit | 34.12 | 54.88 | 71.47 | 71.91 | 71.74 | 65.84 | 59.41 | | | | | | | |
| | AUC | Audit | 0.495 | 0.658 | 0.761 | 0.753 | 0.765 | 0.765 | 0.502 | – | – | – | – | – | – | – |
| | TPR@0.1%FPR | Audit | 0.0 | 0.0 | 6.0 | 5.8 | 5.9 | 0.0 | 1.8 | – | – | – | – | – | – | – |
| 110th | Acc | Attack | 50.56 | 60.54 | 70.07 | 70.91 | 70.77 | 71.39 | 52.40 | 55.47 | 52.09 | 56.26 | 70.75 | 65.27 | 65.87 | 53.35 |
| | | Audit | 50.42 | 60.76 | 73.03 | 73.76 | 73.72 | 70.88 | 56.49 | | | | | | | |
| | Prec@P | Attack | 50.72 | 59.42 | 63.99 | 64.58 | 64.37 | 64.80 | 51.26 | 52.90 | 51.07 | 53.24 | 63.98 | 59.02 | 59.49 | 52.44 |
| | | Audit | 50.28 | 59.35 | 68.92 | 69.23 | 69.27 | 64.55 | 57.77 | | | | | | | |
| | Rec@P | Attack | 39.28 | 66.46 | 91.80 | 92.60 | 93.06 | 93.64 | 97.92 | 99.84 | 100.00 | 100.00 | 94.96 | 99.96 | 99.46 | 71.90 |
| | | Audit | 75.02 | 68.28 | 83.90 | 85.54 | 85.26 | 92.64 | 48.24 | | | | | | | |
| | F1@P | Attack | 44.27 | 62.75 | 75.41 | 76.09 | 76.10 | 76.60 | 67.29 | 69.16 | 67.61 | 69.57 | 76.45 | 74.21 | 74.45 | 60.65 |
| | | Audit | 60.21 | 63.50 | 75.67 | 76.53 | 76.44 | 76.08 | 52.58 | | | | | | | |
| | Prec@N | Attack | 50.46 | 61.96 | 85.50 | 86.93 | 87.48 | 88.54 | 76.79 | 98.58 | 100.00 | 100.00 | 90.23 | 99.87 | 98.35 | 55.33 |
| | | Audit | 50.83 | 62.66 | 79.43 | 81.08 | 80.84 | 86.97 | 55.57 | | | | | | | |
| | Rec@N | Attack | 61.84 | 54.62 | 48.34 | 49.22 | 48.48 | 49.14 | 6.88 | 11.10 | 4.18 | 12.52 | 46.54 | 30.58 | 32.28 | 34.80 |
| | | Audit | 25.82 | 53.24 | 62.16 | 61.98 | 62.18 | 49.12 | 64.74 | | | | | | | |
| | F1@N | Attack | 55.57 | 58.06 | 61.76 | 62.85 | 62.39 | 63.20 | 12.63 | 19.95 | 8.02 | 22.25 | 61.41 | 46.82 | 48.61 | 42.73 |
| | | Audit | 34.24 | 57.57 | 69.71 | 70.26 | 70.29 | 62.78 | 59.81 | | | | | | | |
| | AUC | Audit | 0.496 | 0.667 | 0.747 | 0.747 | 0.747 | 0.756 | 0.508 | – | – | – | – | – | – | – |
| | TPR@0.1%FPR | Audit | 0.0 | 0.0 | 6.3 | 6.3 | 6.2 | 1.7 | 2.3 | – | – | – | – | – | – | – |
| 10th | Acc | Attack | 50.16 | 59.92 | 60.72 | 62.86 | 62.86 | 63.05 | 52.49 | 50.00 | 50.17 | 54.04 | 53.26 | 62.69 | 61.18 | 56.44 |
| | | Audit | 50.18 | 60.32 | 65.40 | 67.32 | 67.20 | 63.39 | 56.70 | | | | | | | |
| | Prec@P | Attack | 50.16 | 58.99 | 57.70 | 58.56 | 58.71 | 58.23 | 51.41 | 50.00 | 50.09 | 52.11 | 51.69 | 57.82 | 57.21 | 55.00 |
| | | Audit | 50.11 | 57.16 | 62.69 | 63.42 | 63.56 | 58.66 | 57.68 | | | | | | | |
| | Rec@P | Attack | 49.14 | 65.08 | 80.36 | 87.94 | 86.70 | 92.32 | 91.08 | 100.00 | 100.00 | 99.58 | 99.58 | 93.84 | 88.70 | 70.88 |
| | | Audit | 81.74 | 82.38 | 76.10 | 81.42 | 80.62 | 90.74 | 50.34 | | | | | | | |
| | F1@P | Attack | 49.65 | 61.89 | 67.17 | 70.31 | 70.01 | 71.42 | 65.72 | 66.67 | 66.74 | 68.42 | 68.06 | 71.55 | 69.59 | 61.94 |
| | | Audit | 62.13 | 67.49 | 68.74 | 71.30 | 71.08 | 71.25 | 53.76 | | | | | | | |
| | Prec@N | Attack | 50.16 | 61.06 | 67.65 | 75.80 | 74.58 | 81.48 | 60.91 | 0.00 | 100.00 | 95.29 | 94.29 | 83.66 | 74.87 | 59.06 |
| | | Audit | 50.49 | 68.47 | 69.59 | 74.06 | 73.51 | 79.56 | 55.94 | | | | | | | |
| | Rec@N | Attack | 51.18 | 54.62 | 41.08 | 37.78 | 39.02 | 33.78 | 13.90 | 0.00 | 0.34 | 8.50 | 6.94 | 31.54 | 33.66 | 42.00 |
| | | Audit | 18.62 | 38.26 | 54.70 | 53.04 | 53.78 | 36.04 | 63.06 | | | | | | | |
| | F1@N | Attack | 50.66 | 57.74 | 51.12 | 50.43 | 51.23 | 47.76 | 22.63 | 0.00 | 0.68 | 15.61 | 12.93 | 45.81 | 46.44 | 49.09 |
| | | Audit | 27.21 | 49.09 | 61.25 | 61.81 | 62.12 | 49.61 | 59.29 | | | | | | | |
| | AUC | Audit | 0.493 | 0.651 | 0.651 | 0.666 | 0.669 | 0.656 | 0.514 | – | – | – | – | – | – | – |
| | TPR@0.1%FPR | Audit | 0.5 | 0.0 | 0.0 | 0.0 | 0.0 | 0.0 | 0.0 | – | – | – | – | – | – | – |

Table 20: The performance of different MIA methods for CIFAR-100 under ResNet-18 trained with DP-SGD.

| Epoch | Metric | Mode | LiRA | RMIA | Entropy | Modified Entropy | Confidence | Quantile | Merlin | MLLeak1 | MLLeak3 | Shadow | One-class | Diff-w | Diff-single | Diff-bi |
|---|---|---|---|---|---|---|---|---|---|---|---|---|---|---|---|---|
| 200th | Acc | Attack | 50.55 | 63.98 | 59.93 | 64.58 | 64.60 | 65.93 | 58.69 | 57.13 | 60.90 | 60.98 | 60.33 | 65.27 | 58.01 | 53.88 |
| | | Audit | 50.54 | 64.74 | 64.32 | 68.11 | 68.07 | 66.52 | 56.18 | | | | | | | |
| | Prec@P | Attack | 52.73 | 58.75 | 56.98 | 60.12 | 60.17 | 60.95 | 55.44 | 54.13 | 58.58 | 56.45 | 58.38 | 59.92 | 54.75 | 52.71 |
| | | Audit | 52.68 | 59.18 | 62.15 | 64.23 | 64.16 | 61.77 | 57.11 | | | | | | | |
| | Rec@P | Attack | 10.64 | 93.86 | 81.10 | 86.60 | 86.38 | 88.68 | 88.50 | 93.54 | 74.42 | 96.12 | 72.00 | 92.24 | 92.32 | 75.40 |
| | | Audit | 10.62 | 95.04 | 73.26 | 81.76 | 81.86 | 86.68 | 49.62 | | | | | | | |
| | F1@P | Attack | 17.71 | 72.27 | 66.93 | 70.97 | 70.93 | 72.24 | 68.18 | 68.57 | 65.56 | 71.13 | 64.48 | 72.65 | 68.74 | 62.05 |
| | | Audit | 17.68 | 72.94 | 67.25 | 71.94 | 71.94 | 72.14 | 53.10 | | | | | | | |
| | Prec@N | Attack | 50.31 | 84.74 | 67.22 | 76.05 | 75.87 | 79.23 | 71.52 | 76.23 | 64.94 | 86.94 | 63.48 | 83.15 | 75.53 | 56.81 |
| | | Audit | 50.30 | 87.41 | 67.44 | 74.91 | 74.95 | 77.68 | 55.46 | | | | | | | |
| | Rec@N | Attack | 90.46 | 34.10 | 38.76 | 42.56 | 42.82 | 43.18 | 28.88 | 20.72 | 47.38 | 25.84 | 48.66 | 38.30 | 23.70 | 32.36 |
| | | Audit | 90.46 | 34.44 | 55.38 | 54.46 | 54.28 | 46.36 | 62.74 | | | | | | | |
| | F1@N | Attack | 64.66 | 48.63 | 49.17 | 54.58 | 54.74 | 55.90 | 41.15 | 32.58 | 54.79 | 39.84 | 55.09 | 52.44 | 36.08 | 41.23 |
| | | Audit | 64.65 | 49.41 | 60.82 | 63.07 | 62.96 | 58.07 | 58.88 | | | | | | | |
| | AUC | Audit | 0.350 | 0.721 | 0.624 | 0.667 | 0.667 | 0.688 | 0.519 | – | – | – | – | – | – | – |
| | TPR@0.1%FPR | Audit | 0.1 | 6.9 | 2.2 | 2.8 | 2.8 | 0.2 | 1.8 | – | – | – | – | – | – | – |
| 110th | Acc | Attack | 51.17 | 58.16 | 54.92 | 58.20 | 58.02 | 58.92 | 54.99 | 50.00 | 54.94 | 52.60 | 54.44 | 58.52 | 54.10 | 52.95 |
| | | Audit | 51.26 | 59.40 | 59.67 | 62.55 | 62.50 | 59.26 | 56.54 | | | | | | | |
| | Prec@P | Attack | 54.62 | 55.61 | 54.43 | 56.21 | 56.06 | 56.21 | 53.07 | 50.00 | 55.10 | 51.39 | 54.86 | 56.53 | 52.66 | 52.17 |
| | | Audit | 56.30 | 56.68 | 60.17 | 61.55 | 61.49 | 56.69 | 57.10 | | | | | | | |
| | Rec@P | Attack | 13.82 | 80.88 | 60.48 | 74.22 | 74.16 | 80.70 | 86.16 | 100.00 | 53.36 | 96.28 | 50.14 | 73.80 | 81.22 | 70.78 |
| | | Audit | 11.26 | 79.72 | 57.20 | 66.88 | 66.88 | 78.44 | 52.60 | | | | | | | |
| | F1@P | Attack | 22.06 | 65.91 | 57.29 | 63.97 | 63.85 | 66.27 | 65.69 | 66.67 | 54.22 | 67.01 | 52.39 | 64.02 | 63.89 | 60.07 |
| | | Audit | 18.77 | 66.26 | 58.65 | 64.10 | 64.07 | 65.82 | 54.76 | | | | | | | |
| | Prec@N | Attack | 50.67 | 64.96 | 55.54 | 62.07 | 61.84 | 65.80 | 63.25 | 0.00 | 54.79 | 70.57 | 54.09 | 62.27 | 58.96 | 54.59 |
| | | Audit | 50.70 | 65.84 | 59.21 | 63.74 | 63.70 | 65.02 | 56.06 | | | | | | | |
| | Rec@N | Attack | 88.52 | 35.44 | 49.36 | 42.18 | 41.88 | 37.14 | 23.82 | 0.00 | 56.52 | 8.92 | 58.74 | 43.24 | 26.98 | 35.12 |
| | | Audit | 91.26 | 39.08 | 62.14 | 58.22 | 58.12 | 40.08 | 60.48 | | | | | | | |
| | F1@N | Attack | 64.45 | 45.86 | 52.27 | 50.23 | 49.94 | 47.48 | 34.61 | 0.00 | 55.64 | 15.84 | 56.32 | 51.04 | 37.02 | 42.74 |
| | | Audit | 65.19 | 49.05 | 60.64 | 60.86 | 60.78 | 49.59 | 58.19 | | | | | | | |
| | AUC | Audit | 0.446 | 0.639 | 0.562 | 0.606 | 0.606 | 0.609 | 0.522 | – | – | – | – | – | – | – |
| | TPR@0.1%FPR | Audit | 0.08 | 2.8 | 2.1 | 2.9 | 2.9 | 0.6 | 1.6 | – | – | – | – | – | – | – |
| 10th | Acc | Attack | 49.69 | 51.97 | 50.58 | 51.78 | 51.94 | 51.87 | 50.89 | 50.00 | 51.25 | 50.05 | 51.25 | 50.64 | 50.45 | 50.78 |
| | | Audit | 50.17 | 52.45 | 57.36 | 57.94 | 57.92 | 52.19 | 55.91 | | | | | | | |
| | Prec@P | Attack | 44.56 | 52.39 | 50.59 | 51.57 | 51.75 | 51.38 | 50.65 | 50.00 | 52.31 | 50.03 | 52.81 | 54.49 | 50.33 | 51.00 |
| | | Audit | 50.10 | 52.75 | 58.17 | 58.74 | 58.34 | 53.37 | 56.72 | | | | | | | |
| | Rec@P | Attack | 2.54 | 43.26 | 50.00 | 58.60 | 57.38 | 69.76 | 69.04 | 100.00 | 28.34 | 93.68 | 23.50 | 7.76 | 67.80 | 39.80 |
| | | Audit | 86.14 | 47.02 | 52.42 | 53.34 | 55.38 | 34.66 | 49.90 | | | | | | | |
| | F1@P | Attack | 4.81 | 47.39 | 50.29 | 54.86 | 54.42 | 59.17 | 58.43 | 66.67 | 36.76 | 65.22 | 32.53 | 13.59 | 57.78 | 44.71 |
| | | Audit | 63.35 | 49.72 | 55.14 | 55.91 | 56.82 | 42.03 | 53.09 | | | | | | | |
| | Prec@N | Attack | 49.84 | 51.68 | 50.57 | 52.06 | 52.18 | 52.91 | 51.40 | 0.00 | 50.86 | 50.39 | 50.80 | 50.34 | 50.69 | 50.64 |
| | | Audit | 50.61 | 52.21 | 56.70 | 57.27 | 57.54 | 51.62 | 55.28 | | | | | | | |
| | Rec@N | Attack | 96.84 | 60.68 | 51.16 | 44.96 | 46.50 | 33.98 | 32.74 | 0.00 | 74.16 | 6.42 | 79.00 | 93.52 | 33.10 | 61.76 |
| | | Audit | 14.20 | 57.88 | 62.30 | 62.54 | 60.46 | 69.72 | 61.92 | | | | | | | |
| | F1@N | Attack | 65.81 | 55.82 | 50.86 | 48.25 | 49.18 | 41.38 | 40.00 | 0.00 | 60.34 | 11.39 | 61.84 | 65.45 | 40.05 | 55.65 |
| | | Audit | 22.18 | 54.90 | 59.37 | 59.79 | 58.96 | 59.32 | 58.41 | | | | | | | |
| | AUC | Audit | 0.487 | 0.531 | 0.513 | 0.529 | 0.530 | 0.528 | 0.501 | – | – | – | – | – | – | – |
| | TPR@0.1%FPR | Audit | 0.0 | 1.5 | 2.3 | 2.3 | 2.8 | 1.0 | 1.6 | – | – | – | – | – | – | – |

## G.7 PERFORMANCE OF MIA METHODS UNDER UNLEARNING SCENARIO

## G.8 PERFORMANCE OF MIA METHODS WITH DIFFERENT NUMBER SHADOW MODELS

Table 21: The performance of different MIA methods for CIFAR-100 under ResNet-18 across pretrain, retrain, SalUn and SFR-on (**full class unlearn setting**). The performance of each MIA method is evaluated on **retain data** (members) and **forget data** (non-members).

| Model | Metric | | | | | | | | | | | | | | |
|---|---|---|---|---|---|---|---|---|---|---|---|---|---|---|---|
| | | Metric MIA | | | | | | | ML-Leaks | | | BlindMI | | | |
| | | LiRA | RMIA | Entropy | Modified Entropy | Confidence | Quantile | Merlin | MLLeak1 | MLLeak3 | Shadow | One-class | Diff-w | Diff-single | Diff-bi |
| Pre-train | Accuracy | 64.10 | 61.30 | 61.40 | 61.40 | 61.50 | 63.90 | 57.30 | 50.00 | 47.20 | 50.00 | 50.00 | 50.00 | 47.00 | 47.80 |
| | Precision@P | 58.36 | 59.31 | 84.34 | 85.62 | 85.71 | 58.19 | 71.60 | 50.00 | 48.12 | 50.00 | 50.00 | 50.00 | 48.12 | 48.29 |
| | Recall@P | 98.40 | 72.00 | 28.00 | 27.40 | 27.60 | 98.80 | 24.20 | 100.00 | 71.60 | 100.00 | 100.00 | 100.00 | 76.80 | 62.00 |
| | F1@P | 73.27 | 65.04 | 42.04 | 41.52 | 41.75 | 73.24 | 36.17 | 66.67 | 57.56 | 66.67 | 66.67 | 66.67 | 59.17 | 54.29 |
| | Precision@N | 94.90 | 64.38 | 56.83 | 56.79 | 56.85 | 96.03 | 54.39 | 0.00 | 44.53 | 0.00 | 0.00 | 0.00 | 42.57 | 46.93 |
| | Recall@N | 29.80 | 50.60 | 94.80 | 95.40 | 95.40 | 29.00 | 90.40 | 0.00 | 22.80 | 0.00 | 0.00 | 0.00 | 17.20 | 33.60 |
| | F1@N | 45.36 | 56.66 | 71.06 | 71.19 | 71.25 | 44.55 | 67.92 | 0.00 | 30.16 | 0.00 | 0.00 | 0.00 | 24.50 | 39.16 |
| | AUC | 0.64 | 0.61 | 0.61 | 0.61 | 0.61 | 0.64 | 0.57 | 0.50 | 0.47 | 0.50 | 0.50 | 0.50 | 0.47 | 0.48 |
| | TPR@0.1FPR | 0.00 | 0.00 | 0.00 | 0.00 | 0.00 | 0.00 | 0.00 | 0.00 | 0.00 | 0.00 | 0.00 | 0.00 | 0.00 | 0.00 |
| Retrain | Accuracy | 63.50 | 63.90 | 60.90 | 61.10 | 61.00 | 63.50 | 57.70 | 83.60 | 83.40 | 90.00 | 86.00 | 96.20 | 81.00 | 69.00 |
| | Precision@P | 70.15 | 74.05 | 84.28 | 84.05 | 84.38 | 69.68 | 75.50 | 76.09 | 90.34 | 83.33 | 84.35 | 100.00 | 73.00 | 66.67 |
| | Recall@P | 47.00 | 42.80 | 26.80 | 27.40 | 27.00 | 47.80 | 22.80 | 98.00 | 74.80 | 100.00 | 88.40 | 92.40 | 98.40 | 76.00 |
| | F1@P | 56.29 | 54.25 | 40.67 | 41.33 | 40.91 | 56.70 | 35.02 | 85.66 | 81.84 | 90.91 | 86.33 | 96.05 | 83.82 | 71.03 |
| | Precision@N | 60.15 | 59.77 | 56.48 | 56.63 | 56.55 | 60.27 | 54.53 | 97.19 | 78.50 | 100.00 | 87.82 | 92.94 | 97.55 | 72.09 |
| | Recall@N | 80.00 | 85.00 | 95.00 | 94.80 | 95.00 | 79.20 | 92.60 | 69.20 | 92.00 | 80.00 | 83.60 | 100.00 | 63.60 | 62.00 |
| | F1@N | 68.67 | 70.19 | 70.84 | 70.91 | 70.90 | 68.45 | 68.64 | 80.84 | 84.71 | 88.89 | 85.66 | 96.34 | 77.00 | 66.67 |
| | AUC | 0.64 | 0.64 | 0.61 | 0.61 | 0.61 | 0.64 | 0.58 | 0.84 | 0.83 | 0.90 | 0.86 | 0.96 | 0.81 | 0.69 |
| | TPR@0.1FPR | 0.00 | 0.00 | 0.00 | 0.00 | 0.00 | 0.00 | 0.00 | 0.00 | 0.00 | 0.00 | 0.00 | 0.92 | 0.00 | 0.00 |
| SalUn | Accuracy | 63.50 | 61.30 | 61.30 | 61.40 | 61.40 | 63.50 | 56.50 | 99.60 | 77.20 | 100.00 | 99.80 | 67.20 | 99.80 | 99.80 |
| | Precision@P | 69.01 | 67.28 | 84.66 | 84.76 | 84.76 | 69.79 | 69.70 | 100.00 | 100.00 | 100.00 | 100.00 | 60.59 | 100.00 | 100.00 |
| | Recall@P | 49.00 | 44.00 | 27.60 | 27.80 | 27.80 | 47.60 | 23.00 | 99.20 | 54.40 | 100.00 | 99.60 | 98.40 | 99.60 | 100.00 |
| | F1@P | 57.31 | 53.20 | 41.63 | 41.87 | 41.87 | 56.60 | 34.59 | 99.60 | 70.47 | 100.00 | 99.80 | 75.00 | 99.80 | 99.80 |
| | Precision@N | 60.47 | 58.40 | 56.75 | 56.82 | 56.82 | 60.24 | 53.89 | 99.21 | 68.68 | 100.00 | 99.60 | 95.74 | 99.60 | 100.00 |
| | Recall@N | 78.00 | 78.60 | 95.00 | 95.00 | 95.00 | 79.40 | 90.00 | 100.00 | 100.00 | 100.00 | 100.00 | 36.00 | 100.00 | 99.60 |
| | F1@N | 68.12 | 67.01 | 71.05 | 71.11 | 71.11 | 68.51 | 67.42 | 99.60 | 81.43 | 100.00 | 99.80 | 52.33 | 99.80 | 99.80 |
| | AUC | 0.64 | 0.61 | 0.61 | 0.61 | 0.61 | 0.64 | 0.56 | 1.00 | 0.77 | 1.00 | 1.00 | 0.67 | 1.00 | 1.00 |
| | TPR@0.1FPR | 0.00 | 0.00 | 0.00 | 0.00 | 0.00 | 0.00 | 0.00 | 0.99 | 0.54 | 1.00 | 1.00 | 0.00 | 1.00 | 0.00 |
| SFR-on | Accuracy | 62.70 | 60.90 | 61.00 | 61.20 | 61.20 | 63.10 | 56.10 | 80.60 | 68.80 | 98.00 | 76.80 | 98.40 | 78.00 | 73.60 |
| | Precision@P | 68.30 | 64.23 | 81.61 | 83.33 | 82.18 | 68.77 | 72.93 | 72.43 | 84.56 | 96.15 | 68.82 | 100.00 | 69.89 | 69.41 |
| | Recall@P | 47.40 | 49.20 | 28.40 | 28.00 | 28.60 | 48.00 | 19.40 | 98.80 | 46.00 | 100.00 | 98.00 | 96.80 | 98.40 | 84.40 |
| | F1@P | 55.96 | 55.72 | 42.14 | 41.92 | 42.43 | 56.54 | 30.65 | 83.59 | 59.59 | 98.04 | 80.86 | 98.37 | 81.73 | 76.17 |
| | Precision@N | 59.72 | 58.83 | 56.66 | 56.73 | 56.78 | 60.06 | 53.52 | 98.11 | 62.91 | 100.00 | 96.53 | 96.90 | 97.30 | 80.10 |
| | Recall@N | 78.00 | 72.60 | 93.60 | 94.40 | 93.80 | 78.20 | 92.80 | 62.40 | 91.60 | 96.00 | 55.60 | 100.00 | 57.60 | 62.80 |
| | F1@N | 67.65 | 65.00 | 70.59 | 70.87 | 70.74 | 67.94 | 67.89 | 76.28 | 74.59 | 97.96 | 70.56 | 98.43 | 72.36 | 70.40 |
| | AUC | 0.63 | 0.61 | 0.61 | 0.61 | 0.61 | 0.63 | 0.56 | 0.81 | 0.69 | 0.98 | 0.77 | 0.98 | 0.78 | 0.74 |
| | TPR@0.1FPR | 0.00 | 0.00 | 0.00 | 0.00 | 0.00 | 0.00 | 0.00 | 0.00 | 0.00 | 0.00 | 0.00 | 0.97 | 0.00 | 0.00 |

Table 22: The performance of different MIA methods for CIFAR-100 under ResNet-18 across pretrain, retrain, SalUn and SFR-on (**full class unlearn setting**). The performance of each MIA method is evaluated on **retain data** (members) and **test data** (non-members).

| Model | Metric | Membership Inference Attack Algorithms | | | | | | | | | | | | | |
| | | Metric MIA | | | | | | ML-Leaks | | | BlindMI | | | |
| | | Lira | RMIA | Entropy | Modified Entropy | Confidence | Quantile | Merlin | ML-Leak1 | ML-Leak3 | Shadow | One-class | Diff-w | Diff-single | Diff-bi |
| Pre-train | Accuracy | 69.53 | 74.09 | 80.84 | 82.16 | 82.11 | 80.81 | 56.54 | 72.26 | 75.19 | 62.98 | 68.84 | 78.29 | 75.32 | 67.30 |
| | Precision@P | 70.06 | 66.30 | 75.53 | 76.31 | 76.33 | 73.99 | 57.22 | 64.35 | 74.86 | 57.46 | 61.62 | 69.84 | 67.06 | 63.13 |
| | Recall@P | 68.20 | 97.96 | 91.25 | 93.27 | 93.09 | 95.03 | 51.83 | 99.82 | 75.85 | 100.00 | 99.92 | 99.58 | 99.54 | 83.15 |
| | F1@P | 69.12 | 79.08 | 82.65 | 83.94 | 83.88 | 83.20 | 54.39 | 78.25 | 75.35 | 72.98 | 76.23 | 82.10 | 80.13 | 71.77 |
| | Precision@N | 69.02 | 96.09 | 88.95 | 91.35 | 91.14 | 93.05 | 55.97 | 99.59 | 75.53 | 100.00 | 99.79 | 99.26 | 99.10 | 75.33 |
| | Recall@N | 70.86 | 50.21 | 70.44 | 71.04 | 71.13 | 66.60 | 61.24 | 44.70 | 74.52 | 25.96 | 37.77 | 57.00 | 51.10 | 51.44 |
| | F1@N | 69.93 | 65.96 | 78.62 | 79.93 | 79.90 | 77.64 | 58.49 | 61.70 | 75.02 | 41.23 | 54.79 | 72.42 | 67.43 | 61.14 |
| | AUC | 0.75 | 0.82 | 0.82 | 0.83 | 0.83 | 0.83 | 0.51 | – | – | – | – | – | – | – |
| | TPR@0.1FPR | 0.01 | 0.19 | 0.09 | 0.09 | 0.09 | 0.00 | 0.02 | – | – | – | – | – | – | – |
| Retrain | Accuracy | 67.00 | 77.41 | 80.05 | 82.07 | 82.04 | 80.71 | 56.45 | 73.22 | 74.90 | 65.44 | 78.46 | 80.14 | 75.84 | 68.24 |
| | Precision@P | 71.20 | 69.49 | 74.59 | 76.31 | 76.22 | 73.88 | 57.36 | 65.27 | 73.53 | 59.13 | 72.22 | 72.17 | 67.80 | 64.61 |
| | Recall@P | 57.10 | 97.72 | 91.15 | 93.01 | 93.13 | 95.03 | 50.25 | 99.25 | 77.83 | 100.00 | 92.50 | 98.12 | 98.44 | 80.66 |
| | F1@P | 63.38 | 81.22 | 82.04 | 83.84 | 83.83 | 83.13 | 53.57 | 78.75 | 75.62 | 74.31 | 81.11 | 83.16 | 80.30 | 71.75 |
| | Precision@N | 64.19 | 96.16 | 88.62 | 91.05 | 91.17 | 93.04 | 55.74 | 98.44 | 76.45 | 100.00 | 89.58 | 97.07 | 97.16 | 74.27 |
| | Recall@N | 76.90 | 57.10 | 68.94 | 71.13 | 70.94 | 66.40 | 62.64 | 47.18 | 71.97 | 30.87 | 64.42 | 62.15 | 53.24 | 55.81 |
| | F1@N | 69.98 | 71.65 | 77.55 | 79.86 | 79.80 | 77.49 | 58.99 | 63.79 | 74.15 | 47.18 | 74.94 | 75.78 | 68.79 | 63.73 |
| | AUC | 0.71 | 0.86 | 0.81 | 0.82 | 0.82 | 0.83 | 0.51 | – | – | – | – | – | – | – |
| | TPR@0.1FPR | 0.01 | 0.22 | 0.06 | 0.07 | 0.07 | 0.00 | 0.01 | – | – | – | – | – | – | – |
| SalUn | Accuracy | 68.31 | 74.51 | 79.98 | 81.36 | 81.31 | 80.30 | 56.90 | 74.71 | 69.01 | 63.68 | 68.66 | 77.90 | 74.59 | 67.45 |
| | Precision@P | 69.50 | 66.72 | 74.85 | 75.76 | 75.89 | 73.28 | 57.50 | 66.59 | 72.34 | 57.93 | 61.50 | 69.63 | 66.58 | 63.80 |
| | Recall@P | 65.25 | 97.82 | 90.30 | 92.24 | 91.78 | 95.37 | 52.90 | 99.17 | 57.16 | 99.98 | 99.80 | 98.97 | 98.75 | 80.68 |
| | F1@P | 67.31 | 79.33 | 81.85 | 83.19 | 83.08 | 82.88 | 55.10 | 79.68 | 64.85 | 73.35 | 76.10 | 81.75 | 79.53 | 71.25 |
| | Precision@N | 67.25 | 95.91 | 87.78 | 90.08 | 89.60 | 93.38 | 56.39 | 98.38 | 65.37 | 99.93 | 99.46 | 98.22 | 97.58 | 73.73 |
| | Recall@N | 71.37 | 51.20 | 69.65 | 70.48 | 70.84 | 65.23 | 60.90 | 50.25 | 80.86 | 27.38 | 37.52 | 56.84 | 50.43 | 54.21 |
| | F1@N | 69.25 | 66.76 | 77.67 | 79.08 | 79.12 | 76.80 | 58.56 | 66.52 | 72.30 | 42.98 | 54.49 | 72.01 | 66.50 | 62.48 |
| | AUC | 0.73 | 0.83 | 0.82 | 0.83 | 0.83 | 0.83 | 0.51 | – | – | – | – | – | – | – |
| | TPR@0.1FPR | 0.01 | 0.19 | 0.08 | 0.08 | 0.08 | 0.00 | 0.02 | – | – | – | – | – | – | – |
| SFR-on | Accuracy | 68.81 | 73.13 | 78.33 | 80.30 | 80.20 | 78.86 | 56.88 | 58.66 | 65.04 | 63.06 | 72.55 | 77.16 | 74.20 | 66.84 |
| | Precision@P | 68.58 | 68.77 | 73.55 | 75.08 | 74.99 | 72.24 | 57.79 | 54.74 | 72.26 | 57.52 | 64.76 | 69.12 | 66.41 | 63.30 |
| | Recall@P | 69.43 | 84.72 | 88.48 | 90.71 | 90.62 | 93.76 | 51.06 | 99.94 | 48.84 | 99.92 | 98.93 | 98.18 | 97.94 | 80.14 |
| | F1@P | 69.00 | 75.92 | 80.33 | 82.16 | 82.07 | 81.60 | 54.22 | 70.74 | 58.28 | 73.01 | 78.28 | 81.13 | 79.15 | 70.73 |
| | Precision@N | 69.05 | 80.11 | 85.55 | 88.26 | 88.15 | 91.11 | 56.16 | 99.65 | 61.36 | 99.69 | 97.73 | 96.86 | 96.08 | 72.94 |
| | Recall@N | 68.20 | 61.53 | 68.18 | 69.89 | 69.77 | 63.97 | 62.70 | 17.38 | 81.25 | 26.21 | 46.17 | 56.13 | 50.45 | 53.55 |
| | F1@N | 68.62 | 69.60 | 75.88 | 78.01 | 77.89 | 75.17 | 59.25 | 29.59 | 69.92 | 41.50 | 62.71 | 71.08 | 66.16 | 61.76 |
| | AUC | 0.74 | 0.82 | 0.80 | 0.82 | 0.82 | 0.82 | 0.52 | – | – | – | – | – | – | – |
| | TPR@0.1FPR | 0.01 | 0.18 | 0.08 | 0.08 | 0.08 | 0.00 | 0.02 | – | – | – | – | – | – | – |

Table 23: The performance of different MIA methods for CIFAR-100 under ResNet-18 on retrain model (10% **random subset unlearn setting**). The performance of each MIA method is evaluated on **retain data** (members) and **forget data** (non-members).

| Model | Metric | Membership Inference Attack Algorithms | | | | | | | | | | | | | |
| | | Metric MIA | | | | | | ML-Leaks | | | BlindMI | | | |
| | | Lira | RMIA | Entropy | Modified Entropy | Confidence | Quantile | Merlin | MLLeak1 | MLLeak3 | Shadow | One-class | Diff-w | Diff-single | Diff-bi |
| Retrain | Accuracy | 66.22 | 78.58 | 80.48 | 82.74 | 82.66 | 81.12 | 58.60 | 76.32 | 69.00 | 65.90 | 77.18 | 80.66 | 76.68 | 68.50 |
| | Precision@P | 70.12 | 70.96 | 76.22 | 78.18 | 78.09 | 74.50 | 59.13 | 68.16 | 74.04 | 59.46 | 73.04 | 72.70 | 68.56 | 64.98 |
| | Recall@P | 56.52 | 96.76 | 88.60 | 90.84 | 90.80 | 94.64 | 55.68 | 98.80 | 58.52 | 99.96 | 86.16 | 98.20 | 98.56 | 80.24 |
| | F1@P | 62.59 | 81.88 | 81.95 | 84.03 | 83.97 | 83.37 | 57.35 | 80.67 | 65.37 | 74.56 | 79.06 | 83.55 | 80.87 | 71.81 |
| | Precision@N | 63.58 | 94.91 | 86.39 | 89.07 | 89.01 | 92.65 | 58.13 | 97.82 | 65.71 | 99.87 | 83.13 | 97.23 | 97.44 | 74.18 |
| | Recall@N | 75.92 | 60.40 | 72.36 | 74.64 | 74.52 | 67.60 | 61.52 | 53.84 | 79.48 | 31.84 | 68.20 | 63.12 | 54.80 | 56.76 |
| | F1@N | 69.21 | 73.82 | 78.75 | 81.22 | 81.12 | 78.17 | 59.77 | 69.45 | 71.94 | 48.29 | 74.93 | 76.55 | 70.15 | 64.31 |
| | AUC | 0.70 | 0.87 | 0.81 | 0.83 | 0.83 | 0.84 | 0.50 | – | – | – | – | – | – | – |
| | TPR@0.1FPR | 0.00 | 0.20 | 0.13 | 0.14 | 0.14 | 0.00 | 0.03 | – | – | – | – | – | – | – |

Table 24: The performance of LiRA and RMIA methods for CIFAR-100 under ResNet-18 with different numbers of shadow models .

| Epochs | Metrics | Mode | LIRA | | | | RMIA | | |
|---|---|---|---|---|---|---|---|---|---|
| | | # shadow | 5 | 16 | 32 | 64 | 5 | 16 | 32 |
| 200 epochs | Acc | Attack | 65.77 | 67.45 | 68.07 | 67.97 | 72.62 | 74.10 | 73.17 |
| | | Audit | 69.37 | 72.24 | 72.55 | 72.65 | 75.18 | 74.39 | 74.29 |
| | Prec@P | Attack | 69.41 | 75.78 | 75.87 | 73.02 | 73.02 | 70.50 | 72.58 |
| | | Audit | 70 | 77.63 | 77.32 | 77.09 | 69.26 | 66.20 | 66.10 |
| | Rec@P | Attack | 56.4 | 51.3 | 53 | 53.34 | 71.16 | 82.88 | 74.48 |
| | | Audit | 67.8 | 62.48 | 63.82 | 64.46 | 90.54 | 99.68 | 99.74 |
| | F1@P | Attack | 62.23 | 61.18 | 62.4 | 62.48 | 72.38 | 76.19 | 73.52 |
| | | Audit | 68.88 | 69.24 | 69.92 | 70.21 | 78.48 | 79.56 | 79.51 |
| | Prec@N | Attack | 63.2 | 63.19 | 63.89 | 63.9 | 72.24 | 79.23 | 73.79 |
| | | Audit | 68.78 | 68.61 | 69.2 | 69.46 | 86.35 | 99.35 | 99.47 |
| | Rec@N | Attack | 75.14 | 83.6 | 83.14 | 82.6 | 73.48 | 65.32 | 71.86 |
| | | Audit | 70.94 | 82 | 81.28 | 80.84 | 59.82 | 49.10 | 48.84 |
| | F1@N | Attack | 68.7 | 71.98 | 72.25 | 77.06 | 72.85 | 71.61 | 72.81 |
| | | Audit | 69.84 | 74.71 | 74.75 | 74.72 | 70.76 | 65.72 | 65.51 |
| | Auc | Audit | 0.745 | 0.772 | 0.774 | 0.776 | 0.839 | 0.839 | 0.839 |
| | PR@0.1 FPF | Audit | 2.0 | 4.0 | 6.0 | 7.4 | 9.3 | 11.0 | 11.0 |
| 110 epochs | Acc | Attack | 62.49 | 64.41 | 64.37 | 64.36 | 72.33 | 73.28 | 73.11 |
| | | Audit | 66.72 | 69.06 | 69.46 | 69.68 | 74.23 | 74.31 | 73.61 |
| | Prec@P | Attack | 63.81 | 70.4 | 68.92 | 70.87 | 71.71 | 70.20 | 71.31 |
| | | Audit | 67.17 | 71.35 | 73.7 | 73.94 | 67.36 | 67.30 | 65.88 |
| | Rec@P | Attack | 57.7 | 58.37 | 52.34 | 48.76 | 73.76 | 80.90 | 77.34 |
| | | Audit | 65.4 | 63.7 | 60.52 | 60.78 | 94.00 | 94.56 | 97.94 |
| | F1@P | Attack | 60.6 | 49.9 | 59.5 | 57.77 | 72.72 | 75.17 | 74.20 |
| | | Audit | 66.27 | 67.31 | 66.46 | 66.72 | 78.48 | 78.64 | 78.77 |
| | Prec@N | Attack | 61.4 | 61.17 | 61.58 | 60.95 | 72.99 | 77.47 | 75.25 |
| | | Audit | 66.29 | 67.21 | 66.51 | 66.71 | 90.08 | 90.86 | 95.99 |
| | Rec@N | Attack | 67.28 | 78.92 | 76.4 | 79.96 | 70.90 | 65.66 | 68.88 |
| | | Audit | 68.04 | 74.42 | 78.4 | 78.58 | 54.46 | 54.06 | 49.28 |
| | F1@N | Attack | 64.2 | 68.92 | 68.2 | 69.17 | 71.93 | 71.08 | 71.92 |
| | | Audit | 67.15 | 70.63 | 71.92 | 72.16 | 67.88 | 67.79 | 65.12 |
| | Auc | Audit | 0.711 | 0.74 | 0.742 | 0.744 | 0.830 | 0.833 | 0.832 |
| | PR@0.1 FPF | Audit | 1.4 | 3.8 | 3.2 | 5.7 | 6.7 | 8.5 | 7.3 |
| 10 epochs | Acc | Attack | 52.35 | 52.5 | 53.38 | 53.12 | 53.79 | 53.65 | 53.64 |
| | | Audit | 52.46 | 53.01 | 53.19 | 53.09 | 53.91 | 53.98 | 53.96 |
| | Prec@P | Attack | 54.56 | 54.86 | 56.25 | 56.39 | 53.27 | 53.46 | 53.29 |
| | | Audit | 52.88 | 53.51 | 53.6 | 53.51 | 53.32 | 53.48 | 53.38 |
| | Rec@P | Attack | 28.1 | 28.2 | 30.44 | 27.54 | 61.80 | 60.14 | 58.43 |
| | | Audit | 45.14 | 45.9 | 47.44 | 47.08 | 62.80 | 61.16 | 62.56 |
| | F1@P | Attack | 37.1 | 37.25 | 39.5 | 37.01 | 57.22 | 56.60 | 55.97 |
| | | Audit | 48.71 | 49.41 | 50.33 | 50.09 | 54.76 | 57.06 | 57.61 |
| | Prec@N | Attack | 51.58 | 51.68 | 52.32 | 52.06 | 54.51 | 54.45 | 54.07 |
| | | Audit | 52.15 | 52.64 | 52.86 | 52.76 | 54.76 | 54.65 | 54.78 |
| | Rec@N | Attack | 76.6 | 76.8 | 76.32 | 78.7 | 45.78 | 47.64 | 48.34 |
| | | Audit | 59.78 | 60.12 | 58.94 | 59.1 | 45.02 | 46.80 | 45.36 |
| | F1@N | Attack | 61.65 | 61.79 | 62.08 | 62.67 | 49.77 | 50.82 | 51.05 |
| | | Audit | 55.7 | 56.13 | 55.74 | 55.75 | 49.91 | 50.42 | 49.36 |
| | Auc | Audit | 0.526 | 0.532 | 0.535 | 0.536 | 0.549 | 0.550 | 0.550 |
| | PR@0.1 FPF | Audit | 0.02 | 0.28 | 0.38 | 0.2 | 0.1 | 0.2 | 0.2 |

