# OpenReview forum: "MIABench: Full-Pipeline Evaluation of Membership Inference Attacks"
_ICLR.cc/2026/Conference — Submitted to ICLR 2026_

### Official Review · Reviewer_GeVM · 2025-10-26

**Soundness:** 3
**Presentation:** 3
**Contribution:** 2
**Rating:** 6
**Confidence:** 4

**Summary:**

This paper proposes a benchmark suite that streamlines the application of diverse membership inference attacks (MIAs) based on the evaluation criteria (attack or audit) for the purpose of evaluating various MIA methods on deep learning models. MIA performance is used as a metric to assess privacy-preserving capabilities of machine learning algorithms, but there is a dearth of work which allows for fair and comprehensive comparison of diverse MIA approaches.

**Strengths:**

- The benchmark suite in this paper supports the evaluation of different MIAs within a common [black-box] threat model, thereby making any results obtained from using the code base fair and comparable.
- In attack mode, the suite selects hyperparameters for MIA, such as the scaler threshold, using the shadow models.
- The suite also allows for using MIAs to evaluate the performance of different machine unlearning methods.
- It provides evidence in support of the argument that metric-based MIAs, which are tailored to be sensitive towards membership signal as compared to non-membership signal, might be a poor choice to judge the efficiency of machine unlearning methods. This is because the samples "unlearned" via these approaches ought to be confidently regarded as non-members by MIAs to claim efficient unlearning, and this is a task for which SOTA metric-based MIA methods might yield misleading estimates.
- In section G1, they comprehensively detail how the utility of the model is affected in different experiments undertaken in the paper.
- The author(s) have shared the code for review.

**Weaknesses:**

- In lines 317-319, the authors mention that using DP-SGD to train the models does not compromise their utility, but they do not quantify the privacy guarantees using any conventional metrics of DP, such as $\epsilon$ in $\epsilon$-DP [1] or $\mu$ in $\mu$-GDP [2].
- LiRA is known to perform poorly with a low # of shadow models (such as the 5 used in the paper). Compared to that, RMIA offers much better results, as evident from your experiments. Using a weaker version of the attack does not make for a fair comparison against other MIAs in the experiments.
- In the standard MIA threat model (the attack mode), the target model's training hyperparameters are presumed to be known and are repurposed for training the shadow models. If we go by the author(s)'s differentiation between attack and audit mode based on the knowledge of the training dataset, knowledge of training hyperparameters would also be a feasible assumption in the latter, but not always in the former. The author(s) do not account for this in their benchmark suite's MIA pipeline.

[1] Dwork et al. “Calibrating Noise to Sensitivity in Private Data Analysis.” TCC 2006.

[2] Dong, J. et al. “Gaussian Differential Privacy.” CoRR 2019.

**Questions:**

**Questions**: This work is a commendable step towards creating a benchmark suite for evaluating different MIA configurations in a fair manner, allowing comparison of performance across multiple evaluation metrics. Although there are parts of it that need to be addressed, as listed in the weaknesses. I am amenable to revising my initial assessment provided that the authors can address the concerns highlighted in the weaknesses as listed above.

**Suggestions**: The following minor corrections/ suggestions can help improve the presentation of the paper:

- Minor Correction #1: Figure 2(a) says "Epoch" along the x-axis when it should be mislabel rate as stated in the caption.
- Minor Correction #2: Line 291 mentions 2 sentences which imply the same thing. It is an unnecessary repetition.
- Minor Correction #3: In line 678, it should be "we then use", not "we then then use".
- Minor Correction #4: In line 705, missing parentheses around Bertran et al. (2024).
- Minor Correction #5: Line 708 "RMIA: RMIAZarifzadeh et al. (2024)" is missing space indentation and parentheses around the reference. Also, in the same line, it's "LiRA", not "LiRa".
- Minor Suggestion: Most results/ claims are made with respect to MIA accuracy, which papers like Carlini et al. (2022) have asserted is a metric that weighs non-members and members equally. Instead, using F1 or Recall (TPR) would be much more appropriate as metrics. For example, a high F1@P (or low F1@N) would imply that MIAs are more efficient at detecting members, which is significantly more detrimental as a privacy risk than correctly identifying non-members (@N).

---

> ### Author Response · Authors · 2025-11-22
>
> **Q1: Privacy Loss of DP-SGD**
>
> **A1:** We did not report the privacy loss in our paper because it is excessively large and thus meaningless. Moreover, a reasonable privacy loss would lead to significant utility degradation, as supported by [1]. Additionally, we observe that a small amount of calibrated Gaussian noise can protect the model from membership inference attacks without severe utility degradation. Therefore, we do not provide the exact privacy loss.
>
> [1] De S, Berrada L, Hayes J, et al. Unlocking high-accuracy differentially private image classification through scale[J]. arXiv preprint arXiv:2204.13650, 2022.
>
> **Q2: The performance of LiRA**
>
> **A2:** In our main experiments, we used the same shadow model to ensure a fair comparison. Table 24 further presents the performance of LiRA and RMIA with varying numbers of shadow models. The results demonstrate that while increasing the number of shadow models can enhance LiRA's performance, this improvement quickly saturates and does not affect the main conclusions of our study.
>
> **Q3: The misleading of threat model:**
>
> **A3:** Our study does not consider the training hyperparameters of the target model. In both auditing and attacking modes, we use the same target model and shadow model to ensure a fair comparison.
>
> **Suggestions and correctness**
>
> **A4:** Thanks for your suggestion, we will fix these problems in the revised version. Moreover, our experiments results usually provides the result for both accuracy and TPR@0.1%FPR (e.g. Table4)

---

> > ### Comment · Reviewer_GeVM · 2025-11-25
> > **Follow-up To The Author(s) Response**
> >
> > Thanks for your response. After going through the comments/concerns highlighted by other reviewers, I have decided to retain my score.

---

### Official Review · Reviewer_ff2B · 2025-10-26

**Soundness:** 2
**Presentation:** 2
**Contribution:** 2
**Rating:** 2
**Confidence:** 3

**Summary:**

This paper proposes a reproducible benchmark suite that standardizes the evaluation process of membership inference attacks. It introduces two distinct evaluation scenarios: an "attacking mode" (where the adversary lacks membership knowledge for hyperparameter tuning and must rely on shadow models) and an "auditing mode" (where a model owner has access to membership information to find the optimal attack hyperparameters). Using this benchmark, the paper evaluates a wide range of MIA methods across different datasets, model architectures, and training algorithms. The key findings highlight that a model's generalization gap (the difference between training and test accuracy) is a primary indicator of its vulnerability to MIAs. The authors also observe a high agreement among the predictions of the top-performing attacks, suggesting they exploit similar model properties. Finally, the paper provides a practical guide for selecting appropriate MIA methods based on the evaluation scenario.

**Strengths:**

* Comprehensive Pipeline: The paper's main strength is its holistic approach, evaluating MIAs across the machine learning pipeline (data, model, training algorithm, and hyperparameter selection) rather than in an isolated step.
*  The paper provides practical takeaways. The strong correlation found between the generalization gap and MIA vulnerability provides a good metric for developers to monitor.
* Extensive Experimentation: The benchmark is tested across numerous variables: MIA methods, datasets, architectures, and training algorithms (including DP-SGD and machine unlearning), providing a robust empirical foundation for the paper's conclusions.

**Weaknesses:**

* Scope Limited to Vision: As acknowledged by the authors, the benchmark is currently focused exclusively on computer vision datasets (CIFAR-10, CIFAR-100, ImageNet100) and classification models. The findings may not directly generalize to other domains, such as NLP, time-series, or generative models, which have different data characteristics and model architectures.
* Focus on Black-Box Access: The study limits its scope to black-box attacks where the adversary only has access to the model's output logits. While common, this excludes more powerful white-box attacks (which have access to gradients or parameters) that are also a significant threat.
* Overly Pessimistic Attacker Model: The "attacking mode" assumes an adversary with minimal knowledge of the training data distribution. This may not represent sophisticated, real-world attackers who often have strong prior knowledge (e.g., knowing a model was trained on public data like Common Crawl or Wikipedia), allowing them to create far more accurate shadow models and blur the distinction between the "attacking" and "auditing" scenarios.

**Questions:**

- Could the benchmark be extended to include a scenario that models an adversary with partial, high-level knowledge of the training data distribution (e.g., "trained on public web data"), bridging the gap between the current "attacking" and "auditing" modes?

- How much more effective would these MIAs be in a white-box scenario?

---

> ### Author Response · Authors · 2025-11-22
>
> **W1: Scope Limited to Vision**
>
> **A1:** As outlined in our paper, the primary motivation is to establish a **fair and comprehensive benchmark** for evaluating the performance of various membership inference attack (MIA) methods. We recognize that these methods often adopt divergent assumptions. For instance, approaches such as LiRA and RMIA operate under the premise that MIA serves as an audit method, where adversaries leverage member/non-member information to optimize hyper-parameters. In contrast, other methods, including shadow-MIA and metric-MIA, assume that adversaries lack access to such information. The training configurations (data, architecture, training trajectory, etc.) for different target models also exhibit considerable variability among different papers.
>
> To address this, our study first defines a standardized training pipeline and subsequently assesses the performance of diverse MIA methods under controlled variations in specific pipeline components. To the best of our knowledge, this constitutes the first MIA benchmark that fairly and comprehensively evaluates MIA methodologies by accounting for variations across each segment of the pipeline.
>
>
> As for the multimodal data and generative model. We argue that research in this area cannot be effectively integrated into our benchmark because **the definition of MIA is unclear for  multimodal data and generative models**. The conventional membership inference game defines a member as a precisely identical record present in the training data. However, this definition may fail to adequately capture information leakage in generative modeling paradigms. As discussed in the Section 5 of  [1], adversaries and privacy auditors might be concerned about semantically similar samples, even in the absence of an exact match.
>
> [1] Duan M, Suri A, Mireshghallah N, et al. Do membership inference attacks work on large language models?[J]. arXiv preprint arXiv:2402.07841, 2024.
>
>
> **W2: Focus on Black-Box Access Only**
>
> A2: Thanks for your suggestion. In our review of membership inference attack (MIA) research, we found that black-box attacks are the predominant approach, with limited methods available for white-box attacks. Moreover, white-box attacks are often associated with differential privacy, particularly for estimating the performance of DP-SGD. Additionally, [2] theoretically demonstrates that white-box attacks (with access to all parameters) do not provide more information than black-box attacks (which only access the loss).  On the other hand, the incorporation of white-box attacks would necessitate modifications to the methodological pipeline and the underlying assumptions. Consequently, to maintain a concise and well-structured presentation, we have excluded white-box attacks from the current study.
>
> [2] Sablayrolles A, Douze M, Schmid C, et al. White-box vs black-box: Bayes optimal strategies for membership inference[C]//International Conference on Machine Learning. PMLR, 2019: 5558-5567.
>
> **W3:  Attacker Model Overly Pessimistic**
>
> **A3:** Sorry for the misunderstanding, the  "attacking mode" assumes an adversary knows the training data’s distribution. It is not pessimistic as it is the standard assumption for some MIA methods [3,4].  Sorry we do not highlight this part in our paper, we will fix this problem in the revised version.
>
> [3] Reza Shokri, Marco Stronati, Congzheng Song, and Vitaly Shmatikov. Membership inference attacks against machine learning models. In 2017 IEEE symposium on security and privacy (SP), pp. 3–18. IEEE, 2017.
>
> [4] Hui B, Yang Y, Yuan H, et al. Practical blind membership inference attack via differential comparisons[J]. arXiv preprint arXiv:2101.01341, 2021.
>
> **Q1:Misleading of Attacking Mode and Auditing Mode**
>
> **A1:** Thanks for your question. As we have claimed in answering weakness 3, the  "attacking mode" assumes an adversary knows the training data’s distribution.
>
> The distinction between "attacking mode" and "auditing mode" lies in whether adversaries have access to true member/non-member information, which enables hyper-parameter selection. We introduced these modes to accommodate the differing assumptions in MIA methods. For instance, approaches like LiRA and RMIA treat MIA as an auditing tool, where adversaries leverage member/non-member data to optimize hyper-parameters. In contrast, methods such as shadow-MIA and metric-MIA assume that adversaries cannot access.
>
> **Q2: Effectiveness of white-box MIA?**
>
> **A2:** Thanks for your suggestion, as we have answered in weakness 2. Theoretically, white-box attacks (with access to all parameters) do not provide more information than black-box attacks (which only access the loss). On the other hand, the incorporation of white-box attacks would necessitate modifications to the methodological pipeline and the underlying assumptions. Consequently, to maintain a concise and well-structured presentation, we have excluded white-box attacks from the current study.

---

### Official Review · Reviewer_Zs8w · 2025-10-31

**Soundness:** 2
**Presentation:** 3
**Contribution:** 1
**Rating:** 2
**Confidence:** 4

**Summary:**

This paper studies the problem of membership inference attacks (MIAs), a key technique for evaluating a model’s vulnerability to privacy leakage by determining whether specific data samples were part of its training set. In particular, the paper attempts to conduct a systematic analysis of the entire MIA pipeline and introduces a benchmark for evaluating and comparing different MIA methods across standard deep learning models and benchmark datasets.

**Strengths:**

- Benchmarking privacy leakage is of practical importance, and this is a timely contribution that provides a useful benchmark.

- The presentation is generally clear and the paper is well structured overall.

- The experimental evaluation is extensive in quantity

**Weaknesses:**

- It is unclear what is new or novel in this work. The paper seems present very limited new insights or findings for the community,
the key observations that *MIA performance correlates with the generalization gap* and that *hyperparameter selection plays a significant role* are already well established in the privacy literature.

- The data modality is limited to images and the models are restricted to CNN architectures. While this setup is standard in the MIA literature, it lacks diversity, which is limiting for a paper that aims to advance the frontier of MIA benchmarking in the current community.

- The paper lacks a clear description of the threat model, i.e., it does not specify explicitly what information or access each attack assumes. For example, based on this submission alone, it remains unclear whether each method has knowledge of the full dataset, partial dataset, partial data distribution, or whether the attacks operate in a white-box or black-box setting (e.g., logit or label access). The current comparisons are therefore not entirely transparent, even though some settings (such as the distinction between attacking vs. auditing modes) are implicitly defined.

**Questions:**

- Section 2 Table 1: It would be better to directly include references to the corresponding papers in the table (or clearly indicate the naming correspondence in the text). As it stands, it is not entirely clear which paper each entry or column in the table refers to. While some of this information is available in the Appendix, it would still be helpful to make the mapping more explicit within the main text or table itself.

- The current distinction between attacking and auditing modes is somewhat confusing. In this paper, the difference appears to be limited to how hyperparameters are selected, whereas in the broader literature, auditing typically refers to a stronger attack assumption that often reflecting a worst-case scenario where the adversary can actively select or construct “hard examples” (e.g., canaries) to better probe model privacy. This conceptual misalignment makes the terminology here potentially misleading.

- Related to the above point (see "Weakness" section): the paper is not fully self-contained. While this may be understandable for a large-scale benchmark, it would still be very helpful if the authors could provide clearer and more detailed descriptions of each method used in the experiments, e.g., summarizing the key ideas, main formulas, and assumed threat models for each attack under a consistent notation framework.

---

> ### Author Response · Authors · 2025-11-22
>
> **Weakness1: The novelty of this paper**
>
> **A1:** Thanks for your question. As outlined in our paper, the primary motivation is to establish a **fair and comprehensive benchmark** for evaluating the performance of various membership inference attack (MIA) methods. We recognize that these methods often adopt divergent assumptions. For instance, approaches such as LiRA and RMIA operate under the premise that MIA serves as an audit method, where adversaries leverage member/non-member information to optimize hyper-parameters. In contrast, other methods, including shadow-MIA and metric-MIA, assume that adversaries lack access to such information. The training configurations (data, architecture, training trajectory, etc.) for different target models also exhibit considerable variability among different papers.
>
> To address this, our study first defines a standardized training pipeline and subsequently assesses the performance of diverse MIA methods under controlled variations in specific pipeline components. To the best of our knowledge, this constitutes the first MIA benchmark that fairly and comprehensively evaluates MIA methodologies by accounting for variations across each segment of the pipeline.
>
> We would like to provide a brief review of the MIA evaluation framework to highlight the novelty of our work . [1] examines the impact of different data characteristics on MIA, [2] investigates MIA performance under heterogeneous datasets. However, no existing work comprehensively evaluates various MIA methods from a full pipeline perspective, considering their assumptions to provide a **fair and comprehensive** comparison.
>
>
> [1] Niu J, Zhu X, Zeng M, et al. Comparing Different Membership Inference Attacks with a Comprehensive Benchmark[J]. IEEE Transactions on Information Forensics and Security, 2025.
> [2] van Dartel B, Damie M, Hahn F. Evaluating Membership Inference Attacks in heterogeneous-data setups[C]//International Conference on Applied Cryptography and Network Security. Cham: Springer Nature Switzerland, 2025: 109-117.
>
>
> **Weakness2: Data modality Issues**
>
> **A2:** As for the multimodal data and generative model. We argue that research in this area cannot be effectively integrated into our benchmark because **the definition of MIA is unclear for  multimodal data and generative models**. The conventional membership inference game defines a member as a precisely identical record present in the training data. However, this definition may fail to adequately capture information leakage in generative modeling paradigms. As discussed in the Section 5 of  [3], adversaries and privacy auditors might be concerned about semantically similar samples, even in the absence of an exact match.
>
> [3] Duan M, Suri A, Mireshghallah N, et al. Do membership inference attacks work on large language models?[J]. arXiv preprint arXiv:2402.07841, 2024.
>
>
> **Weakness3: lacks a clear description of the threat model**
>
> **A3:** This paper focuses exclusively on black-box attacks. To delineate the data accessibility in membership inference attack (MIA) methods, we introduce two distinct modes: auditing and attacking. In the auditing mode, adversaries are assumed to have full access to the training dataset. Conversely, in the attacking mode, adversaries possess knowledge of the training data distribution. We would like to clarify this information in the revised version.
>
> **Q1: Suggestions to improve the clarify of each MIA method in the main body**
>
> **A1:**  Thanks for your suggestions, we would like to improve the Table 1 in our revised version.
>
> **Q2: Confusing about the terminology:**
>
> **A2:** Privacy auditing is a new research topic that involves empirically verifying a model's privacy guarantee under specific conditions, such as using canaries. Our naming convention is inspired by this approach, as in audit mode, we similarly focus on empirically evaluating the best performance of membership inference attacks (MIA). However, our auditing method does not incorporate any special settings. We recognize that this may cause confusion, and we will address these distinctions in more detail in the revised version.
>
> **Q3:  Summarizing the key ideas, main formulas, and assumed threat models for each attack under a consistent notation framework**
>
> **A3:** The key ideas and a concise introduction for each membership inference attack (MIA) are detailed in the Appendix. We would like to enhance the clarity of these elements in the revised version. However, constructing a unified framework to summarize the key ideas, main formulas, and assumed threat models poses considerable challenges, as this represents a novel undertaking in the field that does not align with our paper’s motivation.

---

### Official Review · Reviewer_Ctoz · 2025-11-01

**Soundness:** 3
**Presentation:** 2
**Contribution:** 2
**Rating:** 4
**Confidence:** 4

**Summary:**

This paper presents MIABench, a comprehensive benchmark for evaluating Membership Inference Attacks (MIAs) on classical deep learning classifiers under black-box access. MIABench includes a reproducible suite (including code, datasets, models, etc.) and defines two practical evaluation modes, which are Attacking mode and Auditing mode. The extensive experiments have explored the impact of mutiple different factors on MIAs.

**Strengths:**

- Comprehensive Coverage: evaluations span critical dimensions like data, architecture, training, hyperparameters and metrics.
- Utility for the Community: rhe benchmark’s reproducibility and extensibility enable future MIA research.

**Weaknesses:**

- The MIABench only targets traditional visual classification models, e.g., CNN models.However, most state-of-the-art MIAs have spread to more models and scenarios, like [1] and [2]. It does not address multimodal data (text, audio) or the state-of-the-art large generative models (e.g., GPT, diffusion models, image autoregressive models), which are increasingly relevant for privacy research.
- The table formatting is overly crude and non-standardized. For instance, Table 3 lacks dividing lines. The author is requested to carefully review the entire manuscript for similar standardization issues.
- The related work mentions MIA methods but does not explicitly compare MIABench to existing MIA evaluation frameworks to highlight unique advantages.

[1] Yu H, Qiu Y, Yang Y, et al. Icas: Detecting training data from autoregressive image generative models[J]. arXiv preprint arXiv:2507.05068, 2025.

[2] Kowalczuk A, Dubiński J, Boenisch F, et al. Privacy Attacks on Image AutoRegressive Models[C]//Forty-second International Conference on Machine Learning.

**Questions:**

See Weaknesses

---

> ### Author Response · Authors · 2025-11-22
>
> **Q1: Do not consider the multimodal data and generative model:**
>
> **A1:** As outlined in our paper, the primary motivation is to establish a **fair and comprehensive benchmark** for evaluating the performance of various membership inference attack (MIA) methods. We recognize that these methods often adopt divergent assumptions. For instance, approaches such as LiRA and RMIA operate under the premise that MIA serves as an audit method, where adversaries leverage member/non-member information to optimize hyper-parameters. In contrast, other methods, including shadow-MIA and metric-MIA, assume that adversaries lack access to such information. The training configurations (data, architecture, training trajectory, etc.) for different target models also exhibit considerable variability among different papers.
>
> To address this, our study first defines a standardized training pipeline and subsequently assesses the performance of diverse MIA methods under controlled variations in specific pipeline components. To the best of our knowledge, this constitutes the first MIA benchmark that fairly and comprehensively evaluates MIA methodologies by accounting for variations across each segment of the pipeline.
>
>
> As for the multimodal data and generative model. We argue that research in this area cannot be effectively integrated into our benchmark because **the definition of MIA is unclear for  multimodal data and generative models**. The conventional membership inference game defines a member as a precisely identical record present in the training data. However, this definition may fail to adequately capture information leakage in generative modeling paradigms. As discussed in the Section 5 of  [1], adversaries and privacy auditors might be concerned about semantically similar samples, even in the absence of an exact match.
>
> [1] Duan M, Suri A, Mireshghallah N, et al. Do membership inference attacks work on large language models?[J]. arXiv preprint arXiv:2402.07841, 2024.
>
> **Q2: The table formatting is overly crude and non-standardized.**
>
> **A2:** Thanks for your suggestion, we will fix these problems in the revised version.
>
>
> **Q3: The related work mentions MIA methods but does not explicitly compare MIABench to existing MIA evaluation frameworks to highlight unique advantages.**
>
> **A3:** Thanks for your suggestions. There is a scarcity of literature on evaluation frameworks for membership inference attacks (MIA), which motivates our development of an MIA benchmark. Specifically, [2] examines the impact of different data characteristics on MIA, [3] investigates MIA performance under heterogeneous datasets. However, no existing work comprehensively evaluates various MIA methods from a full pipeline perspective, considering their assumptions to provide a fair and comprehensive comparison.
>
>
> [2] Niu J, Zhu X, Zeng M, et al. Comparing Different Membership Inference Attacks with a Comprehensive Benchmark[J]. IEEE Transactions on Information Forensics and Security, 2025.
>
> [3] van Dartel B, Damie M, Hahn F. Evaluating Membership Inference Attacks in heterogeneous-data setups[C]//International Conference on Applied Cryptography and Network Security. Cham: Springer Nature Switzerland, 2025: 109-117.

---

### Meta-Review · Area_Chair_dJ5B · 2025-12-12

**Summary:**

The reviewers raised a number of concerns, including limited scope excluding generative models, unclear presentation of the evaluated methods, unclear presentation of the threat and evaluation models, choice of the number of shadow models that is clearly suboptimal for LiRA.

As additional concerns not raised by reviewers, I would add:
1. Lack of systematic uncertainty measures (Fig. 2, most tables), and use of misleading uncertainty measure (variance instead of standard deviation) in Tables 3 and 4 where these are provided.
2. Lack of transparency for unusually small number of shadow models which according to previous results and some presented results will seriously harm the performance of many attacks, especially LiRA. This is potentially dangerous, as readers may be led to believe that five shadow models is always good enough, and therefore use weak attacks that lead to overoptimistic privacy evaluations (cf. e.g. arXiv:2403.01218). The paper should be transparent of the unusual choice and its implications. An important recent reference to include is arXiv:2505.18773 that clearly shows that five shadow models is far too little for good performance for LiRA.

**Reviewer Concerns:**

The author rebuttal provides some clarifications and promises corrections, but no actual revisions have been made to address the concerns.

**Reviewer Scores:**

I do not see that the author response, especially lacking a revision, would have addressed the reviewer concerns.

---

### Decision · Program_Chairs · 2026-01-26

Reject